# Networked Communication for Decentralised Agents in Mean-Field Games

## Abstract

We introduce networked communication to the mean-field game framework, in particular to oracle-free settings where $N$ decentralised agents learn along a single, non-episodic run of the empirical system. We prove that our architecture has sample guarantees bounded between those of the centralised- and independent-learning cases. We provide the order of the difference in these bounds in terms of network structure and number of communication rounds, and also contribute a policy-update stability guarantee. We discuss how the sample guarantees of the three theoretical algorithms do not actually result in practical convergence times. We thus contribute practical enhancements to all three algorithms allowing us to present their first empirical demonstrations, where we do not need to enforce several of the theoretically required assumptions. We then show that in practical settings where the theoretical hyperparameters are not observed (leading to poor estimation of the Q-function), our communication scheme considerably accelerates learning over the independent case, which hardly seems to learn at all. Indeed networked agents often perform similarly to the centralised case, while removing the restrictive assumption of the latter. We provide ablations and additional studies showing that our networked approach also has advantages over both alternatives in terms of robustness to update failures and to changes in population size.

## 1 Introduction

The mean-field game (MFG) framework (Huang et al., 2006; Lasry & Lions, 2007) models a representative agent as interacting not with the other individuals in the population on a per-agent basis, but instead with a distribution of other agents, known as the *mean field*. The framework analyses the limiting case when the population consists of an infinite number of symmetric and anonymous agents, that is, they have identical reward and transition functions which depend on the mean field rather than on the actions of specific other players. In this work we focus on MFGs with stationary population distributions ('stationary MFGs', where learning is more tractable than in non-stationary ones) (Xie et al., 2021; Anahtarci et al., 2023; Zaman et al., 2023; Yardim et al., 2023; Li et al., 2025b; Osborne & Smears, 2025), for which the solution concept is the MFG-Nash equilibrium (MFG-NE), which reflects the situation when each agent responds optimally to the population distribution that arises when all other agents follow that same optimal behaviour. The MFG-NE can be used as an approximation for the Nash equilibrium (NE) in a finite-agent game, with the error in the solution reducing as the number of agents $N$ tends to infinity (Saldi et al., 2018; Anahtarci et al., 2023; Yardim et al., 2024; Toumi et al., 2024; Hu & Zhang, 2024; Chen et al., 2024e; Yang et al., 2025; Yardim et al., 2025).

MFGs can therefore be used to address the difficulty faced by multi-agent reinforcement learning (MARL), where it has been computationally difficult to scale algorithms beyond configurations with agents numbering in the low tens, as the joint state and action spaces grow exponentially with the number of agents (Daskalakis et al., 2006; Vinyals et al., 2019; Mcaleer et al., 2020; Shavandi & Khedmati, 2022; Li et al.; Yardim & He, 2024). The value of reasoning about interactions among very large populations of agents has been recognised, and an informal distinction is sometimes drawn between multi- and *many*-agent systems (Zheng et al., 2018; Wang et al., 2020a; Cui et al., 2022). The latter situation can be more useful (as in cases where better

solutions arise from the presence of more agents (Shiri et al., 2019; Ornia et al., 2022; Orr & Dutta, 2023; Eck et al., 2023)), more parallelisable (Andréen et al., 2016), more fault tolerant (Chang et al., 2023), or otherwise more reflective of certain real-world systems involving large numbers of decision makers (Rashedi et al., 2016; Meigs et al., 2020; Shavandi & Khedmati, 2022; Eck et al., 2023). The MFG-NE has therefore been used to find approximate solutions for a wide variety of real-world problems involving a large but finite number of agents, which might otherwise have been too difficult to solve, including:

- financial/energy markets, ticket pricing and the green economy (Trimborn et al., 2018; Tchuendom et al., 2024; Becherer & Hesse, 2024; Zhang & Ren, 2024; Chen et al., 2024a;e; Bernasconi et al., 2023; Cecchin et al., 2025; Bo et al., 2025; Fu & Horst, 2025; Wang et al., 2025b; Moll & Ryzhik, 2025; Tchuendom et al., 2025; Wang, 2025; Li et al., 2025a; Aydin et al., 2025; Grosset & Sartori, 2025; Aksamit et al., 2025; Feng & Liu, 2025; He & Liu, 2025);

- autonomous vehicles, traffic signal control, ride-hailing platforms and electric vehicle charging (Huang et al., 2020; Hu et al., 2023; Dey & Xu, 2023; Hedel & Nguyen, 2024; 2025; Mo et al., 2024; Pande et al., 2025; Niu et al., 2025; Chen et al., 2024c; Li et al., 2025c);

- cryptocurrency mining, edge computing, cloud resource management, smart grids, and other large-scale cyber-physical systems (Bauso & Tembine, 2016; Benamor et al., 2022; Mao et al., 2022; Mishra et al., 2023; Gao et al., 2023; Wang et al., 2024a; Wu et al., 2024b; Xu et al., 2024b; Aggarwal et al., 2024; Shen et al., 2024; Li et al., 2024b; Miao et al., 2024; Aggarwal et al., 2025; Kang et al., 2025a; Yang et al., 2025; Garcia et al., 2025);

- swarms, defence, communication networks and data collection by UAVs (Wang et al., 2020b; 2024c; Le Ménec, 2024; Lei et al., 2024; Emami et al., 2024; Zhou et al., 2024; You et al., 2024; Kang et al., 2025c; Choutri et al., 2025; Xu et al., 2025c; Bai et al., 2025);

- social network modelling, crowd modelling, crowdsensing (Yang et al., 2023; Kang et al., 2025b; Glukhov et al., 2025);

- pollution regulation, resource management in fisheries and political governance (Del Sarto et al., 2024; Yoshioka et al., 2024; Dayanikli & Lauriere, 2025; Chu et al., 2025).

For such large, complex many-agent systems in the real world, it may be infeasible to find MFG-NEs analytically or via oracles/simulations of an infinite population (as they have been traditionally), such that learning must instead be conducted directly by the original finite population in its deployed environment. In such settings, in contrast to many previous methods, desirable qualities for MFG algorithms include: learning from the empirical distribution of $N$ agents (i.e. this distribution is generated only by the policies of the agents, rather than being updated by the algorithm itself or an external oracle/simulator); learning online from a single, non-episodic system run (also referred to in other works as a single sample path/trajectory (Zaman et al., 2023; Yardim et al., 2023)) - i.e. similar to the above, the population is not arbitrarily reset by an external controller; model-free learning; decentralisation; fast practical convergence (Huang & Lai, 2025); and robustness to unexpected failures of decentralised learners or changes in population size (Korecki et al., 2023).

Conversely, works on MFGs have traditionally been largely theoretical (Huang et al., 2006; Lasry & Lions, 2007) (often works do not present any empirical results (Yardim et al., 2023; Li et al., 2025b; Huang & Warnett, 2025; Ferreira et al., 2025; Lascu & Majka, 2025)), and methods for finding equilibria have often relied on assumptions that are too strong for real-world applications. The MFG-NE is classically found by solving a coupled system of dynamical equations: a forward evolution equation for the mean-field distribution, and a backwards equation for the representative agent's optimal response to the mean field, as in Def. 3.5 below[1]; crucially, these methods generally relied on the assumption of an infinite population (Laurière et al.,

---

[1]See, for example, Yoshioka et al. (2024); Wang et al. (2024b); Li et al. (2024a); Zhou et al. (2024); Chen et al. (2024b); Ren et al. (2024); Si & Shi (2024); Federico et al. (2024); Lee et al. (2024); Yang et al. (2025); Bai et al. (2025); Cecchin et al. (2025); Sun & Trafalis (2025); Dayanikli & Lauriere (2025); Wang et al. (2025a); Yang & Zhang (2025); Ersland et al. (2025); Ghosh (2025b); Aydin et al. (2025); Yang & Song (2025); Pande et al. (2025); Cao & Laurière (2025); Osborne & Smears (2025);

2022a). Early work solved the coupled equations using numerical methods that did not scale well for more complex state and action spaces (Achdou & Capuzzo-Dolcetta, 2010; Carlini & Silva, 2014; Briceño-Arias et al., 2018; Achdou et al., 2020); or, even if they could handle higher-dimensional problems, the methods were based on known models of the environment's dynamics (i.e. they were model-based) (Guo et al., 2019a; Fouque & Zhang, 2020; Cao et al., 2020; Carmona & Laurière, 2021; Germain et al., 2022; Anahtarci et al., 2023; Huang et al., 2024a;b; Barreiro-Gomez & Park, 2025), and/or computed a best-response to the mean-field distribution (Huang et al., 2006; Guo et al., 2019a; Elie et al., 2020; Perrin et al., 2020; 2021; Laurière et al., 2022a;b; Algumaei et al., 2023). The latter approach is both computationally inefficient in non-trivial settings (Laurière et al., 2022a; Yardim et al., 2023), and in many cases is not convergent (as in general it does not induce a contractive operator) (Cui & Koeppl, 2021; Laurière et al., 2022b). Subsequent work, including our own, has therefore moved towards model-free and/or policy-improvement scenarios (Subramanian & Mahajan, 2019; Mishra et al., 2020; Cacace, Simone et al., 2021; Perolat et al., 2021; Lee et al., 2021; Laurière et al., 2022a; Angiuli et al., 2022; Mishra et al., 2023; Guo et al., 2023), possibly with learning taking place by observing *N*-agent *empirical* population distributions (Yongacoglu et al., 2024; Yardim et al., 2023; Hu & Zhang, 2024).

Most prior works, including algorithms designed to solve MFGs using an *N*-agent empirical distribution, have also assumed an oracle that can generate samples of the game dynamics (for any distribution) to be provided to the learning agent (Anahtarci et al., 2019; Fu et al., 2019; Guo et al., 2019a; 2023; Anahtarci et al., 2023), or otherwise that the algorithm (rather than agents' policies) has direct control over the population distribution at each time step (Zhang et al., 2024; Chen et al., 2024d; 2023), such as cases where the agents' policies and distribution are updated on different timescales (Angiuli et al., 2023; Zeng et al., 2024), with the 'fictitious play' method being particularly popular (Tembine et al., 2012; Cardaliaguet, Pierre & Hadikhanloo, Saeed, 2017; Mguni et al., 2018; Subramanian & Mahajan, 2019; Perrin et al., 2020; 2021; Xie et al., 2021; Geist et al., 2021; Frédéric Bonnans et al., 2021; Laurière, 2021; Angiuli et al., 2022; Mao et al., 2022; Laurière et al., 2022b; Zaman et al., 2023; Cui et al., 2024; Yu et al., 2024b). In practice, many-agent problems may not admit such arbitrary generation or manipulation (for example, in the context of robotics or controlling vehicle traffic), and so a desirable quality of learning algorithms is that they update only the agents' policies, rather than being able to arbitrarily reset their states. Learning may thus also need to leverage continuing, rather than episodic, tasks (Sutton & Barto, 2018). Yardim et al. (2023), Yongacoglu et al. (2024) and our own work therefore present algorithms that seek the MFG-NE using only a single run of the empirical population.

Almost all prior work relies on a centralised node to learn on behalf of all the agents. In this context 'centralised' does not necessarily imply global observability of the whole population's actions - which would generally make computation infeasible given the complexity of the problem - but rather that learning is only conducted from the samples of a single representative agent, whose policy updates are assumed to be automatically pushed to the rest of the population by the central node (Guo et al., 2019b; Xie et al., 2021; Laurière et al., 2022a; Anahtarci et al., 2023; Zaman et al., 2023; Inoue et al., 2023; Yardim et al., 2023; Jeloka et al., 2025; Yang & Song, 2025). However, outside of MFGs, the multi-agent systems community has recognised that the existence of a central coordinator is a very strong assumption even without global observability, and one that can both restrict scalability by constituting a bottleneck for computation and communication, and reveal a single point of failure for the whole system (Wai et al., 2018; Zhang et al., 2018; 2021a;b; Chen et al., 2021; Jiang et al., 2024; Xu et al., 2025a; Agyeman et al., 2025; Horyna et al., 2025). For example, if the single server coordinating all of a smart city's autonomous vehicles were to crash, the entire road network would cease to operate. As an alternative, some work has explored MFG algorithms for independent learning with *N* agents (Parise et al., 2015; Grammatico et al., 2015a;b; 2016; Mguni et al., 2018; Yongacoglu et al., 2024; 2022; Yardim et al., 2023; Li et al., 2024a; He & Liu, 2025). However, those works generally focus on existence proofs for equilibria or theoretical sample guarantees, instead of practical convergence speed, and have largely not considered robustness in the senses we address, despite fault-tolerance being an original motivation behind many-agent systems.

Martinez-Garcia et al. (2025); Opper & Reich (2025); Carlini & Coscetti (2025); Plank & Zhang (2025); Hua & Luo (2025); Moll & Ryzhik (2025); Ferreira et al. (2025); Chen et al. (2025); Tchuendom et al. (2025); Dey & Xu (2025); Fedorov (2025); Hedel & Nguyen (2025); Wang (2025); Li et al. (2026); Xiang & Shi (2025); Li et al. (2025a); Ghosh (2025a); Xu et al. (2025b); Si & Shi (2025a).

We address *all* of the desiderata discussed above by novelly introducing a communication network to the MFG setting. Communication networks have had success in other multi-agent settings, removing the reliance on inflexible, centralised structures (Zhang et al., 2021a;b; Chen et al., 2021).[2] We focus on 'coordination games', where agents can increase their individual rewards by following the same strategy as others and therefore have an incentive to communicate policies, even if the MFG setting itself is technically non-cooperative. Thus our work can be applied to real-world problems in e.g. traffic signal control, formation control in swarm robotics, and consensus and synchronisation e.g. for sensor networks (Soleimani et al., 2024).

We prove that our networked algorithm's theoretical sample guarantees lie between those of earlier centralised and independent algorithms. As in previous works, 'centralised' continues to mean that the updates of a representative agent are pushed from the central learner to the whole population, without implying global observability of the whole population's actions. While 'centralised learning' is the term used in prior works, we sometimes refer to 'central-agent learning' to reduce confusion. Next, to compare the architectures experimentally, we extend all three theoretical algorithms with experience replay buffers, without which we found them unable to learn in practical time. We show empirically that when the agents' Q-functions can be only roughly estimated due to fewer samples/updates, possibly leading to high variance in policy updates, then using the communication network to propagate better-performing policies through the population leads to faster learning than that achieved by agents learning entirely independently, which still hardly appear to learn at all. This is crucial in large complex environments that may be encountered in real applications, where the idealised hyperparameter choices (such as learning rates and numbers of iterations) required in previous works for theoretical convergence guarantees will be infeasible in practice. As well as demonstrating the empirical benefits of our scheme for learning speed, we conduct additional studies showing the advantages of communication for system robustness. In summary, our contributions include the following:

- We prove that a theoretical version of our networked algorithm (Alg. 1) has sample guarantees bounded between those of centralised (i.e. learning from a representative agent) and independent algorithms for learning with a non-episodic run of the empirical system. We provide the order of the difference in these bounds in terms of network structure and number of communication rounds, and contribute a policy-update stability guarantee (Sec. 5).

- All three theoretical algorithms do not permit any learning in practical time. We modify all three (Alg. 2, Sec. 6) to make their practical convergence feasible by including an experience replay buffer, allowing us to contribute the first empirical demonstrations of all three algorithms. An ablation study of the replay buffer is given in Sec. 7.4.5 - agents do not seem to learn at all without it.

- Our experiments demonstrate that in practical settings our communication scheme can markedly benefit learning speed over the independent case, sometimes performing similarly to the centralised case while removing the restrictive assumption of the latter. We also show that via our practical modifications we can learn without enforcing several of the algorithms' other theoretical assumptions (a goal shared by other works on practical MFG algorithms (Cui et al., 2024)) (Sec. 7.4).

- We provide ablations and additional empirical studies showing that our decentralised communication architecture brings further benefits over both the central-agent and independent alternatives in terms of robustness to unexpected update failures and changes in population size. For further discussion of the relevance of these scenarios in large multi-agent systems, see Sec. 7.4.2.

The paper structure is as follows: we give further related work in Sec. 2 and preliminaries in Sec. 3. We present our theoretical algorithms in Sec. 4 and theoretical results in Sec. 5. We give enhancements to the

---

[2]We preempt objections that communication with neighbours might violate the anonymity that is characteristic of the mean-field paradigm, by emphasising that the communication in our algorithm takes place outside of the ongoing learning-and-updating parts of each iteration. Thus the core learning assumptions of the mean-field paradigm are unaffected, as they essentially apply at a different level of abstraction (a convenient approximation) to the reality we face of $N$ agents that interact within the same environment. Indeed, prior works have combined networks with mean-field theory in different ways, such as using a mean field to describe adaptive dynamical networks (Berner et al., 2023).

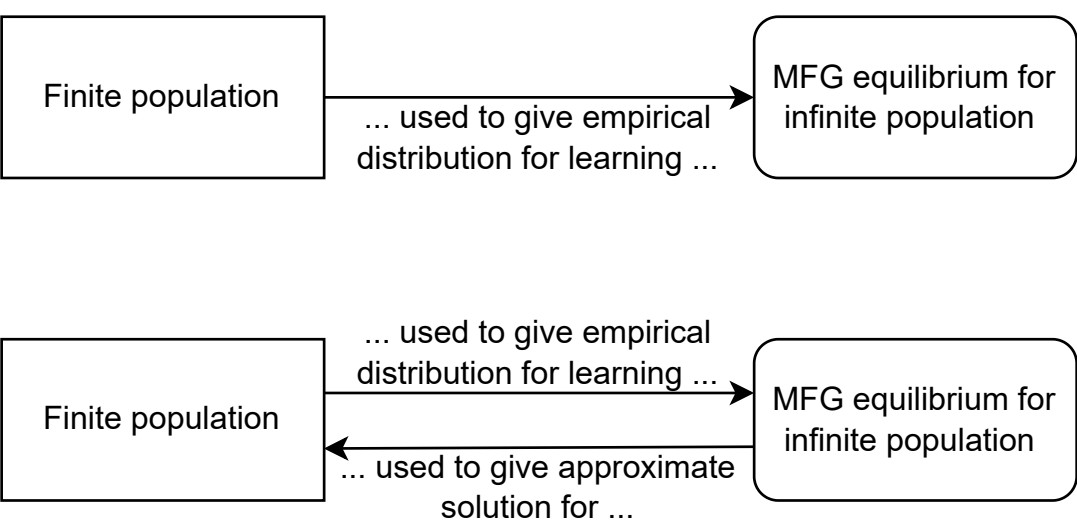

Figure 1: Two possible ways to conceive of our work regarding the relationship between the infinite- and finite-population games. Note that using the finite empirical population to try to learn a single MFG-NE policy $\boldsymbol{\pi} = (\pi^*, \ldots, \pi^*)$ that is to be followed by the whole infinite population (Def. 3.5) is *not* the same as directly finding $\boldsymbol{\pi}^* = (\pi^1, \ldots, \pi^N)$, i.e. the tuple of *individual* policies that gives the finite-population NE in Def. 3.3, a problem known to be hard (Daskalakis et al., 2006; Vinyals et al., 2019; Mcaleer et al., 2020; Shavandi & Khedmati, 2022; Li et al.; Yardim & He, 2024).

algorithms necessary for learning in practical time in Sec. 6, and provide experiments and discussion in Sec. 7. We conclude by discussing limitations and suggestions for future work in Sec. 8.

**Remark 1.1.** Solving the theoretical MFG problem involves finding the single policy that, when given to all agents in the infinite population, best responds to the resulting mean-field distribution. We preempt objections to our use of a finite population for learning in a mean-field context by giving two ways to conceive of our work (illustrated in Fig. 1), which mirror and make more explicit the similar motivations underpinning many other MFG works (Cui et al., 2023b; Dayanikli et al., 2024; Zaman et al., 2024; Bayraktar & Kara, 2024; Yongacoglu et al., 2024; Yang et al., 2025; Jeloka et al., 2025; Cecchin et al., 2025; Bo et al., 2025; Tchuendom et al., 2025; Magnino et al., 2025; Graber, 2025; Aggarwal et al., 2025; Yardim et al., 2025; Höfer et al., 2025; Feng & Liu, 2025; Si & Shi, 2025a; Yang et al., 2025). Firstly, we contribute algorithms that allow the solution to a MFG problem to be learnt using the empirical distribution of a decentralised finite population, without needing to make unrealistic assumptions about access to an oracle for the infinite population. Note that it is impractical to assume that the decentralised agents always follow a single identical policy throughout training, a logic also followed by earlier works (Yardim et al., 2023).

Alternatively, we may have originally been interested in finding a NE for a large, finite population, but, due to the scalability issues of learning approaches like MARL, forced to turn to the MFG framework to find a policy that gives an approximate solution to the finite-population problem. We contribute algorithms that allow the deployed finite population to find the MFG solution that in turn approximately solves the original problem, without unrealistic assumptions about centralised training. Under this framing, it may matter less whether all agents follow a single policy throughout training.

## 2 Related work

In our introduction in Sec. 1 we place our work in the general context of the MFG algorithms that preceded it. We now discuss research specifically relevant to our own work in more detail.

Naturally, decentralised communication is most applicable in settings where learning takes place along a continuing system run, rather than the distribution being manipulated by an oracle or arbitrarily reset for new episodes, since these imply a level of external control over the population that results in centralised learning. Equally, it is in situations of learning from finite numbers of real, deployed agents (rather than settings able to simulate infinite populations) that we are most likely to be concerned with fault tolerance. As such, our work is most closely related to Yardim et al. (2023) and Yongacoglu et al. (2024), which provide algorithms for centralised and independent learning with empirical distributions along non-episodic system runs: we contribute a networked learning algorithm in this setting. Yongacoglu et al. (2024) empirically demonstrates an independent learning algorithm when agents observe compressed information about the mean-field distribution as well as their local state, but they do not compare this to any other algorithms or baselines. Yardim et al. (2023) compares algorithms for centralised and independent learning theoretically, but does not provide empirical demonstrations. In contrast, in addition to providing theoretical guarantees, we empirically demonstrate our networked learning algorithm, where agents observe only their local state, in comparison to both centralised and independent baselines, as well as concerning ourselves with the speed of practical convergence and robustness, unlike these works.

More generally, a number of works refer to 'decentralisation' in MFGs, but often in a different sense to our understanding of it. In particular, many works that say they consider decentralisation actually learn/derive policies via a centralised method (often involving a representative player), and simply mean that agents' policies are *executed* independently based on local information, which we take as a given across our learning architectures (Wang, 2025; Choutri et al., 2025; Xiang & Shi, 2025; Feng & Liu, 2025; Si & Shi, 2025a;b). He & Liu (2025) use reinforcement learning (RL) to solve a two-level mean-field problem, where there is a MFG between 'aggregators', each of which is solving a local mean-field control (MFC) problem (the cooperative alternative to a MFG). They solve the MFG via decentralised learning by the $N$ aggregators, but each aggregator solves its MFC problem in a centralised manner via the assumption of a single agent that is representative of the heterogenous population. Moreover, they prove the existence of and convergence to a unique equilibrium, but do not provide sample guarantees or a convergence rate. Other works involve decentralisation in learning but under different MFG settings to our own: Li et al. (2025a); Ghosh (2025a); Xu et al. (2025b) derive controls in a decentralised way, but rely on a model of the environment, while Yardim et al. (2025) uses independent learning but not via RL, as they focus on repeated play of static, stateless games.

Improving the training speed and sample efficiency of (deep) (multi-agent) RL is gaining increasing attention (Wiggins et al., 2023; Yu et al., 2024a; Wu et al., 2024a; Patel et al., 2024), though our own work is one of the only on MFGs to be concerned with this. Huang & Lai (2024) trains on a distribution of MFG configurations to speed up inference on unseen problems, but does not learn online in a decentralised manner as in our own work. Similarly, while some attention has been given to the robustness of multi-agent systems to changes in population size, where it is sometimes referred to as 'ad-hoc teaming', 'open-agent systems', 'scalability' or 'generalisation' (Eck et al., 2023), it has more commonly been addressed in MARL (Dawood et al., 2023; Gao et al., 2024) than in MFGs (Wu et al., 2024c). Wu et al. (2024c) presents an MFG approach that allows new agents to join the population during *execution*, but training itself takes place offline in a centralised, episodic manner. Our networked communication framework, on the other hand, allows decentralised agents to join the population during online learning and to have minimal impact on the learning process by adopting policies from existing members of the population through communication (Sec. 7.4.2).

An existing area of work called 'robust mean-field games' studies the robustness of these games to uncertainty in the transition and reward functions (Bauso et al., 2012; 2016; Bauso & Tembine, 2016; Moon & Başar, 2017; Huang & Huang, 2017; Yang et al., 2018a; Tirumalai & Baras, 2022; Aydın & Saldi, 2023), but does not consider resilience to agent update failures, despite fault tolerance being one of the original motivations behind many-agent systems. On the other hand, we focus on robustness to failures and changes in the agent population itself.

We note a similarity between 1. our method for deciding which policies to propagate through the population (described in Sec. 6.2) and 2. the computation of evaluation/fitness functions within evolutionary algorithms to indicate which solutions are desirable to keep in the population for the next generation (Eiben & Smith, 2015; Sissodia et al., 2025). Moreover, the research avenue broadly referred to as 'distributed embodied

evolution' involves swarms of agents independently running evolutionary algorithms while operating within a physical/simulated environment and communicating behaviour parameters to neighbours (Haasdijk et al., 2014; Trueba et al., 2015), and is therefore even more similar to our setting, where decentralised RL updates are computed locally and then shared with neighbours. In distributed embodied evolution, the computed fitness of solutions helps determine both which are preserved by agents during local updates, and also which are chosen for broadcast or adoption between neighbours (Hart et al., 2015; Fernández Pérez et al., 2018; Fernández Pérez & Sanchez, 2019). Indeed, some works on distributed embodied evolution specifically consider features or rewards relating to the joint behaviour of the whole population (Gomes & Christensen, 2013; Prieto et al., 2016), similar to MFGs. The adjacent research area of cultural/language evolution for swarm robotics (Cambier et al., 2018; 2020; 2021) has similarly demonstrated the combination of evolutionary approaches and multi-agent communication networks for self-organised behaviours in swarms. However, unlike our own work, none of these areas employ reinforcement learning in the update of policies or the computation of the fitness functions.

Our work also shares parallels with 'population-based training' (Jaderberg et al., 2017), an approach that is likewise related to evolutionary algorithms. Population-based training involves optimising neural networks by performance-based transfer of parameters and hyperparameters among a population of concurrent processes. Our algorithms are tabular rather than neural network-based, and we are also interested in the interactive behaviour of the population itself rather than simply using it for parallelising the optimisation.

## 3 Preliminaries

We use the following notation. $N$ is the number of agents in a population, with $\mathcal{S}$ and $\mathcal{A}$ representing the finite state and common action spaces, respectively. The sets $\mathcal{S}$ and $\mathcal{A}$ are equipped with the discrete metric $d(x,y) = \mathbb{1}_{x \neq y}$. The set of probability measures on a finite set $\mathcal{X}$ is denoted $\Delta_{\mathcal{X}}$, and $\mathbf{e}_x \in \Delta_{\mathcal{X}}$ for $x \in \mathcal{X}$ is a one-hot vector with only the entry corresponding to $x$ set to 1, and all others set to 0. For time $t \geq 0$, $\hat{\mu}_t = \frac{1}{N} \sum_{i=1}^{N} \sum_{s \in \mathcal{S}} \mathbb{1}_{s_t^i = s} \mathbf{e}_s \in \Delta_{\mathcal{S}}$ is a vector denoting the empirical state distribution of the $N$ agents at time $t$. The set of policies is $\Pi = \{\pi : \mathcal{S} \to \Delta_{\mathcal{A}}\}$, and the set of Q-functions is denoted $\mathcal{Q} = \{q : \mathcal{S} \times \mathcal{A} \to \mathbb{R}\}$. For $\pi, \pi' \in \Pi$ and $q, q' \in \mathcal{Q}$, we have the norms $||\pi - \pi'||_1 := \sup_{s \in \mathcal{S}} ||\pi(s) - \pi'(s)||_1$ and $||q - q'||_\infty := \sup_{s \in \mathcal{S}, a \in \mathcal{A}} |q(s,a) - q'(s,a)|$.

Function $h : \Delta_{\mathcal{A}} \to \mathbb{R}_{\geq 0}$ denotes a strongly concave function, which we implement in our experiments as the scaled entropy regulariser $\lambda h_{ent}(u) = -\lambda \sum_a u(a) \log u(a)$, for $a \in \mathcal{A}$, $u \in \Delta_{\mathcal{A}}$ and $\lambda > 0$. As in many earlier works (Cui & Koeppl, 2021; Guo et al., 2022; Anahtarci et al., 2023; Algumaei et al., 2023; Yu & Yuan, 2023; Yardim et al., 2023; 2025; Lu & Monmarché, 2025; Ferreira et al., 2025; Lascu & Majka, 2025; He & Liu, 2025), regularisation is theoretically required to ensure the contractivity of operators and continued exploration, and hence algorithmic convergence. However, it has been recognised that modifying the RL objective in this way can bias the NE (Laurière et al., 2022b; Su & Lu, 2022; Yardim et al., 2023; Hu & Zhang, 2024; Lu & Monmarché, 2025). We show in our experiments that we are able to reduce $\lambda$ to 0 with no detriment to convergence.

**Definition 3.1** (*N*-player symmetric anonymous games). An N-player stochastic game with symmetric, anonymous agents is given by the tuple $\langle N, \mathcal{S}, \mathcal{A}, P, R, \gamma \rangle$, where $\mathcal{A}$ is the action space, identical for each agent; $\mathcal{S}$ is the identical state space of each agent, such that their initial states are $\{s_0^i\}_{i=1}^N \in \mathcal{S}^N$ and their policies are $\{\pi^i\}_{i=1}^N \in \Pi^N$. $P : \mathcal{S} \times \mathcal{A} \times \Delta_{\mathcal{S}} \to \Delta_{\mathcal{S}}$ is the transition function and $R : \mathcal{S} \times \mathcal{A} \times \Delta_{\mathcal{S}} \to [0,1]$ is the reward function, which map each agent's local state and action and the population's empirical distribution to transition probabilities and bounded rewards, respectively, i.e. $\forall i \in \{1, \dots, N\}$

$$s_{t+1}^i \sim P(\cdot | s_t^i, a_t^i, \hat{\mu}_t) \quad and \quad r_t^i = R(s_t^i, a_t^i, \hat{\mu}_t).$$

The policy of an agent is given by $a_t^i \sim \pi^i(s_t^i)$, that is, each agent only observes its own state, and not the joint state or empirical distribution of the population.

**Definition 3.2** (*N*-player discounted regularised return). With joint policies $\boldsymbol{\pi} := (\pi^1, \dots, \pi^N) \in \Pi^N$, initial states sampled from a distribution $\upsilon_0 \in \Delta_{\mathcal{S}}$ and $\gamma \in [0,1)$ as a discount factor, the expected discounted

regularised returns of each agent $i$ in the symmetric anonymous game are given by, $\forall i, j \in \{1, \ldots, N\}$,

$$\Psi_h^i(\boldsymbol{\pi}, \upsilon_0) = \mathbb{E}\left[\sum_{t=0}^{\infty} \gamma^t (R(s_t^i, a_t^i, \hat{\mu}_t) + h(\pi^i(s_t^i)))\bigg|_{\substack{s_0^j \sim \upsilon_0 \\ a_t^j \sim \pi^j(s_t^j) \\ s_{t+1}^j \sim P(\cdot|s_t^j, a_t^j, \hat{\mu}_t)}}\right].$$

**Definition 3.3** ($\delta$-NE). Say $\delta > 0$ and $(\pi, \boldsymbol{\pi}^{-i}) := (\pi^1, \ldots \pi^{i-1}, \pi, \pi^{i+1}, \ldots, \pi^N) \in \Pi^N$. An initial distribution $\upsilon_0 \in \Delta_{\mathcal{S}}$ and an $N$-tuple of policies $\boldsymbol{\pi} := (\pi^1, \ldots, \pi^N) \in \Pi^N$ form a $\delta$-NE $(\boldsymbol{\pi}, \upsilon_0)$ if

$$\Psi_h^i(\boldsymbol{\pi}, \upsilon_0) \geq \max_{\pi \in \Pi} \Psi_h^i((\pi, \boldsymbol{\pi}^{-i}), \upsilon_0) - \delta \quad \forall i \in \{1, \ldots, N\}.$$

At the limit as $N \to \infty$, the population of infinitely many agents can be characterised as a limit distribution $\mu \in \Delta_{\mathcal{S}}$. We denote the expected discounted return of the representative agent in the infinite-agent game - termed a MFG - as $V$, rather than $\Psi$ as in the finite $N$-agent case.

**Definition 3.4** (Mean-field discounted regularised return). For a policy-population pair $(\pi, \mu) \in \Pi \times \Delta_{\mathcal{S}}$,

$$V_h(\pi, \mu) = \mathbb{E}\left[\sum_{t=0}^{\infty} \gamma^t (R(s_t, a_t, \mu) + h(\pi(s_t)))\bigg|_{\substack{s_0 \sim \mu \\ a_t \sim \pi(s_t) \\ s_{t+1} \sim P(\cdot|s_t, a_t, \mu)}}\right].$$

A stationary MFG is one that has a unique population distribution that is stable with respect to a given policy, and the agents' policies are not time- or population-dependent.

**Definition 3.5** (NE of stationary MFG). For a policy $\pi^* \in \Pi$ and a population distribution $\mu^* \in \Delta_{\mathcal{S}}$, the pair $(\pi^*, \mu^*)$ is a stationary MFG-NE if the following optimality and stability conditions hold:

$$\text{optimality:} \quad V_h(\pi^*, \mu^*) = \max_{\pi} V_h(\pi, \mu^*),$$
$$\text{stability:} \quad \mu^*(s) = \sum_{s', a'} \mu^*(s')\pi^*(a'|s')P(s|s', a', \mu^*).$$

If the optimality condition is only satisfied with $V_h(\pi_\delta^*, \mu_\delta^*) \geq \max_\pi V_h(\pi, \mu_\delta^*) - \delta$, then $(\pi_\delta^*, \mu_\delta^*)$ is a $\delta$-NE of the MFG, where $\mu_\delta^*$ is obtained from the stability equation and $\pi_\delta^*$.

The MFG-NE is an approximate NE of the finite $N$-player game, in which we may have originally been interested but which is difficult to solve in itself (Laurière et al., 2022b; Yardim et al., 2023):

**Proposition 3.6** ($N$-player NE and MFG-NE (Thm. 1, (Anahtarci et al., 2023))). *If $(\pi^*, \mu^*)$ is a MFG-NE, then, under certain Lipschitz conditions (Anahtarci et al., 2023), for any $\delta > 0$, there exists $N(\delta) \in \mathbb{N}_{>0}$ such that, for all $N \geq N(\delta)$, the joint policy $\boldsymbol{\pi} = \{\pi^*, \pi^*, \ldots, \pi^*\} \in \Pi^N$ is a $\delta$-NE of the $N$-player game.*

**Remark 3.7.** We can show that $\delta$ can be characterised further in terms of $N$, with $(\pi^*, \mu^*)$ being an $\mathcal{O}(\frac{1}{\sqrt{N}})$-NE of the $N$-player symmetric anonymous game (Yardim et al., 2023; Chen et al., 2024e; Yardim et al., 2025).

For our new, networked learning algorithm, we also introduce the concept of a time-varying communication network, where the links between agents that make up the network may change at each time step $t$. Most commonly we might think of such a network as depending on the spatial locations of decentralised agents, such as physical robots, which can communicate with neighbours that fall within a given broadcast radius. When the agents move in the environment, their neighbours and therefore communication links may change. However, the dynamic network can also depend on other factors that may or may not depend on each agent's state $s_t^i$. For example, even a network of fixed-location agents can change depending on which agents are active and broadcasting at a given time $t$, or if their broadcast radius changes, perhaps in relation to signal or battery strength.

**Definition 3.8** (Time-varying communication network). The time-varying communication network $\{\mathcal{G}_t\}_{t \geq 0}$ is given by $\mathcal{G}_t = (\mathcal{N}, \mathcal{E}_t)$, where $\mathcal{N}$ is the set of vertices each representing an agent $i \in \{1, \ldots, N\}$, and the edge set $\mathcal{E}_t \subseteq \{(i,j) : i, j \in \mathcal{N}, i \neq j\}$ is the set of undirected communication links by which information can be shared at time $t$.

A network is *connected* if there is a sequence of distinct edges forming a path between each distinct pair of vertices. The *union* of a collection of graphs $\{\mathcal{G}_t, \mathcal{G}_{t+1}, \cdots, \mathcal{G}_{t+\omega}\}$ ($\omega \in \mathbb{N}$) is the graph with vertices and edge set equalling the union of the vertices and edge sets of the graphs in the collection (Jadbabaie et al., 2003). A collection is *jointly connected* if its members' union is connected. A network's *diameter $d_{\mathcal{G}}$* is the maximum of the shortest path length between any pair of nodes.

### 3.1 Further technical conditions for algorithms and theorems

Our theoretical results, which compare our networked algorithm with the centralised and independent alternatives from Yardim et al. (2023), rely on several further definitions from their work and assumptions from their theorems. We introduce these here as some values are referenced when describing our algorithm in Sec. 4, in advance of the theoretical analysis in Sec. 5.

**Assumption 3.9** (Lipschitz continuity of $P$ and $R$)**.** There exist constants $K_\mu, K_s, K_a, L_\mu, L_s, L_a \in \mathbb{R}_{\geq 0}$ such that $\forall s, s' \in \mathcal{S}, \forall a, a' \in \mathcal{A}, \forall \mu, \mu' \in \Delta_{\mathcal{S}}$,

$$||P(\cdot|s, a, \mu) - P(\cdot|s', a', \mu')||_1 \leq K_\mu ||\mu - \mu'||_1 + K_s d(s, s') + K_a d(a, a'),$$

$$|R(s, a, \mu) - R(s', a', \mu')| \leq L_\mu ||\mu - \mu'||_1 + L_s d(s, s') + L_a d(a, a').$$

**Definition 3.10** (Population update operator)**.** The single-step population update operator $\Gamma_{pop} : \Delta_{\mathcal{S}} \times \Pi \to \Delta_{\mathcal{S}}$ is defined as, $\forall s \in \mathcal{S}$:

$$\Gamma_{pop}(\mu, \pi)(s) := \sum_{s' \in \mathcal{S}} \sum_{a' \in \mathcal{A}} \mu(s')\pi(a'|s')P(s|s', a', \mu).$$

We will use the short hand notation $\Gamma_{pop}^n(\mu, \pi) := \underbrace{\Gamma_{pop}(\ldots \Gamma_{pop}(\Gamma_{pop}(\mu, \pi), \pi), \ldots, \pi)}_{n \text{ times}}$.

We recall that $\Gamma_{pop}$ is known to be Lipschitz:

**Lemma 3.11** (Lipschitz population updates)**.** $\Gamma_{pop}$ *is Lipschitz with*

$$||\Gamma_{pop}(\mu, \pi) - \Gamma_{pop}(\mu', \pi')||_1 \leq L_{pop,\mu} ||\mu - \mu'||_1 + \frac{K_a}{2}||\pi - \pi'||_1,$$

*where* $L_{pop,\mu} := \left(\frac{K_s}{2} + \frac{K_a}{2} + K_\mu\right)$, $\forall \pi \in \Pi, \mu \in \Delta_{\mathcal{S}}$.

For stationary MFGs the population distribution must be stable with respect to a policy, requiring that $\Gamma_{pop}(\cdot, \pi)$ is contractive $\forall \pi \in \Pi$:

**Assumption 3.12** (Stable population)**.** Population updates are stable, i.e. $L_{pop,\mu} < 1$.

**Definition 3.13** (Stable population operator $\Gamma_{pop}^\infty$)**.** Given Assumption 3.12, the operator $\Gamma_{pop}^\infty : \Pi \to \Delta_{\mathcal{S}}$ maps a given policy to its unique stable population distribution such that $\Gamma_{pop}(\Gamma_{pop}^\infty(\pi), \pi) = \Gamma_{pop}^\infty(\pi)$, i.e. the unique fixed point of $\Gamma_{pop}(\cdot, \pi) : \Delta_{\mathcal{S}} \to \Delta_{\mathcal{S}}$.

**Definition 3.14** ($Q_h$ and $q_h$ functions)**.** We define, for any pair $(s, a) \in \mathcal{S} \times \mathcal{A}$:

$$Q_h(s, a|\pi, \mu) := \mathbb{E}\left[\sum_{t=0}^\infty \gamma^t(R(s_t, a_t, \mu) + h(\pi(s_t))) \Big|_{a_0=a,}^{s_0=s,} {}_{a_{t+1}\sim\pi(\cdot|s_{t+1})}^{s_{t+1}\sim P(\cdot|s_t,a_t,\mu),}, \forall t \geq 0\right]$$

and

$$q_h(s, a|\pi, \mu) := R(s, a, \mu) + \gamma \sum_{s', a'} P(s'|s, a, \mu)\pi(a'|s')Q_h(s', a'|\pi, \mu).$$

**Definition 3.15** ($\Gamma_q$ operator)**.** The operator $\Gamma_q : \Pi \times \Delta_{\mathcal{S}} \to \mathcal{Q}$, which maps population-policy pairs to Q-functions, is defined as $\Gamma_q(\pi, \mu) := q_h(\cdot, \cdot|\pi, \mu) \in \mathcal{Q} \ \forall \pi \in \Pi, \mu \in \Delta_{\mathcal{S}}$.

We define, for $h_{\max} > 0$ and $h : \Delta_{\mathcal{A}} \to [0, h_{\max}]$, $u_{\max} \in \Delta_{\mathcal{A}}$ such that $h(u_{\max}) = h_{\max}$. We further define $Q_{\max} := \frac{1+h_{\max}}{1-\gamma}$, and set $\pi_{\max} \in \Pi$ such that $\pi_{\max}(s) = u_{\max}, \forall s \in \mathcal{S}$. For any $\Delta h \in \mathbb{R}_{>0}$, we also define the convex set $\mathcal{U}_{\Delta h} := \{u \in \Delta_{\mathcal{A}} : h(u) \geq h_{\max} - \Delta h\}$. We assume that the regulariser $h$ ensures that all actions at all states are explored with non-zero probability:

**Assumption 3.16** (Persistence of excitation)**.** We assume there exists $p_{inf} > 0$ such that:

1. $\pi_{\max}(a|s) \geq p_{inf} \ \forall s \in \mathcal{S}, a \in \mathcal{A}$,

2. For any $\pi \in \Pi$ and $q \in \mathcal{Q}$ that satisfy, $\forall (s, a) \in \mathcal{S} \times \mathcal{A}$, $\pi(a|s) \geq p_{inf}$ and $0 \leq q(s, a) \leq Q_{\max}$, it holds that $\Gamma_\eta^{md}(q, \pi)(a|s) \geq p_{inf}, \forall (s, a) \in \mathcal{S} \times \mathcal{A}$.

**Assumption 3.17** (Sufficient mixing)**.** For any $\pi \in \Pi$ satisfying $\pi(a|s) \geq p_{inf} > 0 \ \forall s \in \mathcal{S}, a \in \mathcal{A}$, and any initial states $\{s_0^i\}_i \in \mathcal{S}^N$, there exist $T_{mix} > 0, \delta_{mix} > 0$ such that $\mathbb{P}(s_{T_{mix}}^j = s'|\{s_0^i\}_i) \geq \delta_{mix}$, $\forall s' \in \mathcal{S}, j \in [N]$.

**Definition 3.18** (Nested learning operator)**.** For a learning rate $\eta > 0$, $\Gamma_\eta : \Pi \to \Pi$ is defined as

$$\Gamma_\eta(\pi) := \Gamma_\eta^{md}(\Gamma_q(\pi, \Gamma_{pop}^\infty(\pi)), \pi).$$

**Lemma 3.19** (Lipschitz continuity of $\Gamma_\eta$)**.** *For any $\eta > 0$, the operator $\Gamma_\eta : \Pi \to \Pi$ is Lipschitz with constant $L_{\Gamma_\eta}$ on $(\Pi, ||\cdot||_1)$.*

## 4 Learning with networked, decentralised agents

**Roadmap** We first introduce theoretical versions of our operators and algorithm (Secs. 4.1, 4.2), in order to show that our networked framework has sample guarantees bounded between those of the centralised- and independent-learning cases (Sec. 5). We then show that our novel incorporation of an experience replay buffer (Sec. 6.1), along with networked communication, means that empirically we can remove many of the theoretical assumptions and practically infeasible hyperparameter choices that are required by the sample guarantees of the theoretical algorithms, in which cases we demonstrate experimentally that our networked algorithm can significantly outperform the independent algorithm, often performing similarly to the central-agent one (Sec. 7).

### 4.1 Learning with $N$ agents from a single run

We begin by outlining the basic procedure for solving the MFG using the $N$-agent empirical distribution and a single, non-episodic system run. The two underlying operators are the same for the centralised, independent and networked architectures; in the latter two cases all agents apply the operators individually, while in the centralised setting a single representative agent (the agent with arbitrary index $i = 1$) estimates the Q-function and computes an updated policy that is pushed to all the other agents.

Learning agents use the stochastic temporal difference (TD)-learning operator to repeatedly update an estimate of the Q-function of their current policy with respect to the current empirical distribution, i.e. to approximate the operator $\Gamma_q$ (Def. 3.15, Sec. 3.1):

**Definition 4.1** (Stochastic TD-learning operator, simplified from Def. 4.1 in Yardim et al. (2023))**.** We define $\mathcal{Z} := \mathcal{S} \times \mathcal{A} \times [0, 1] \times \mathcal{S} \times \mathcal{A}$, and say that $\zeta_t^i$ is the transition observed by agent $i$ at time $t$, given by $\zeta_t^i = (s_t^i, a_t^i, r_t^i, s_{t+1}^i, a_{t+1}^i)$. The TD-learning operator $\tilde{F}_\beta^\pi : \mathcal{Q} \times \mathcal{Z} \to \mathcal{Q}$ is defined, for any $Q \in \mathcal{Q}, \zeta_t \in \mathcal{Z}, \beta \in \mathbb{R}$, as

$$\tilde{F}_\beta^\pi(Q, \zeta_t) = Q(s_t, a_t) - \beta \Big( Q(s_t, a_t) - r_t - h(\pi(s_t)) - \gamma Q(s_{t+1}, a_{t+1}) \Big).$$

Having estimated the Q-function of their current policy, agents update this policy by selecting, for each state, a probability distribution over their actions that maximises the combination of three terms (Def. 4.2): 1. the value of the given state with respect to the estimated Q-function; 2. a regulariser over the action probability distribution (in practice, we maximise the scaled entropy of the distribution); 3. a metric of similarity between the new action probabilities for the given state and those of the previous policy, given by

---

**Algorithm 1** Networked learning with single system run

---

**Require:** loop parameters $K, M_{pg}, M_{td}, C$, learning parameters $\eta, \{\beta_m\}_{m\in\{0,\ldots,M_{pg}-1\}}$, $\lambda, \gamma$, $\{\tau_k\}_{k\in\{0,\ldots,K-1\}}$

**Require:** initial states $\{s_0^i\}_{i=1}^N$

1: Set $\pi_0^i = \pi_{\max}, \forall i$ and $t \leftarrow 0$
2: **for** $k = 0, \ldots, K-1$ **do**
3:    $\forall s, a, i : \hat{Q}_0^i(s,a) = Q_{\max}$
4:    **for** $m = 0, \ldots, M_{pg}-1$ **do**
5:       **for** $M_{td}$ iterations **do**
6:          Take step $\forall i : a_t^i \sim \pi_k^i(\cdot|s_t^i), r_t^i = R(s_t^i, a_t^i, \hat{\mu}_t), s_{t+1}^i \sim P(\cdot|s_t^i, a_t^i, \hat{\mu}_t); t \leftarrow t+1$
7:       **end for**
8:       Compute TD update $(\forall i)$: $\hat{Q}_{m+1}^i = \tilde{F}_{\beta_m}^{\pi_k^i}(\hat{Q}_m^i, \zeta_{t-2}^i)$ (Def. 4.1)
9:    **end for**
10:   PMA step $\forall i : \pi_{k+1}^i = \Gamma_\eta^{md}(\hat{Q}_{M_{pg}}^i, \pi_k^i)$ (Def. 4.2)
11:   $\forall i :$ Generate $\sigma_{k+1}^i$ associated with $\pi_{k+1}^i$
12:   **for** $C$ rounds **do**
13:      $\forall i :$ Broadcast $\sigma_{k+1}^i, \pi_{k+1}^i$
14:      $\forall i : J_t^i = i \cup \{j \in \mathcal{N} : (i,j) \in \mathcal{E}_t\}$
15:      $\forall i :$ Select adopted$^i \sim \Pr(\text{adopted}^i = j) = \frac{\exp(\sigma_{k+1}^j/\tau_k)}{\sum_{x\in J_t^i}\exp(\sigma_{k+1}^x/\tau_k)} \; \forall j \in J_t^i$
16:      $\forall i : \sigma_{k+1}^i \leftarrow \sigma_{k+1}^{\text{adopted}^i}, \pi_{k+1}^i \leftarrow \pi_{k+1}^{\text{adopted}^i}$
17:      Take step $\forall i : a_t^i \sim \pi_{k+1}^i(\cdot|s_t^i), r_t^i = R(s_t^i, a_t^i, \hat{\mu}_t), s_{t+1}^i \sim P(\cdot|s_t^i, a_t^i, \hat{\mu}_t); t \leftarrow t+1$
18:   **end for**
19: **end for**
20: **return** policies $\{\pi_K^i\}_{i=1}^N$

---

the squared two-norm of the difference between the two distributions. We can alter the importance of the similarity metric relative to the other two terms by varying a parameter $\eta$, which is equivalent to changing the learning rate of the policy update. The three terms in the maximisation function can be seen in the policy mirror ascent (PMA) operator:

**Definition 4.2** (Policy mirror ascent operator (Def. 3.5, (Yardim et al., 2023))). For a learning rate $\eta > 0$ and $L_h := L_a + \gamma\frac{L_s K_a}{2-\gamma K_s}$ (where these constants are defined in Assumption 3.9 in Sec. 3.1), the PMA update operator $\Gamma_\eta^{md} : \mathcal{Q} \times \Pi \to \Pi$ is defined as, $\forall s \in \mathcal{S}, \forall Q \in \mathcal{Q}, \forall \pi \in \Pi$

$$\Gamma_\eta^{md}(Q,\pi)(s) := \underset{u\in\mathcal{U}_{L_h}}{\arg\max}\left(\langle u, q(s,\cdot)\rangle + h(u) - \frac{1}{2\eta}||u-\pi(s)||_2^2\right).$$

The theoretical learning algorithm has three nested loops (see Lines 2, 4 and 5 of Alg. 1). The policy update is applied $K$ times. Before the policy update in each of the $K$ loops, agents update their estimate of the Q-function by applying the stochastic TD-learning operator $M_{pg}$ times. Prior to the TD update in each of the $M_{pg}$ loops, agents take $M_{td}$ steps in the environment without updating. The $M_{td}$ loops exist to create a delay between each TD update to reduce bias when using the empirical distribution to approximate the mean field in a non-episodic system run (Kotsalis et al., 2022). However, we find in our experiments that we are able to essentially remove the inner $M_{td}$ loops (Sec. 7.4).

## 4.2 Decentralised communication between agents

In our novel algorithm Alg. 1, agents compute policy updates in a decentralised way as in the independent case (Lines 3-10), before exchanging policies with neighbours in Lines 11-18 by the following method, which allows policies to spread through the population.[3] Coupled to their updated policy $\pi_{k+1}^i$, agents generate

---

[3]As discussed in Sec. 2, our communication method is reminiscent of the use of fitness functions in distributed evolutionary algorithms (Eiben & Smith, 2015; Hart et al., 2015).

a scalar value $\sigma_{k+1}^i$ (Line 11). The value provides information that helps agents decide between policies that they may wish to adopt from neighbours. Different methods for choosing between values received from neighbours, and for generating the values in the first place, lead to different policies spreading through the population. For example, generating or choosing $\sigma_{k+1}^i$ at random leads to policies being exchanged at random (required in Thm. 5.2), whereas generating $\sigma_{k+1}^i$ as an approximation of the return of $\pi_{k+1}^i$ and then selecting the highest received value of $\sigma_{k+1}^j$ leads to better performing policies spreading through the population. The latter is the approach we use for accelerating learning empirically (described in Sec. 6.2 on the practical running of our algorithm), albeit we use a softmax rather than a max function for selecting between received values. However, for generality in our theoretical results, we do not focus on a specific method for generating $\sigma_{k+1}^i$, such that it can be arbitrary for Thms. 5.2 and 5.10 below, and with few restrictions for Thms. 5.3 and 5.6.

Agents broadcast their policy $\pi_{k+1}^i$ and the associated $\sigma_{k+1}^i$ value to their neighbours (Line 13). Agents have a certain broadcast radius, defining the structure of the possibly time-varying communication network. Of the policies and associated values received by a given agent (including its own) (Line 14), the agent selects a $\sigma_{k+1}^j$ with a probability defined by a softmax function over the received values, and *adopts* the policy associated with this $\sigma_{k+1}^i$, i.e. it sets its own current $\pi_{k+1}^i$ and $\sigma_{k+1}^i$ to the ones it has selected (Lines 15, 16). This process repeats for $C$ communication rounds, before the Q-function estimation steps begin again. After each communication round, the agents take a step in the environment (Line 17), such that if the communication network is affected by the agents' states, then agents that are unconnected from any others in a given communication round might become connected in the next. (In our experiments we set $C$ as 1 to show the benefits to convergence speed brought by even a single communication round.) We assume the softmax function is subject to a possibly time-varying temperature parameter $\tau_k$. We discuss the effects of the values of $C$ and $\tau_k$, and the mechanism for generating $\sigma_{k+1}^i$, in subsequent sections.

**Remark 4.3.** Our networked architecture is effectively a generalisation of both the central-agent and independent settings (Algs. 2, 3, Yardim et al. (2023)). The independent setting is the special case where there is no communication, i.e. $C = 0$ - this serves as an implicit ablation of our communication scheme. The central-agent setting is the special case when $\sigma_{k+1}^i$ is generated from a unique ID for each agent, with the central learner agent assumed to generate the highest value by default. In this case we assume $\tau_k \to 0$ (such that the softmax becomes a max function), and that the communication network becomes jointly connected repeatedly, so the central learner's policy is always adopted by the entire population, assuming $C$ is large enough that the number of jointly connected collections of graphs occurring within $C$ is equal to the largest diameter of the union of any collection (Rajagopalan & Shah, 2010; Zhang et al., 2020).

**Remark 4.4.** In practice, when referring to a central-agent version of the networked Alg. 1, for simplicity we assume there is no networked communication and instead that the updated policy $\pi_{k+1}^1$ of the representative learner $i = 1$ is pushed to all agents after Line 10, as in Alg. 2 of (Yardim et al., 2023).

## 5    Theoretical results

We first give two theoretical results comparing the sample guarantees of our networked case with those of the other settings; the results respectively depend on whether the networked agents select which communicated policies to adopt at random or not. We then provide the order of the difference in these bounds in the non-random case in terms of the network structure and number of communication rounds. We finally give a policy-update stability guarantee, which applies in all scenarios.

**Lemma 5.1** (Independent learning, from Thm. 4.5, Yardim et al. (2023)). *For $p_{inf}$ and $\delta_{mix}$ defined in Assumptions 3.16 and 3.17 respectively, define $t_0 := \frac{16(1+\gamma)^2}{((1-\gamma)\delta_{mix}p_{inf})^2}$. Assume that Assumptions 3.9, 3.12, 3.16 and 3.17 hold, and that $\pi^*$ is the unique MFG-NE policy. For $L_{\Gamma_\eta}$ defined in Lem. 3.19, we assume $\eta > 0$ satisfies $L_{\Gamma_\eta} < 1$. The learning rates are $\beta_m = \frac{2}{(1-\gamma)(t_0+m-1)}$ $\forall m \geq 0$, and let $\varepsilon > 0$ be arbitrary. There exists a problem-dependent constant $a \in [0, \infty)$ such that if $K = \frac{\log 8\varepsilon^{-1}}{\log L_{\Gamma_\eta}^{-1}}$, $M_{pg} > \mathcal{O}(\varepsilon^{-2-a})$ and $M_{td} > \mathcal{O}(log^2\varepsilon^{-1})$, then the random output $\{\pi_K^i\}_i$ of Alg. 1 when run with $C = 0$ (such that there is no*

*communication) satisfies for all agents* $i \in \{1, \dots, N\}$,

$$\mathbb{E}\left[||\pi_K^i - \pi^*||_1\right] \leq \varepsilon + \mathcal{O}\left(\frac{1}{\sqrt{N}}\right).$$

We first give a result for the trivial situation of random adoption to provide an intuition that networked communication preserves the sample guarantees of independent learning, before showing the conditions under which the latter can be outperformed.

**Theorem 5.2** (Networked learning with random adoption). *For $p_{inf}$ and $\delta_{mix}$ defined in Assumptions 3.16 and 3.17 respectively, define $t_0 := \frac{16(1+\gamma)^2}{((1-\gamma)\delta_{mix}p_{inf})^2}$. Assume that Assumptions 3.9, 3.12, 3.16 and 3.17 hold, and that $\pi^*$ is the unique MFG-NE policy. For $L_{\Gamma_\eta}$ defined in Lem. 3.19, we assume $\eta > 0$ satisfies $L_{\Gamma_\eta} < 1$. The learning rates are $\beta_m = \frac{2}{(1-\gamma)(t_0+m-1)} \; \forall m \geq 0$, and let $\varepsilon > 0$ be arbitrary. Let us set $C > 0$ and $\tau_k \to \infty$. There exists a problem-dependent constant $a \in [0, \infty)$ such that if $K = \frac{\log 8\varepsilon^{-1}}{\log L_{\Gamma_\eta}^{-1}}$, $M_{pg} > \mathcal{O}(\varepsilon^{-2-a})$ and $M_{td} > \mathcal{O}(log^2\varepsilon^{-1})$, then the random output $\{\pi_K^i\}_i$ of Alg. 1 preserves the sample guarantees of the independent-learning case given in Lem. 5.1, i.e. the output satisfies, for all agents $i \in \{1, \dots, N\}$,*

$$\mathbb{E}\left[||\pi_K^i - \pi^*||_1\right] \leq \varepsilon + \mathcal{O}\left(\frac{1}{\sqrt{N}}\right).$$

*Proof.* If $\tau_k \to \infty$, the softmax function that defines the probability of a received policy being adopted in Line 15 of Alg. 1 gives a uniform distribution. Policies are thus exchanged at random between communicating agents for an arbitrary $C > 0$ rounds, which does not affect the random output of the algorithm, such that the random output satisfies the same expectation as if $C = 0$. □

If $\sigma_{k+1}^i$ is generated arbitrarily and uniquely for each $i$, then for $\tau_k \in \mathbb{R}_{>0}$ (such that the softmax function gives a non-uniform distribution and adoption of received policies is therefore non-random), the sample complexity of the networked algorithm is bounded between that of the centralised and independent algorithms:

**Theorem 5.3** (Networked learning with non-random adoption). *Assume that Assumptions 3.9, 3.12, 3.16 and 3.17 hold, and that Alg. 1 is run with learning rates and constants as defined in Thm. 5.2, except now let us set $\tau_k \in \mathbb{R}_{>0}$. Assume that $\sigma_{k+1}^i$ is generated uniquely for each $i$, in a manner independent of any metric related to $\pi_{k+1}^i$, e.g. $\sigma_{k+1}^i$ is random or related only to the index $i$ (so as not to bias the spread of any particular policy). Let the random output of this Algorithm be denoted as $\{\pi_K^{i,net}\}_i$. Also consider an independent-learning version of the algorithm (i.e. with the same parameters except $C = 0$) and denote its random output $\{\pi_K^{i,ind}\}_i$; and a central-agent version of the algorithm with the same parameters (see Rem. 4.4) and denote its random output as $\pi_K^{cent}$. Then for all agents $i \in \{1, \dots, N\}$, the random outputs $\{\pi_K^{i,net}\}_i$, $\{\pi_K^{i,ind}\}_i$ and $\pi_K^{cent}$ satisfy the following relations, where $ub_{net}$, $ub_{ind}$ and $ub_{cent}$ are respective upper bounds for each case:*

$$\mathbb{E}\left[||\pi_K^{cent} - \pi^*||_1\right] \leq ub_{cent}, \quad \mathbb{E}\left[||\pi_K^{i,net} - \pi^*||_1\right] \leq ub_{net}, \quad \mathbb{E}\left[||\pi_K^{i,ind} - \pi^*||_1\right] \leq ub_{ind},$$

$$where \quad ub_{cent} \leq ub_{net} \leq ub_{ind} = \varepsilon + \mathcal{O}\left(\frac{1}{\sqrt{N}}\right).$$

*Proof.* We build off the proof of our Lem. 5.1, given in Thm. D.9 of Yardim et al. (2023). There the sample guarantees of the independent case are worse than those of the centralised algorithm as a result of the divergence between the decentralised policies due to the stochasticity of the PMA updates. For an arbitrary policy $\bar{\pi}_k \in \Pi$, for all $k = 0, 1, \dots, K$ define the policy divergence as the random variable $\Delta_k := \sum_{i=1}^N ||\pi_k^i - \bar{\pi}_k||_1$. We can say that $\Delta_{k,cent} = 0 \; \forall k$ is the divergence in the central-agent case, while in the networked case the policy divergence is $\Delta_{k+1,c}$ after communication round $c \in 1, \dots, C$. The independent case is equivalent to the scenario when $C = 0$, such that its policy divergence can be written $\Delta_{k+1,0}$.

For $\tau_k \in \mathbb{R}_{>0}$, the adoption probability $\Pr\left(\text{adopted}^i = \sigma_{k+1}^j\right) = \frac{\exp\left(\sigma_{k+1}^j / \tau_k\right)}{\sum_{x=1}^{[J_t^i]} \exp\left(\sigma_{k+1}^x / \tau_k\right)}$ (as in Line 15 of Alg. 1) is higher for some $j \in J_t^i$ than for others. This means that for $c > 0$ for which there are communication links in the population, in expectation the number of unique policies in the population will decrease, as it will likely become that $\pi_{k+1}^i = \pi_{k+1}^j$ for some $i, j \in \{1, \ldots, N\}$. As such, $\Delta_{k+1,cent} \leq \mathbb{E}\left[\Delta_{k+1,C}\right] \leq \mathbb{E}\left[\Delta_{k+1,0}\right]$, i.e. the policy divergence in the independent-learning case is expected to be greater than or equal to that of the networked case.

The proof of Lem. 5.1 given in Thm. D.9 of Yardim et al. (2023) ends with, for constants $\chi$ and $\xi$,

$$\mathbb{E}\left[||\pi_K^i - \pi^*||_1\right] \leq 2L_{\Gamma_\eta}^K + \frac{\chi}{1 - L_{\Gamma_\eta}} + \xi \sum_{k=1}^{K-1} L_{\Gamma_\eta}^{K-k-1} \mathbb{E}\left[\Delta_k\right],$$

where in our context the policy divergence in the independent case $\mathbb{E}\left[\Delta_{k+1}\right]$ is equivalent to $\mathbb{E}\left[\Delta_{k+1,C}\right]$ when $C = 0$, i.e. $\mathbb{E}\left[\Delta_{k+1,0}\right]$.

Thus, for all agents $i \in \{1, \ldots, N\}$, the random outputs $\{\pi_K^{i,net}\}_i$, $\{\pi_K^{i,ind}\}_i$ and $\pi_K^{cent}$ satisfy:

$$\mathbb{E}\left[||\pi_K^{i,ind} - \pi^*||_1\right] \leq ub_{ind} = 2L_{\Gamma_\eta}^K + \frac{\chi}{1 - L_{\Gamma_\eta}} + \xi \sum_{k=1}^{K-1} L_{\Gamma_\eta}^{K-k-1} \mathbb{E}\left[\Delta_{k,0}\right],$$

$$\mathbb{E}\left[||\pi_K^{i,net} - \pi^*||_1\right] \leq ub_{net} = 2L_{\Gamma_\eta}^K + \frac{\chi}{1 - L_{\Gamma_\eta}} + \xi \sum_{k=1}^{K-1} L_{\Gamma_\eta}^{K-k-1} \mathbb{E}\left[\Delta_{k,C}\right],$$

$$\mathbb{E}\left[||\pi_K^{cent} - \pi^*||_1\right] \leq ub_{cent} = 2L_{\Gamma_\eta}^K + \frac{\chi}{1 - L_{\Gamma_\eta}} + \xi \sum_{k=1}^{K-1} L_{\Gamma_\eta}^{K-k-1} \mathbb{E}\left[\Delta_{k,cent}\right].$$

Since $\Delta_{k+1,cent} \leq \mathbb{E}\left[\Delta_{k+1,C}\right] \leq \mathbb{E}\left[\Delta_{k+1,0}\right]$, we obtain our result, i.e.

$$ub_{cent} \leq ub_{net} \leq ub_{ind} = \varepsilon + \mathcal{O}\left(\frac{1}{\sqrt{N}}\right).$$

$\square$

**Lemma 5.4** (Conditional TD learning from a single continuous run of the empirical distribution of $N$ agents, from Thm. 4.2, Yardim et al. (2023)). *Define* $t_0 := \frac{16(1+\gamma)^2}{((1-\gamma)\delta_{mix}p_{inf})^2}$. *Assume that Assumption 3.17 holds and let policies* $\{\pi^i\}_i$ *be given such that* $\pi^i(a|s) \geq p_{inf} \forall i$. *Assume Lines 3-9 of Alg. 1 are run with policies* $\{\pi^i\}_i$, *arbitrary initial agents states* $\{s_0^i\}_i$, *learning rates* $\beta_m = \frac{2}{(1-\gamma)(t_0+m-1)}$, $\forall m \geq 0$ *and* $M_{pg} > \mathcal{O}(\varepsilon^{-2})$, $M_{td} > \mathcal{O}(\log \varepsilon^{-1})$. *If* $\bar{\pi} \in \Pi$ *is an arbitrary policy,* $\Delta := \sum_{i=1}^N ||\pi^i - \bar{\pi}||_1$ *and* $Q^* := Q_h(\cdot, \cdot | \bar{\pi}, \mu_{\bar{\pi}})$, *then the random output* $\hat{Q}_{M_{pg}}^i$ *of Lines 3-9 satisfies*

$$\mathbb{E}\left[||\hat{Q}_{M_{pg}}^i - Q^*||_\infty\right] \leq \varepsilon + \mathcal{O}\left(\frac{1}{\sqrt{N}} + \frac{1}{N}\Delta + ||\pi^i - \bar{\pi}||_1\right).$$

**Remark 5.5.** It may help to see that our Thm. 5.3 is a consequence of the following. Denote $\hat{Q}_{M_{pg}}^{i,net}$, $\hat{Q}_{M_{pg}}^{i,ind}$ and $\hat{Q}_{M_{pg}}^{cent}$ as the random outputs of Lines 3-9 of Alg. 1 in the networked, independent and central-agent cases respectively. In Lem. 5.4, we can see that policy divergence gives bias terms in the estimation of the Q-value. Therefore, given $\Delta_{k+1,cent} \leq \mathbb{E}\left[\Delta_{k+1,C}\right] \leq \mathbb{E}\left[\Delta_{k+1,0}\right]$, we can also say

$$\mathbb{E}\left[||\hat{Q}_{M_{pg}}^{cent} - Q^*||_\infty\right] \leq \mathbb{E}\left[||\hat{Q}_{M_{pg}}^{i,net} - Q^*||_\infty\right] \leq \mathbb{E}\left[||\hat{Q}_{M_{pg}}^{i,ind} - Q^*||_\infty\right].$$

In other words, the networked case will require the same or fewer outer iterations $K$ to reduce the variance caused by this bias than the independent case requires (where the bias is non-vanishing), and the same or more iterations than the central-agent case requires.

**Theorem 5.6** (Relation between communication network structure and order of difference between the architectures' bounds). *In addition to the assumptions in Thm. 5.3, now also assume that the communication network $\mathcal{G}_t$ remains static and connected during the $C$ communication rounds. Assume also the diameter $d_{\mathcal{G}}$ of the network is equal for all $k$. Let us set $\tau_k \; \forall k$ as a small positive constant chosen to be sufficiently close to zero that the softmax essentially becomes a max function. Then, for the tight bound big Theta ($\Theta$), we can say that the difference in the upper bounds $ub_{net}$, $ub_{ind}$ and $ub_{cent}$ from Thm. 5.3 depends on $C$ and the network diameter $d_{\mathcal{G}}$ as follows (where the '$\approx$' relation comes from the approximate spread of policies through the network as explained in the proof):*

$$ub_{cent} + \Theta\left(f(C, d_{\mathcal{G}})\right) \quad \approx \quad ub_{net} \quad \approx \quad ub_{ind} - \Theta\left(1 - f(C, d_{\mathcal{G}})\right),$$

*for the piecewise function $f(C, d_{\mathcal{G}})$ defined as*

$$f(C, d_{\mathcal{G}}) = \begin{cases} \left(1 - \frac{1}{d_{\mathcal{G}}}\right)^C & \text{if } C < d_{\mathcal{G}}, \\ 0 & \text{if } C \geq d_{\mathcal{G}} \end{cases}.$$

*When $C \geq d_{\mathcal{G}}$, $ub_{net} = ub_{cent}$, so for $C > d_{\mathcal{G}}$ there is no additional improvement over the centralised bound. Equally when $C = 0$, we have exactly $ub_{net} = ub_{ind}$.*

*Proof.* From the proof of Thm. 5.3 we have:

$$\mathbb{E}\left[||\pi_K^{i,ind} - \pi^*||_1\right] \leq ub_{ind} = 2L_{\Gamma_\eta}^K + \frac{\chi}{1 - L_{\Gamma_\eta}} + \xi \sum_{k=1}^{K-1} L_{\Gamma_\eta}^{K-k-1} \mathbb{E}\left[\Delta_{k,0}\right],$$

$$\mathbb{E}\left[||\pi_K^{i,net} - \pi^*||_1\right] \leq ub_{net} = 2L_{\Gamma_\eta}^K + \frac{\chi}{1 - L_{\Gamma_\eta}} + \xi \sum_{k=1}^{K-1} L_{\Gamma_\eta}^{K-k-1} \mathbb{E}\left[\Delta_{k,C}\right],$$

$$\mathbb{E}\left[||\pi_K^{cent} - \pi^*||_1\right] \leq ub_{cent} = 2L_{\Gamma_\eta}^K + \frac{\chi}{1 - L_{\Gamma_\eta}} + \xi \sum_{k=1}^{K-1} L_{\Gamma_\eta}^{K-k-1} \mathbb{E}\left[\Delta_{k,cent}\right].$$

Say that $\sigma_{k+1}^{\max}$ is the highest $\sigma^i$ value in the population before the communication rounds at $k+1$. With a static, connected network and $\tau_k$ close to 0 for all $k$, max-consensus will always be reached on $\sigma_{k+1}^{\max}$ after $C = d_{\mathcal{G}}$ communication rounds, such that $\Delta_{k,cent} = \Delta_{k,d_{\mathcal{G}}} = 0$ (Nejad et al., 2009). The convergence rate of the max-consensus algorithm is $\frac{1}{d_{\mathcal{G}}}$ (Nejad et al., 2009), i.e. there is a decrease in the *number of policies in the population* by a factor of **approximately** $\frac{1}{d_{\mathcal{G}}}$ with each communication round up to $C = d_{\mathcal{G}}$, and therefore there is also a decrease in the *policy divergence* $\mathbb{E}\left[\Delta_{k,c}\right]$ by a factor of approximately $\frac{1}{d_{\mathcal{G}}}$ with each communication round. Thus

$$\mathbb{E}\left[\Delta_{k,c+1}\right] \approx \mathbb{E}\left[\Delta_{k,c}\right] - \left(\mathbb{E}\left[\Delta_{k,c}\right] \times \frac{1}{d_{\mathcal{G}}}\right), \text{ simplifying to}$$

$$\mathbb{E}\left[\Delta_{k,c+1}\right] \approx \mathbb{E}\left[\Delta_{k,c}\right] \times \left(1 - \frac{1}{d_{\mathcal{G}}}\right).$$

By induction

$$\mathbb{E}\left[\Delta_{k,C}\right] \approx \mathbb{E}\left[\Delta_{k,0}\right] \times \left(\left(1 - \frac{1}{d_{\mathcal{G}}}\right)^C\right),$$

however, we know that $\Delta_{k,d_{\mathcal{G}}} = 0$, so we can more accurately use the piecewise function $f(C, d_{\mathcal{G}})$, defined as:

$$f(C, d_{\mathcal{G}}) = \begin{cases} \left(1 - \frac{1}{d_{\mathcal{G}}}\right)^C & \text{if } C < d_{\mathcal{G}}, \\ 0 & \text{if } C \geq d_{\mathcal{G}} \end{cases},$$

giving

$$\mathbb{E}\left[\Delta_{k,C}\right] \approx \mathbb{E}\left[\Delta_{k,0}\right] \times f(C, d_{\mathcal{G}}).$$

We can therefore also say:

$$ub_{ind} = 2L_{\Gamma_\eta}^K + \frac{\chi}{1 - L_{\Gamma_\eta}} + \xi \sum_{k=1}^{K-1} L_{\Gamma_\eta}^{K-k-1} \mathbb{E}\left[\Delta_{k,0}\right],$$

$$ub_{net} \approx 2L_{\Gamma_\eta}^K + \frac{\chi}{1 - L_{\Gamma_\eta}} + \xi \sum_{k=1}^{K-1} L_{\Gamma_\eta}^{K-k-1} \mathbb{E}\left[\Delta_{k,0}\right] \times f(C, d_{\mathcal{G}}),$$

$$ub_{cent} = 2L_{\Gamma_\eta}^K + \frac{\chi}{1 - L_{\Gamma_\eta}}.$$

We therefore firstly have

$$ub_{ind} - ub_{net} \approx \xi \sum_{k=1}^{K-1} L_{\Gamma_\eta}^{K-k-1} \mathbb{E}\left[\Delta_{k,0}\right] - \xi \sum_{k=1}^{K-1} L_{\Gamma_\eta}^{K-k-1} \mathbb{E}\left[\Delta_{k,0}\right] \times f(C, d_{\mathcal{G}}),$$

which simplifies to

$$ub_{ind} - ub_{net} \approx \xi \sum_{k=1}^{K-1} L_{\Gamma_\eta}^{K-k-1} \mathbb{E}\left[\Delta_{k,0}\right] \times (1 - f(C, d_{\mathcal{G}})).$$

This gives us one of the results, where we focus on the functional dependence on $C$ and $d_{\mathcal{G}}$ by using the tight bound big Theta ($\Theta$):

$$ub_{net} \approx ub_{ind} - \Theta\left(1 - f(C, d_{\mathcal{G}})\right).$$

Secondly, we have

$$ub_{net} \approx ub_{cent} + \xi \sum_{k=1}^{K-1} L_{\Gamma_\eta}^{K-k-1} \mathbb{E}\left[\Delta_{k,0}\right] \times f(C, d_{\mathcal{G}}),$$

giving us the second result

$$ub_{net} \approx ub_{cent} + \Theta\left(f(C, d_{\mathcal{G}})\right).$$

$\square$

**Remark 5.7.** If it is always $\sigma_{k+1}^1$ and $\pi_{k+1}^1$ that is adopted by the whole population (i.e. $i = 1$), then this is exactly the same as the central-agent case. If the $\sigma_{k+1}^j$ and $\pi_{k+1}^j$ that gets adopted has different $j$ for each $k$, then this is akin to a version of the central-agent setting where the index of the representative learning agent may differ for each $k$.

**Remark 5.8.** Thm. 5.6 depends on the assumptions that the communication network is static and fixed, and has the same diameter $d_{\mathcal{G}}$ for all $k$. If we assume instead that the network is only repeatedly jointly connected, we can replace $d_{\mathcal{G}}$ in the results in Thm. 5.6 with $d_{avg} \cdot \omega$, namely the average diameter of the union of each jointly connected collection of graphs multiplied by the average number $\omega$ of graphs in each jointly connected collection. As noted in Rem. 4.3, max-consensus is reached if $C$ is large enough that the number of jointly connected collections of graphs occurring within $C$ is equal to the largest diameter of the union of any collection. This is equivalent to the central-agent case; there is no added benefit to higher values of $C$ than this.

**Remark 5.9.** Thm. 5.6 assumes $\tau_k$ is a small positive value close to 0 such that the softmax function becomes a max function. If we assume instead $\tau_k \in \mathbb{R}_{>0}$ is not close to 0 such that the softmax function is less peaked, then we have $ub_{net} \to ub_{ind}$ as $C \to 0$, and $ub_{net} \to ub_{cent}$ as $C \to \infty$. This is because the spread of policies is now probabilistic rather than deterministic, and depends on the interplay of $\tau_k$ with how large are the differences in the received values of $\sigma_{k+1}^j$. Therefore consensus (and hence reduction in divergence between policies) is reached only asymptotically. This applies to both static, connected networks and to repeatedly jointly connected ones, assuming the latter becomes jointly connected infinitely often.

For completeness, we finally give a stability guarantee that follows from the earlier theorems.

**Theorem 5.10** (Policy-update stability guarantee)**.** *Let Alg. 1 run as per Thm. 5.2 or Thms. 5.3/5.6, and say that $\varepsilon_k$ is the error term at iteration $k = \frac{\log 8\varepsilon_k^{-1}}{\log L_{\Gamma_\eta}^{-1}}$. For all agents $i$, the maximum possible distance between $\pi_k^{i,net}$ and $\pi_{k+1}^{i,net}$ is given by $\mathbb{E}\left[||\pi_k^{i,net} - \pi_{k+1}^{i,net}||_1\right] \le \varepsilon_k + \varepsilon_{k+1} + \mathcal{O}\left(\frac{1}{\sqrt{N}}\right)$. This bound provides a stability guarantee during the learning process; moreover the bound shrinks with each successive $k$ since $\varepsilon_k$ decreases with $k$. Equivalent analysis can also be conducted for both the centralised and independent cases.*

*Proof.* Thms. 5.2, 5.3 and 5.6 bound the difference between each agent's current policy $\pi_k^i$ and the unique equilibrium policy $\pi^*$, with the difference depending on the bias term $\varepsilon_k$ that relates to the iteration $k$ as indicated. Policies $\pi_k^i$ and $\pi_{k+1}^i$ fall within balls centred on $\pi^*$ with radii of $\varepsilon_k + \mathcal{O}\left(\frac{1}{\sqrt{N}}\right)$ and $\varepsilon_{k+1} + \mathcal{O}\left(\frac{1}{\sqrt{N}}\right)$ respectively. This means that the maximum possible distance between $\pi_k^i$ and $\pi_{k+1}^i$ is the sum of these radii, i.e. $\mathbb{E}\left[||\pi_k^i - \pi_{k+1}^i||_1\right] \le \varepsilon_k + \varepsilon_{k+1} + \mathcal{O}\left(\frac{1}{\sqrt{N}}\right)$, giving the result. □

# 6 Practical modifications to theoretical algorithms for empirical use

The theoretical analysis in Sec. 5 requires algorithmic hyperparameters (see Thm. 5.2) that render convergence impractically slow in all of the centralised, independent and networked cases. In particular, the values of $\delta_{mix}$ and $p_{inf}$ give rise to very large $t_0$, causing very small learning rates $\{\beta_m\}_{m\in\{0,...,M_{pg}-1\}}$, and necessitating very large values for $M_{td}$ and $M_{pg}$. Indeed Yardim et al. (2023) do not provide empirical demonstrations of their algorithms for the centralised and independent cases.

For convergence of the algorithms in practical time, we seek to drastically increase $\{\beta_m\}_m$ and reduce $M_{td}$ and $M_{pg}$. We found empirically that the two algorithmic enhancements below helped achieve feasible convergence times with significantly reduced numbers of loops. The first involves recycling transitions using a buffer, and the second gives a principled way of selecting $\sigma_{k+1}^i$ in Line 11 in Alg. 1. There is therefore only a minimal conceptual gap between our theoretical and empirical algorithms, but the replay buffer and reduced numbers of loops do break the theoretical guarantees above, which we trade off for practical convergence. Future works lies in updating the guarantees in light of the practical enhancements.

## 6.1 Algorithm acceleration by use of experience-replay buffer

We modify our Alg. 1 as follows, shown in *blue* in Alg. 2. Instead of using a transition $\zeta_{t-2}^i$ to compute the TD update within each $M_{pg}$ iteration and then discarding the transition, we store the transition in a buffer (Line 9) until after the $M_{pg}$ loops. Replay buffers are a common (MA)RL tool used especially with deep learning, precisely to improve data efficiency and reduce autocorrelation (Lin, 1992; Fedus et al., 2020; Xu et al., 2024a). When learning does take place in our modified algorithm (Lines 11-16), it involves cycling through the buffer for $L$ iterations - randomly shuffling the buffer between each - and thus conducting the TD update on each stored transition $L$ times. This allows us to reduce the number of $M_{pg}$ loops, as well as not requiring as small a learning rate $\{\beta_m\}_m$, allowing much faster learning in practice. Moreover, by shuffling the buffer before each cycle we reduce bias resulting from the dependency of samples along the continued, non-episodic system run, which may justify being able to achieve adequate stable learning even when reducing the number of $M_{td}$ waiting steps within each $M_{pg}$ loop (Sec. 7.4).

The replay buffer allows the first practical demonstrations of all three architectures for learning from a single continued system run. Without it, the empirical learning of our original algorithm is too slow for practical demonstration, as also in the centralised and independent cases - see the ablation study in Sec. 7.4.5. The intuition behind the better learning efficiency resulting from the buffer is as follows. The value of a state-action pair $p$ is dependent on the values of subsequent states reached, but the value of $p$ is only updated when the TD update is conducted on $p$, rather than every time a subsequent pair is updated. By learning from each stored transition multiple times, we not only make repeated use of the reward and transition information in each costly experience, but also repeatedly update each state-action pair in light of its likewise updated subsequent states.

---

**Algorithm 2** Networked learning with experience replay and performance-related generation of $\sigma_{k+1}^i$

---

**Require:** loop parameters $K, M_{pg}, M_{td}, C, L, E$, learning parameters $\eta, \beta, \lambda, \gamma, \{\tau_k\}_{k \in \{0,\ldots,K-1\}}$
**Require:** initial states $\{s_0^i\}_{i=1}^N$
1: Set $\pi_0^i = \pi_{\max}, \forall i$ and $t \leftarrow 0$
2: **for** $k = 0, \ldots, K-1$ **do**
3:    $\forall s, a, i : \hat{Q}_0^i(s,a) = Q_{\max}$
4:    $\forall i$: Empty $i$'s buffer
5:    **for** $m = 0, \ldots, M_{pg} - 1$ **do**
6:       **for** $M_{td}$ iterations **do**
7:          Take step $\forall i : a_t^i \sim \pi_k^i(\cdot|s_t^i), r_t^i = R(s_t^i, a_t^i, \hat{\mu}_t), s_{t+1}^i \sim P(\cdot|s_t^i, a_t^i, \hat{\mu}_t); t \leftarrow t+1$
8:       **end for**
9:       $\forall i$: Add $\zeta_{t-2}^i$ to $i$'s buffer
10:    **end for**
11:    **for** $l = 0, \ldots, L-1$ **do**
12:       $\forall i$ : Shuffle buffer
13:       **for** transition $\zeta_b^i$ in $i$'s buffer $(\forall i)$ **do**
14:          Compute TD update $(\forall i)$: $\hat{Q}_{m+1}^i = \tilde{F}_\beta^{\pi_k^i}(\hat{Q}_m^i, \zeta_{t-2}^i)$ (see Def. 4.1)
15:       **end for**
16:    **end for**
17:    PMA step $\forall i : \pi_{k+1}^i = \Gamma_\eta^{md}(\hat{Q}_{M_{pg}}^i, \pi_k^i)$ (see Def. 4.2)
18:    $\forall i : \sigma_{k+1}^i \leftarrow 0$
19:    **for** $e = 0, \ldots, E-1$ evaluation steps **do**
20:       Take step $\forall i : a_t^i \sim \pi_{k+1}^i(\cdot|s_t^i), r_t^i = R(s_t^i, a_t^i, \hat{\mu}_t), s_{t+1}^i \sim P(\cdot|s_t^i, a_t^i, \hat{\mu}_t)$
21:       $\forall i : \sigma_{k+1}^i \leftarrow \sigma_{k+1}^i + \gamma^e(r_t^i + h(\pi_{k+1}^i(s_t^i)))$
22:       $t \leftarrow t+1$
23:    **end for**
24:    **for** $C$ rounds **do**
25:       $\forall i$ : Broadcast $\sigma_{k+1}^i, \pi_{k+1}^i$
26:       $\forall i : J_t^i = i \cup \{j \in \mathcal{N} : (i,j) \in \mathcal{E}_t\}$
27:       $\forall i$ : Select adopted$^i \sim \Pr\big(\text{adopted}^i = j\big) = \frac{\exp(\sigma_{k+1}^j/\tau_k)}{\sum_{x \in J_t^i} \exp(\sigma_{k+1}^x/\tau_k)} \; \forall j \in J_t^i$
28:       $\forall i : \sigma_{k+1}^i \leftarrow \sigma_{k+1}^{\text{adopted}^i}, \pi_{k+1}^i \leftarrow \pi_{k+1}^{\text{adopted}^i}$
29:       Take step $\forall i : a_t^i \sim \pi_{k+1}^i(\cdot|s_t^i), r_t^i = R(s_t^i, a_t^i, \hat{\mu}_t), s_{t+1}^i \sim P(\cdot|s_t^i, a_t^i, \hat{\mu}_t); t \leftarrow t+1$
30:    **end for**
31: **end for**
32: **return** policies $\{\pi_K^i\}_{i=1}^N$

---

We leave $\beta$ fixed across all iterations, as we found empirically that this yields sufficient learning. We have not experimented with decreasing $\beta$ as $l$ increases, though this may benefit learning.

The transitions in the buffer are discarded after the replay cycles and a new buffer is initialised for the next iteration $k$, as in Line 4. As such the space complexity of the buffer only grows linearly with the number of $M_{pg}$ iterations within each outer loop $k$, rather than with the number of $K$ loops.

## 6.2 Generation of $\sigma_{k+1}^i$

Reducing the number of loops in the hope of achieving practical convergence times can lead to poorer estimation of the Q-function $\hat{Q}_{M_{pg}}^i$, and hence a greater variance in the quality of the updated policies $\pi_{k+1}^i$. This problem will increase with the size of the state and action spaces. In such cases we found empirically that an appropriate method for generating $\sigma_{k+1}^i$ dependent on $\pi_{k+1}^i$ allows our networked algorithm to significantly outperform the independent case by advantageously biasing the spread of particular policies. This is instead of generating $\sigma_{k+1}^i$ arbitrarily as required in the theoretical settings in Sec. 5.

We do so via the steps added in *orange* in Alg. 2, which replace Line 11 in Alg. 1: for $\boldsymbol{\pi}_{k+1} := (\pi_{k+1}^1, \ldots, \pi_{k+1}^N)$, we set $\sigma_{k+1}^i$ to a finite-step approximation $\widehat{\Psi}_{h,k+1}^i(\boldsymbol{\pi}_{k+1}, \upsilon_0)$ of the discounted return $\Psi_{h,k+1}^i(\boldsymbol{\pi}_{k+1}, \upsilon_0)$ (Def. 3.2). The approximation is given by, $\forall i, j \in \{1, \ldots, N\}$

$$\widehat{\Psi}_{h,k+1}^i(\boldsymbol{\pi}_{k+1}, \upsilon_0) = \left[ \sum_{e=0}^{E} \gamma^e (R(s_t^i, a_t^i, \hat{\mu}_t) + h(\pi^i(s_t^i))) \Bigg|_{\substack{s_{t+1}^j \sim P(\cdot | s_t^j, a_t^j, \hat{\mu}_t)}}^{\substack{t=t+e \\ a_t^j \sim \pi_{k+1}^j(s_t^j)}} \right].$$

This is calculated by tracking each agent's discounted return for $E$ evaluation steps (Lines 19-23).

Generating $\sigma_{k+1}^i$ in this way means policies that are more likely to spread through the network are those estimated to receive a higher return in reality, despite being generated from poorly estimated Q-functions, biasing the population towards faster learning. Naturally the quality of the finite-step approximation depends on the number of evaluation steps $E$, but we found empirically that $E$ can be much smaller than $M_{pg}$ and still give marked convergence benefits.

## 7    Experiments

Our technical contribution of the replay buffer to MFG algorithms for online learning from non-episodic system runs allows us also to contribute the first empirical demonstrations of these algorithms, not just in the networked case but also in the central-agent and independent cases. The latter two serve as baselines to show the advantages of the networked architecture. Experiments were conducted on a MacBook Pro, Apple M1 Max chip, 32 GB, 10 cores. We use `scipy.optimize.minimize` (employing Sequential Least Squares Programming) to conduct the optimisation step in Def. 4.2, and the JAX framework to accelerate and vectorise some elements of our code. For reproducibility, our code is included in the publicly available Supplementary Material.

### 7.1    Games

We follow the gold standard in prior works on stationary MFGs regarding the types of game demonstrated: we focus on grid-world environments where agents can move in the four cardinal directions or remain in place (Laurière, 2021; Laurière et al., 2022b; Zaman et al., 2023; Algumaei et al., 2023; Cui et al., 2023a; Wu et al., 2024c). While this type of experiment is characteristic of similar MFG works, we recognise that these are simple games. They nevertheless serve as useful preliminary demonstrations of the validity of our algorithms and the considerations necessary for achieving practical learning; we leave experiments in more complex environments to future work, which would likely require extending the algorithms to handle non-tabular Q-functions. Moreover, grid-world environments naturally reflect the deployed, spatial applications in which we are interested in our setting, where agents learn online and communicate with neighbours on a network (which is likely to be defined spatially, though is not restricted to such a case).

We conduct numerical tests with two tasks (defined by the agents' reward functions), chosen for being particularly amenable to intuitive understanding of whether the agents are learning behaviours that are appropriate and explainable for the respective objective functions. In all cases, rewards are normalised in [0,1] after they are computed.

**Cluster.**    This is the inverse of the 'exploration' game in (Laurière et al., 2022b), where in our case agents are encouraged to gather together by the reward function $R(s_t^i, a_t^i, \hat{\mu}_t) = \log(\hat{\mu}_t(s_t^i))$. That is, agent $i$ receives a reward that is logarithmically proportional to the fraction of the population that is co-located with it at time $t$. We give the population no indication where they should cluster, agreeing this themselves over time.

**Agree on a single target.**    Unlike in the above 'cluster' game, the agents are given options of locations at which to gather, and they must reach consensus among themselves. If the agents are co-located with one of a number of specified targets $\phi \in \Phi$ (in our experiments we place one target in each of the four corners of the grid), and other agents are also at that target, they get a reward proportional to the fraction of the population found there; otherwise they receive a penalty of -1. In other words, the agents must coordinate

on which of a number of mutually beneficial points will be their single gathering place. The reward function is given by $R(s_t^i, a_t^i, \hat{\mu}_t) = r_{targ}(r_{collab}(\hat{\mu}_t(s_t^i)))$, where

$$r_{targ}(x) = \begin{cases} x & \text{if } \exists \phi \in \Phi \text{ s.t. } \text{dist}(s_t^i, \phi) = 0 \\ -1 & \text{otherwise,} \end{cases}$$

$$r_{collab}(x) = \begin{cases} x & \text{if } \hat{\mu}_t(s_t^i) > 1/N \\ -1 & \text{otherwise.} \end{cases}$$

These are both coordination games, where selfish agents can increase their individual rewards by following the same strategy as others and therefore have an incentive to communicate policies. Moreover, they require more sophisticated solutions than the dispersal/exploration games often considered in similar MFG works (Laurière et al., 2022b; Zaman et al., 2023; Wu et al., 2024c), where a trivial starting policy that encourages agents to move across the grid at random may already be close to the equilibrium policy.

## 7.2 Experimental metrics

To give as informative results as possible about both performance and proximity to the NE, we provide three metrics for each experiment. All metrics are plotted with 2-sigma confidence intervals ($2 \times$ standard deviation), computed over 10 trials (each with a random seed) of the system evolution in each setting. This is computed based on a call to `numpy.std` for each metric over each run.

### 7.2.1 Exploitability

Works on MFGs most commonly use the *exploitability* metric to evaluate how close a given policy $\pi$ is to a NE policy $\pi^*$ (Perrin et al., 2020; Pérolat et al., 2022; Laurière et al., 2022a;b; Algumaei et al., 2023; Wu et al., 2024c). The metric usually assumes that all agents are following the same policy $\pi$, and quantifies how much an agent could benefit by deviating from $\pi$, by measuring the difference between the return $V_h$ (Def. 3.4) gained by $\pi$ and that gained by a policy that best responds to the population distribution generated by $\pi$. Let us denote by $\mu^\pi$ the distribution generated when $\pi$ is the policy followed by all of the population aside from the deviating agent; then the exploitability of policy $\pi$ is defined as follows:

**Definition 7.1** (Exploitability of $\pi$)**.** The exploitability $\mathcal{E}$ of policy $\pi$ is given by:

$$\mathcal{E}(\pi) = \max_{\pi'} V_h(\pi', \mu^\pi) - V_h(\pi, \mu^\pi).$$

If $\pi$ has a large exploitability then an agent can significantly improve its return by deviating from $\pi$, meaning that $\pi$ is far from $\pi^*$, whereas an exploitability of 0 implies that $\pi = \pi^*$ - i.e. lower exploitability is considered better.

Since we do not have access to the exact best response policy $\arg\max_{\pi'} V_h(\pi', \mu^\pi)$ as in some related works (Laurière et al., 2022b; Wu et al., 2024c), we instead approximate the exploitability metric, similarly to (Perrin et al., 2021), as follows. We freeze the policy of all agents apart from a deviating agent, for which we store its current policy and then conduct 40 'deviation' $k$ loops of policy improvement. To approximate the expectations in Def. 7.1, we take the best return of the deviating agent across the 40 $k$ loops, as well as the mean of all the other agents' returns across these same loops. We then revert the agent back to its stored policy, before learning continues for all agents. Due to the expensive computations required for this metric, we evaluate it only on alternate $k$ iterations of the actual system evolution (for our ablation study of the experience replay buffer in Sec. 7.4.5, we evaluate only every 20 $k$).

Since prior works conducting empirical testing have generally focused on the centralised setting, evaluations have not had to consider the exploitability metric when not all agents are following a single policy $\pi_k$, as may occur in the independent or networked settings, i.e. when $\pi_k^i \neq \pi_k^j$ for $i, k \in \{1, \ldots, N\}$. The method described above for approximating exploitability involves calculating the mean return of all non-deviating agents' policies. While this is $\pi_k$ in the centralised case, if the non-deviating agents do not share a single

policy, then this method is in fact approximating the exploitability of their joint policy $\boldsymbol{\pi}_k^{-d}$, where $d$ is the deviating agent.

*The exploitability metric has a number of limitations in our setting.* In coordination games (the setting for our tasks), agents benefit by following the same behaviour as others, and so a deviating agent generally stands to gain less from a 'best-responding' policy than it might in the non-coordination games on which many other works focus. For example, the return of a best-responding agent in the 'cluster' game still depends on the extent to which other agents coordinate on where to cluster, meaning it cannot significantly increase its return by deviating from a badly clustering policy. This means that the downward trajectory of the exploitability metric is less clear in our plots than in other works.

Moreover, our approximation takes place via policy improvement steps (as in the main algorithm) for an independent, deviating agent while the policies of the rest of the population are frozen. As such, the quality of our approximation is limited by the number of policy-improvement/expectation-estimation rounds, which must be restricted for the sake of the running speed of the experiments. Moreover, since one of the findings of our paper is that networked agents can improve their policies faster than independent agents, it is arguably unsurprising that approximating the best response by an independently deviating agent sometimes gives an unclear and noisy metric.

Given the limitations presented by approximating exploitability, we also provide the second metric to indicate the progress of learning.

### 7.2.2 Average discounted return

We record the average finite-step discounted return of the agents' policies $\pi_k^i$ during the $M_{pg}$ steps of each outer $k$ loop. This allows us to observe that settings that converge to similar exploitability values may not have similar average agent returns, suggesting that some algorithms are better than others not just at finding equilibria, but also at finding preferable equilibria (when the assumption of a unique MFG-NE is removed by reducing regularisation; see Sec. 7.4) - cf. Graber (2025); Li et al. (2025c). See, for example, Fig. 8, where the networked agents converge to similar exploitability as the independent agents, but receive higher average reward.

### 7.2.3 Policy divergence

We record the population's average policy divergence $\frac{1}{N}\Delta_k := \frac{1}{N}\sum_{i=1}^{N}||\pi_k^i - \pi_k^1||_1$ for the arbitrary policy $\bar{\pi} = \pi^1$. Many of our theoretical results and proofs relate to the policy divergence, and in Sec. 5 we show extensively how the comparatively worsening sample complexities between the centralised, networked and independent cases are the result of their range of policy divergences. We therefore include this metric to show how this relationship affects learning in practice.

Furthermore, the theoretical guarantees assume that the population is trying to learn the unique equilibrium policy $\pi^*$, with the implication that all agents should end up with this identical policy, regardless of the learning architecture (Sec. 5). However, we find in practice that populations may be converging (in terms of exploitability/return) while having non-diminishing policy divergence, particularly in the independent setting. We therefore also include this metric to indicate the difference between theoretical and empirical convergence.

### 7.3 Hyperparameters

See Table 1 for our hyperparameter choices. In general, we seek to show that our networked algorithm is robust to 'poor' choices of hyperparameters, such as low numbers of iterations, as may be required when aiming for practical convergence times in complex real-world problems. By contrast, the convergence speed of the independent algorithm suffers much more significantly without idealised hyperparameter choices. As such, our experimental demonstrations in the plots generally involve hyperparameter choices at the low end of the values we tested during our research.

Table 1: Hyperparameters

| Hyper-param. | Value | Comment |
|---|---|---|
| Grid-size | 8x8 / 16x16 | Most experiments are run on the smaller grid, while Figs. 8 and 9 showcase learning in a larger state space. |
| Trials | 10 | We run 10 trials with different random seeds for each experiment. We plot the mean and 2-sigma error bars for each metric across the trials. |
| Pop. | 250 | We tested $N$ in $\{25,50,100,200,250\}$, with the networked architecture generally performing equally well with all population sizes $\geq 50$. We chose 250 for our demonstrations, to show that our algorithm can handle large populations, indeed often larger than those demonstrated in other mean-field works, especially for grid-world environments (Yang et al., 2018b; Subramanian & Mahajan, 2019; Ganapathi Subramanian et al., 2020; 2021; Cui & Koeppl, 2021; Yongacoglu et al., 2024; Subramanian et al., 2022; Guo et al., 2023; Cui et al., 2023a). In experiments testing robustness to population increase, the population instead begins at 50 agents and has 200 added at the marked point. |
| $K$ | 200 / 400 | $K$ is chosen to be large enough to see exploitability reducing, and converging where possible. |
| $M_{pg}$ | 500 / 1000 | We wish to illustrate the benefits of our networked architecture and replay buffer in reducing the number of loops required for convergence, i.e. we wish to select a low value that still permits learning. We tested $M_{pg}$ in $\{300,500, 600,800,1000,1200,1300,1400,1500,1800,2000,2500,3000\}$, and chose 500 for demonstrations on the 8x8 grids, and 1000 for the 16x16 grids. It may be possible to optimise these values further in combination with other hyperparameters. |
| $M_{td}$ | 1 | We tested $M_{td}$ in $\{1,2,10,100\}$, and found that we could still achieve convergence with $M_{td} = 1$. This is much lower than the requirements of the theoretical algorithms, essentially allowing us to remove the innermost nested learning loop. |
| $C$ | 1 | We tested $C$ in $\{1,5,10\}$. We choose 1 to show the convergence benefits brought by even a single communication round, even in networks that may have limited connectivity. |
| $L$ | 100 | As with $M_{pg}$, we wish to select a low value that still permits learning. We tested $L$ in $\{50,100,200,300,400,500\}$. In combination with our other hyperparameters, we found $L \leq 50$ led to less good results, but it may be possible to optimise this hyperparameter further. |
| $E$ | 100 | We tested $E$ in $\{100,300,1000\}$, and choose the lowest value to show the benefit to convergence even from few evaluation steps. It may be possible to reduce this value further and still achieve similar results. |
| $\gamma$ | 0.9 | Standard choice across RL literature. |
| $\beta$ | 0.1 | We tested $\beta$ in $\{0.01,0.1\}$ and found 0.1 to be small enough for adequate learning at an acceptable speed. Further optimising this hyperparameter (including by having it decay with increasing $l \in 0,\dots,L-1$, rather than leaving it fixed) may lead to better results. |
| $\eta$ | 0.01 | We tested $\eta$ in $\{0.001,0.01,0.1,1,10\}$ and found that 0.01 gave stable learning that progressed sufficiently quickly. |
| $\lambda$ | 0 | We tested $\lambda$ in $\{0,0.0001,0.001,0.01,0.1,1\}$. Since we can reduce $\lambda$ to 0 with no detriment to empirical convergence, we do so in order not to bias the NE. |
| $\tau_k$ | cf. comment | For fixed $\tau_k$ $\forall k$, we tested $\{1,10,100,1000\}$. In our experiments for fixed $\tau_k$ the value is 100 (see Figs. 10 and 11); this yields learning, but does not perform as well as if we anneal $\tau_k$ as follows. We begin with $\tau_0 = 10000/(10**\lceil(K-1)/10\rceil)$, and multiply $\tau_k$ by 10 whenever $k \bmod 10 = 1$ i.e. every 10 iterations. Further optimising the annealing process may lead to better results. |

### 7.4 Results and discussion

We first provide results for our standard algorithmic setting, before giving robustness studies and numerous ablations (note that the independent setting serves as an implicit ablation of our communication scheme). We summarise findings in the body of each sub-section, while the specific results are discussed fully in each figure's caption. In each plot the decimals refer to each agent's broadcast radius as a fraction of the maximum possible distance in the grid (i.e. the diagonal).

We preempt possible concerns regarding the wide confidence intervals in many of our plots by saying that many works with similar experiments do not report error bars at all, and if they do they usually only give 1-sigma intervals, whereas we give 2-sigma (Laurière, 2021; Laurière et al., 2022b; Zaman et al., 2023; Algumaei et al., 2023; Cui et al., 2023a). Moreover, the central-agent architecture usually has similar or higher variance to the networked agents in the plots, indicating that this is not an issue introduced by our communication algorithm; it is instead likely to be due to poor estimation of the Q-function when using the small numbers of loops required for practical runtimes. The independent agents have very low variance, but this is because they hardly appear to increase their returns at all in most cases.

We also give the following remark regarding the exploitability metric in some of our experimental plots, relating to the issues with this metric in coordination games, as discussed in Sec. 7.2.1:

**Remark 7.2.** The reward structure of our coordination games is such that exploitability sometimes increases from its initial value before it decreases down to 0 (e.g. Fig. 2). This is because agents are rewarded proportionally to how many other agents are co-located with them: when agents are evenly dispersed at the beginning of the run, it is difficult for even a deviating, best-responding agent to significantly increase its reward. However, once some agents start to aggregate, a best-responding agent can take advantage of this to substantially increase its reward (giving higher exploitability), before all the other agents catch up and aggregate at a single point, reducing the exploitability down to 0. Due to this arc, in some of our plots the independent case may have lower exploitability at certain points than the other architectures, but this is not necessarily a sign of good performance. In fact, in such cases we can often see that the independent agents are hardly learning at all, with the independent agents' average return not increasing and the exploitability staying level rather than ultimately decreasing (see, for example, Figs. 2, 4, 6 and 8).

### 7.4.1 Standard experimental setting

Even with only a single communication round in each of the $K$ loops, networked agents learn faster and reach higher returns than independent agents, which hardly appear to learn at all. Moreover networked agents appear to match the central-agent population in the 'cluster' game (Fig. 2). Our experiments show that our practical algorithmic enhancements enable convergence within a practical number of iterations even when we remove a number of the assumptions required for the theoretical algorithms:

- We reduce $M_{pg}$ by many orders of magnitude from its theoretically required value (see Sec. 6), while still converging within a reasonable $K$. We keep the learning rate $\beta$ fixed, removing the annealing scheme for $\{\beta_m\}_{m\in\{0,...,M_{pg}-1\}}$ required in the theorems, and use a much higher value.

- In our experiments we do not ensure that the communication network $\mathcal{G}_t$ remains static and connected, nor that the diameter $d_{\mathcal{G}}$ of the network is equal for all $k$. Nevertheless, even with a single communication round the networked agents learn faster than independent ones (which hardly learn at all), sometimes performing similarly to the centralised case.

- The $M_{td}$ parameter is theoretically required for the learner to wait between collecting samples when learning from the empirical distribution in a non-episodic system run. However, our replay buffer allows us to reduce it to 1, effectively removing the innermost loop of the nested learning algorithm (see Line 5 of Alg. 1).

- We can reduce the scaling parameter $\lambda$ of the entropy regulariser to 0, i.e. we converge even without regularisation, allowing us to leave the MFG-NE unbiased and also removing Assumption 3.16. In general an unregularised MFG-NE is not unique (Yardim et al., 2023); the ability of centralised

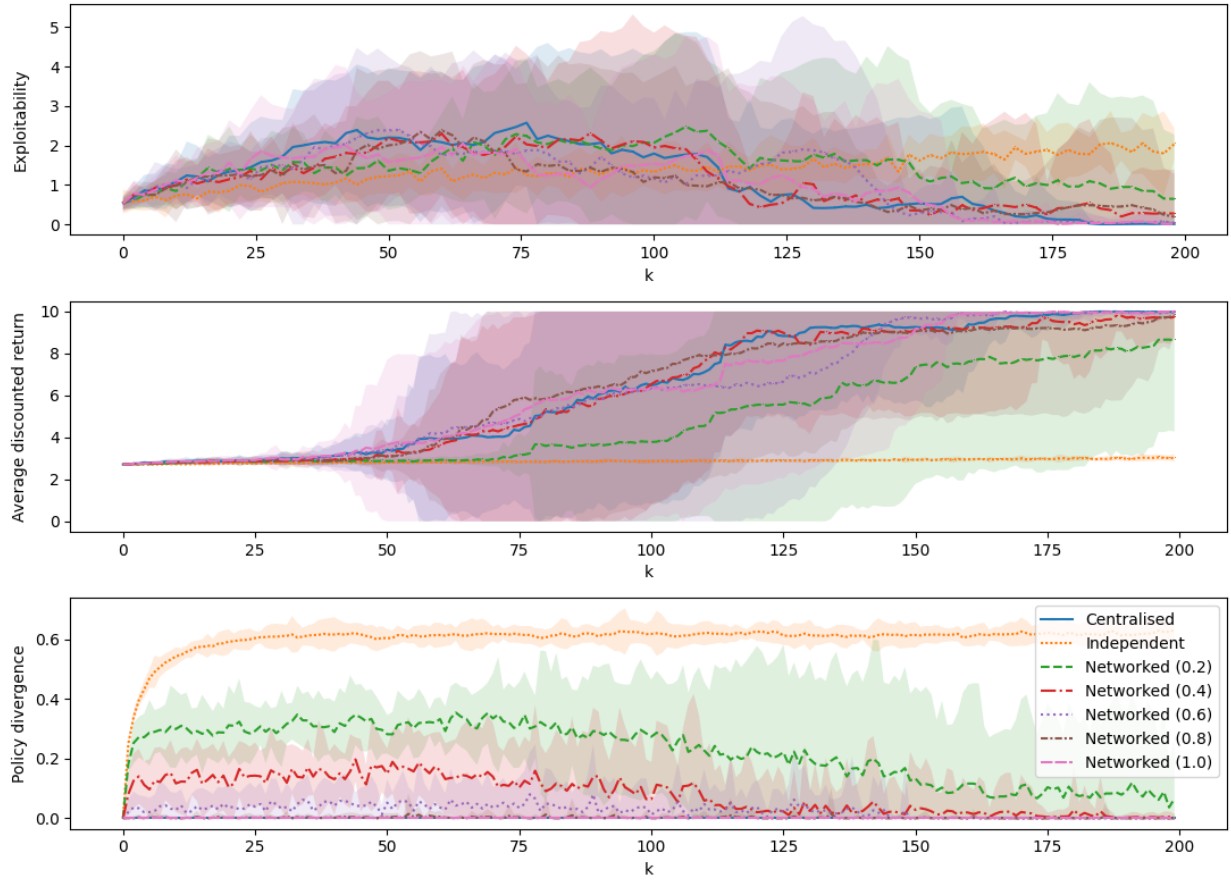

Figure 2: 'Cluster' game. Even with only a single communication round, our networked architecture significantly outperforms the independent case, which hardly appears to be learning at all. All broadcast radii except the smallest (0.2, green) have similar mean exploitability and return to the centralised case.

and networked agents to coordinate on one of the multiple possible solutions may explain why they outperform the independent case, as discussed further below (cf. Graber (2025); Li et al. (2025c)).

- For the PMA operator (Def. 4.2), we conduct the optimisation over the set $u \in \Delta_{\mathcal{A}}$ instead of $u \in \mathcal{U}_{L_h}$, i.e. we can choose from all possible distributions over actions instead of needing to identify the Lipschitz constants given in Assumption 3.9.

We now give further intuition into the benefits of our communication scheme in our empirical settings where multiple equilibria are possible. For sufficiently high $\lambda$ the MFG-NE is unique, and involves all the agents constantly moving about with high entropy, at the cost of biasing the problem. However, when $\lambda$ is 0, the 'target agreement' and 'cluster' tasks both explicitly admit multiple Nash equilibria. In a given trial of the 'target agreement' task, all the agents could converge to remaining stationary at any one of the four corners, and any one of these four situations would lead to the highest possible returns. We found in our experiments that with the different random seeds for each trial, agents did end up converging to a different corner at random each time. Similarly in the 'cluster' task: for a given trial all the agents could converge to remaining stationary in any one of the grid points, and any one of these *height × width* situations would lead to the highest possible returns. (In practice, empirically we found that the agents usually converged at random to one of the corners in the 'cluster' task as well, rather than to anywhere on the grid. This is because in the early stages of the trial, when agents start with random policies, they already spend more time visiting

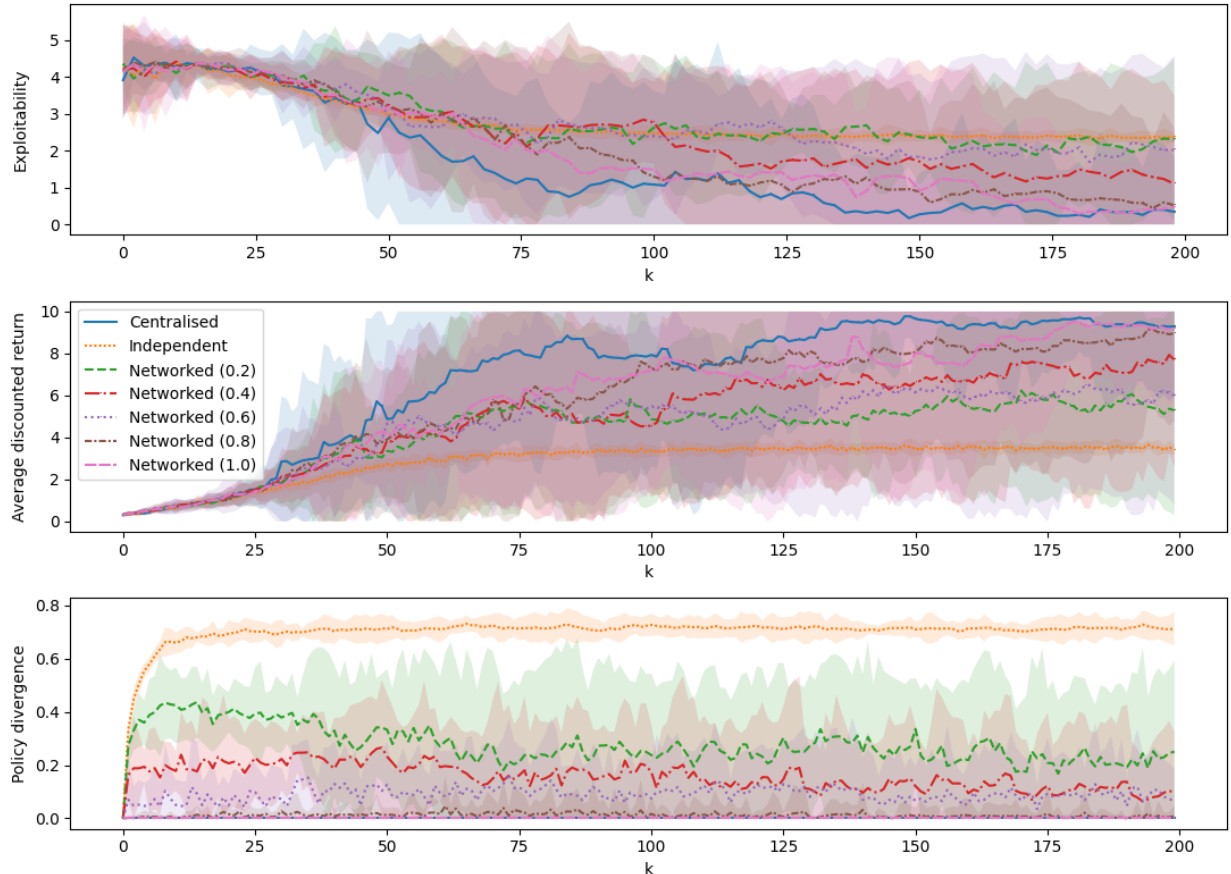

Figure 3: 'Target agreement' game. Even with only a single communication round, our networked case out-performs the independent case with respect to exploitability and return. The fact that the lowest broadcast radius (0.2, green) ends with similar exploitability to the independent case yet higher return suggests our networked algorithm might help agents find 'preferable' equilibria.

corners, because at any corner three actions will keep them in place, since they cannot move off the edge of the grid).

The discussion so far applies to *Nash* equilibria, i.e. the situations where agents end up with the highest possible returns (equivalent to a normalised average return of 10 in the plots). Population distributions can also be at an equilibrium that is not Nash nor one that receives particularly high returns: we can broadly characterise three situations here:

1. Agents, which begin the trial with random policies, never manage to reach any critical mass that breaks the ties between the possible coordination points, so continue moving about the grid with a high degree of entropy forever, even if $\lambda$ is 0. This is most likely what is happening for the independent agents across the experiments, and is why they usually converge to low returns.

2. The population gets segregated into two or more isolated parts of the grid, each of which would otherwise give a Nash equilibrium if the whole population were present e.g. half the population learns a policy that remains in the top left corner while the other half learns to stay at the bottom right. If the policies do not retain enough exploration, the agents will never discover the other isolated groups with which they could combine for mutual benefit (whilst if there is too much exploration, we revert to one of the other suboptimal situations, depending on the value of $\lambda$).

3. The population is not segregated, but oscillates between two or more locations that would otherwise represent Nash equilibria, without ever being able to settle on stable policies that agree on one location. This is similar to Case 1, but with the number of meeting points that are visited having been narrowed down.

Case 1 is likely to receive the worst returns. How much worse Case 2 and 3 are than the Nash equilibria depends on the size of the segregated populations and/or the frequency of the visitations caused by the oscillations. The ability of learning architectures to align the behaviour of the *whole* population on a *single* choice of Nash equilibrium location determines how close to the maximum return the population will receive. The independent case has no way to align policies outside of the signal from the returns themselves; if no critical mass ever forms to show differentiation in the returns, then the independent population will always remain at a low performing equilibrium. The central-agent case has an inherent method for aligning the policies of the whole population, but these policies may still oscillate between locations that would otherwise be Nash equilibria, which is why central-agent populations do not always reach the maximum returns in our plots.

Our communication algorithm provides a method both for 1) aligning agents' policies, and for 2) choosing better performing policies on which to align (where both of these elements contribute to the selection of better equilibria). This is why we see our decentralised, networked populations receiving higher returns than the independent ones, as our algorithm helps agents to get out of the worse performing equilibria. (In principle, under the right conditions, our communication paradigm could even outperform the central-agent case: the latter aligns the population on a policy update of arbitrary quality, generated by arbitrary agent $i = 1$, rather than aligning on better performing policies.) The degree to which our communication algorithm leads to policy consensus depends upon the network connectedness and the number of communication rounds. Since in our experiments we use $C = 1$, it is the network connectedness - determined by the size of the broadcast radius - that has the greatest effect (for greater numbers of communication rounds, this may matter less). This is why we see the populations with higher broadcast radii converging to higher returns faster than populations with lower broadcast radii, which are in turn more capable than entirely independent agents - they are better able to align the population so as to converge to equilibria that are closer to optimal (Nash) equilibria.

In summary, the fact that different populations in our experiments do not just improve their returns at different speeds, but actually appear to converge to different final returns, is reflective of them settling at different equilibria that give different returns. Our communication algorithm actively helps populations to settle at equilibria that are closer to optimal, i.e. 'preferable' (and in so doing, to choose between multiple possible Nash equilibria).

### 7.4.2 Robustness experiments

We consider two scenarios to which we desire real-world many-agent systems (e.g. robotic swarms, autonomous vehicle traffic, etc.) to be robust. The networked setup affords population **fault-tolerance** and **online scalability**, which are motivating qualities of many-agent systems.

**Fault-tolerance** We consider a scenario in which the learning/updating procedure of agents fails with a certain probability within each iteration, in which cases $\pi_{k+1}^i = \pi_k^i$ (see Figs. 4 and 5 for our experimental results in this scenario). In real-life decentralised settings, this might be particularly liable to occur since the updating process might only be synchronised between agents by internal clock ticks, such that some agents may not complete their update in the allotted time but will nevertheless be required to take the next step in the environment. Regardless of their cause, such failures slow the improvement of the population in the independent case, and in the central-agent population it means no improvement occurs at all in any iteration in which failure occurs, as there is a single point of failure. Networked communication instead provides redundancy in case of update failures, with the updated policies of any agents that have managed to learn spreading through the population to those that have not (cf. Horyna et al. (2025)). This feature thus ensures that improvement can continue for potentially the whole population even if a high number of agents do not manage to learn at a given iteration.

Our experimental setup for this scenario is as follows: at every $k$ iteration each learner (whether centralised or decentralised) fails to update its policy (i.e. Line 10 of Alg. 1 is not executed such that $\pi^i_{k+1} = \pi^i_k$) with a 50% probability. The communication network allows agents that have successfully updated their policies to spread this information to those that have not, providing redundancy that the centralised and independent settings do not have. See Figs. 4 and 5.

**Online scalability**  We may want to arbitrarily increase the size of a population of agents that are already learning or operating in the environment (we can imagine extra fleets of autonomous cars or drones being deployed) - see Sec. 2 for comparison with other works considering this type of robustness (Dawood et al., 2023; Eck et al., 2023; Gao et al., 2024; Wu et al., 2024c). A purely independent setting would require all the new agents to learn a policy individually given the existing distribution, and the process of their following and improving policies from scratch may itself disturb the MFG-NE that has already been achieved by the original population. With a communication network, however, the policies that have been learnt so far can quickly be shared with the new agents in a decentralised way, hopefully before their unoptimised policies can destabilise the current MFG-NE. This would provide, for example, a way to bootstrap a large population from a smaller pre-trained group, if training were considered expensive in a given setting.

Our experimental setup for this scenario is as follows: instead of having 250 agents throughout, the population begins with 50 agents learning normally, and a further 200 agents are added to the population at the marked point. The networked architectures are quickly able to spread the learnt policies to the newly arrived agents such that learning progress is minimally disturbed, whereas convergence is significantly impacted in the independent case. See Figs. 6 and 7.

The remainder of our experiments provide further studies and ablations in the standard settings (i.e. not the robustness scenarios):

### 7.4.3  Experiments on larger grid

Figs. 8 and 9 show the result of learning on a grid of size 16x16 instead of 8x8 as in all other experiments. There is at times greater differentiation in this setting than in the 8x8 grid between the performances of the different broadcast radii of the networked architecture (as is to be expected in a less densely populated environment). The networked architecture continues to outperform the independent case for most broadcast radii.

### 7.4.4  Ablation study of softmax temperature annealing scheme

Figs. 10 and 11 illustrate the effect of fixed $\{\tau_k\}_{k \in \{0,\dots,K-1\}} = 100$, where the networked architecture does not perform as well as if we use the stepped annealing scheme employed in all the other experiments and detailed in Table 1. The intuition behind the better performance achieved with the annealing scheme is as follows. If we begin with small $\tau_k$ (such that the softmax approaches being a max function), we heavily favour the adoption of the highest rewarded policies to speed up progress in the early stages of learning. Subsequently we increase $\tau_k$ in steps, promoting greater randomness in adoption, so that as the agents come closer to equilibrium, poorer policy updates that nevertheless receive a high return (due to randomness) do not introduce too much instability to learning and prevent convergence.

### 7.4.5  Ablation study of experience replay buffer

Figs. 12 and 13 illustrate the importance of our incorporation of the experience replay buffer. Without it, as in the original theoretical version of the algorithms, there is no noticeable improvement in any of the agents' returns, i.e. no noticeable learning, even after $K = 400$ iterations. When removing the buffer for these experiments we run the core learning section of the algorithm as in Lines 3-10 of Alg. 1, keeping the hyperparameters the same as in our main experiments, i.e. $M_{pg} = 500$, $M_{td} = 1$, etc. (see Tab. 1). These experiments are run for 5 trials rather than 10 as in all other cases, and with exploitability evaluated every 20 $k$ instead of every 2 $k$ for computational efficiency.

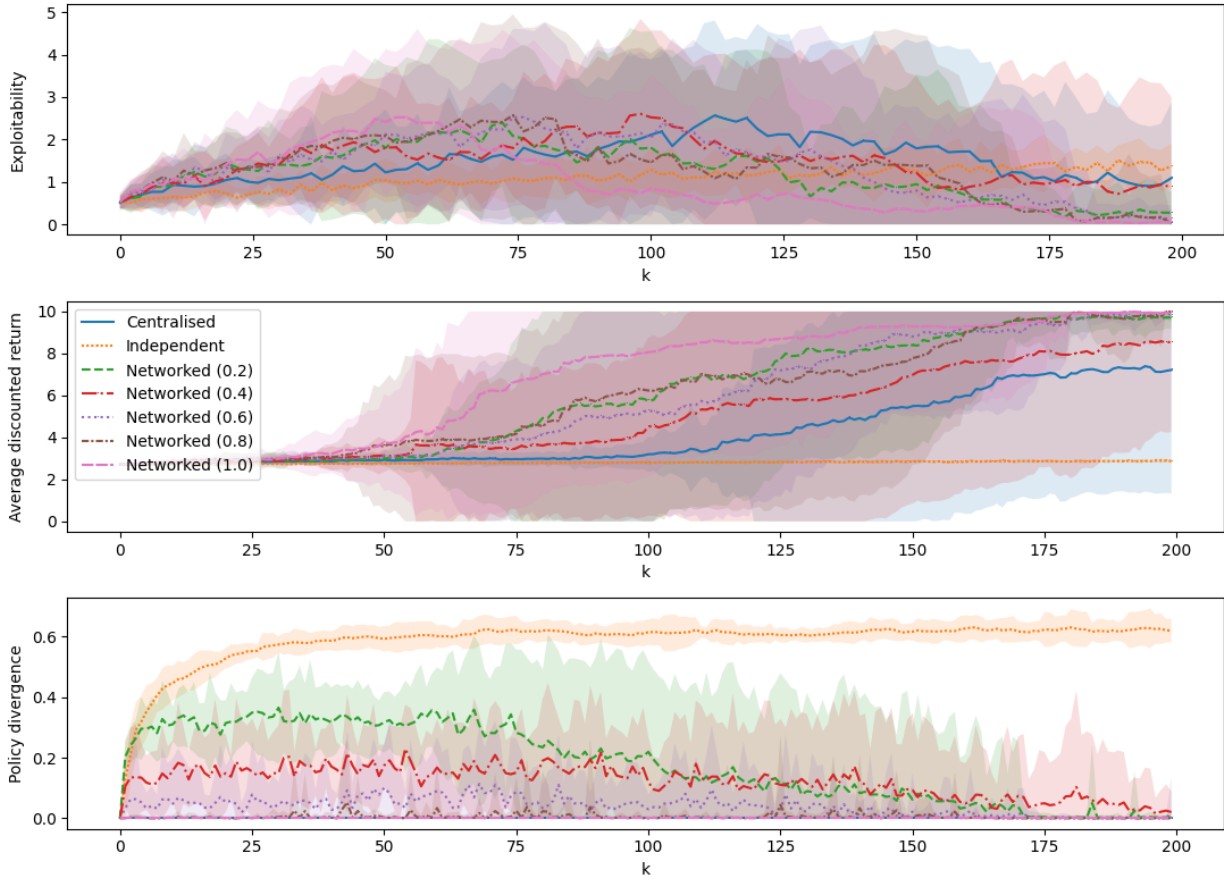

Figure 4: 'Cluster' game, testing robustness to 50% probability of policy update failure. The communication network allows agents that have successfully updated their policies to spread this information to those that have not, providing redundancy. Independent learners cannot do this and hardly appear to learn at all (no increase in return); likewise the centralised population is susceptible to its single point of failure and learns slower than before. Thus our networked architecture outperforms both the centralised and independent cases.

## 8 Conclusion and future work

We contributed networked communication as a novel framework for learning MFGs from the empirical distribution, and provided accompanying theoretical and practical algorithms. We showed theoretically and experimentally that networked agents can considerably outperform independent ones, often performing similarly to the central-agent architecture while avoiding the restrictive assumption of the latter and its single point of failure.

Our experiments are based on relatively simple examples that demonstrate the advantages of our new approach, but which lack the complexity of the real-world applications to which we wish to address the approach. Moreover in our current experiments only the reward function depends on the mean-field distribution, and not the transition function, even though this is possible in theory; we will explore this element in future experiments. It is feasible that in more complex problems, it may not be possible to reduce hyperparameter values to the same extent we have demonstrated in our experimental examples.

Moreover, real-world examples would likely require handling larger and continuous state/action spaces (the latter perhaps building on related work such as Tang et al. (2024)), which in turn may require (non-linear)

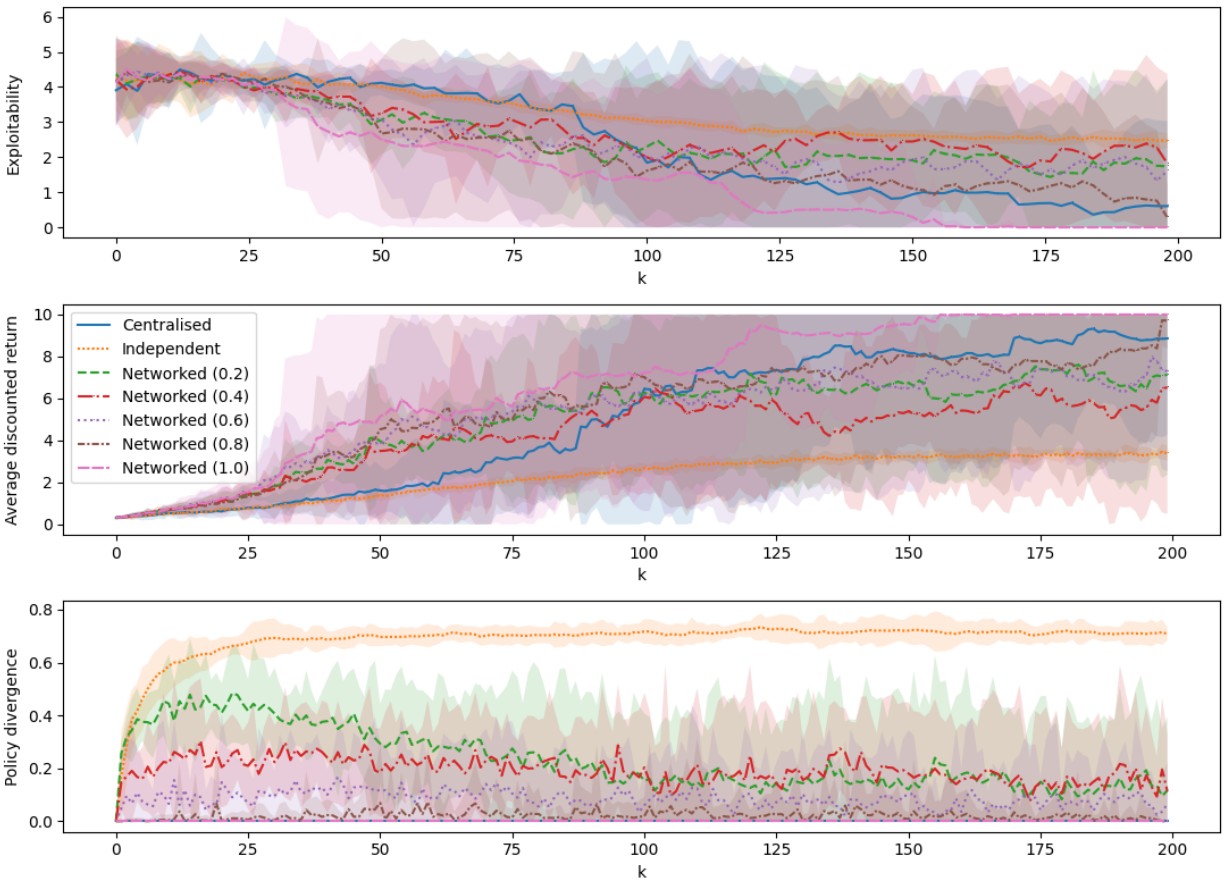

Figure 5: 'Target agreement' game, testing robustness to 50% probability of policy update failure. All the networked cases outperform the independent case and also learn faster than the centralised case for long periods. The communication network allows agents that have successfully updated their policies to spread this information to those that have not, providing redundancy. Independent learners cannot do this so have even slower convergence than normal in this task; likewise the centralised architecture is susceptible to its single point of failure, hence learning can be slower than in the networked case.

function approximation. Future work therefore involves incorporating neural networks into our networked communication architecture for oracle-free, non-episodic MFG settings. Extending our algorithms in this way, which would depend on modifying the PMA step (Vieillard et al., 2020; Wu et al., 2024c), would allow us to introduce communication networks to MFGs with *non-stationary* equilibria, in addition to those with larger state/action spaces. Our method for non-stationary games will likely have agents' policies depending both on their local state and also on the population distribution (Mishra et al., 2020; Laurière et al., 2022a; Perrin et al., 2022; Carmona et al., 2023), but such a high-dimensional observation object is only possible with function approximation. The present work demonstrates the benefits of the networked communication architecture when the Q-function is poorly estimated and introduces experience relay buffers to the setting of learning from a non-episodic run of the empirical system. Both elements are an important bridge to employing (non-linear) function approximation in this setting, where the problems of data efficiency and imprecise value estimation can be even more acute, and where we may want to employ experience replay buffers to provide uncorrelated data to train the neural networks (Zhang & Sutton, 2017). When the policy functions are approximated rather than tabular, our agents would communicate the functions' parameters instead of the whole policy as now.

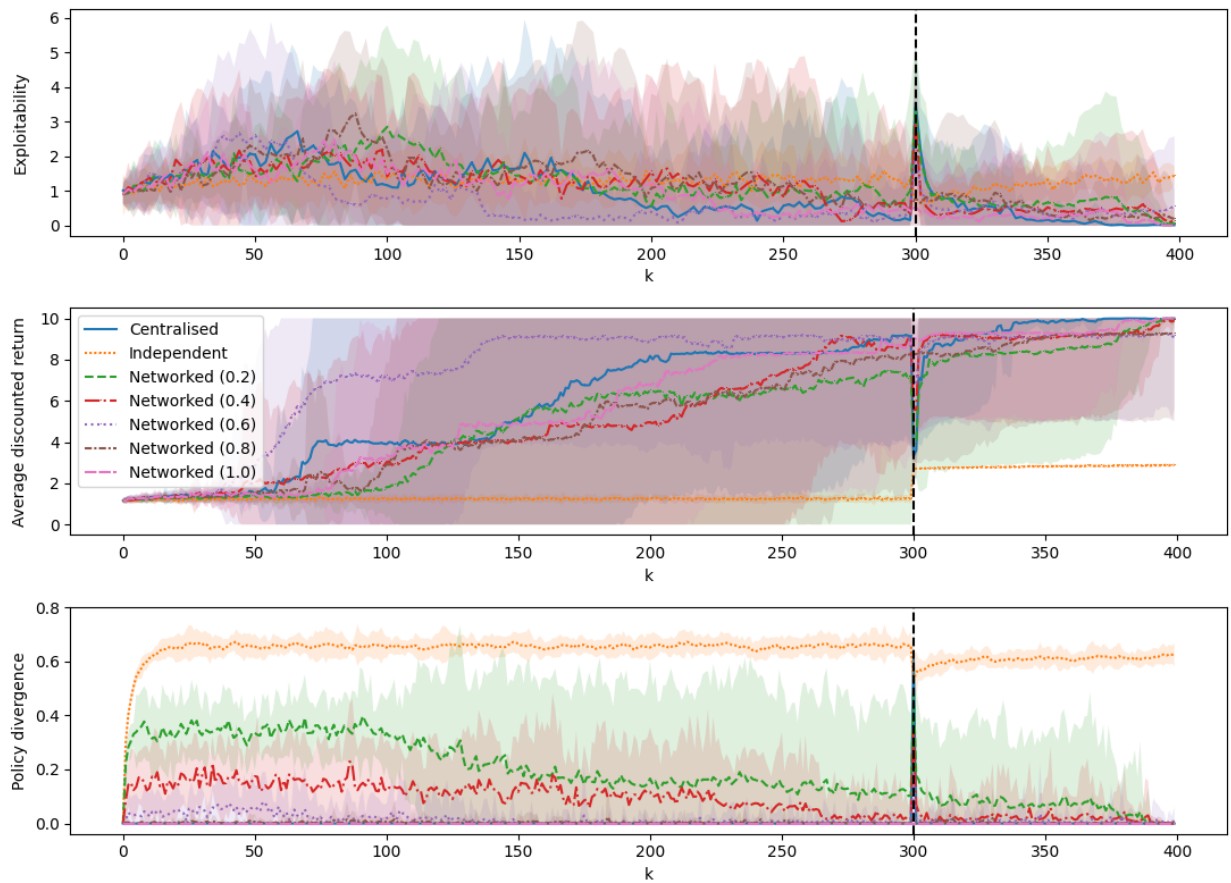

Figure 6: 'Cluster' game, testing robustness to a five-times increase in population. While the independent algorithm appears to enjoy similar exploitability to the other cases (see Rem. 7.2), we can see from its average return that it is not in fact learning at all; while the return rises after the increase in population size this is only because there are now more agents with which to be co-located, rather than because learning has progressed. Since here, unlike in the 'target agreement' game in Fig. 7, independent agents have hardly improved their return in the first place, we do not see the adverse effect that the addition of agents to the population has on the progress of learning. All networked populations perform similarly to or outperform the centralised case, and all markedly outperform the independent case in terms of return. The communication network allows the learnt policies to quickly spread to the newly arrived agents, such that the progression of learning is minimally disturbed, without needing to rely on the assumption of a centralised learner. The fact that, in all cases, the return prior to the population increase at $k = 300$ is lower than in Fig. 2, is reflective of the fact that the error in the solution reduces as $N$ tends to infinity.

In our future work with non-stationary equilibria, where agents' policies will also depend on the population distribution, it may be a strong assumption to suppose that decentralised agents with local state observations and limited communication radius would be able to observe the entire population distribution. We will therefore explore a framework of networked agents estimating the empirical distribution from only their local neighbourhood as in (Ganapathi Subramanian et al., 2021), and possibly also improving this estimation by communicating with neighbours (Yongacoglu et al., 2024), such that this useful information spreads through the network along with policy parameters.

Our algorithm for the networked case (Alg. 1), as well as prior work on the centralised and independent cases (Yardim et al., 2023), all have multiple nested loops. This is a potential limitation for real-world

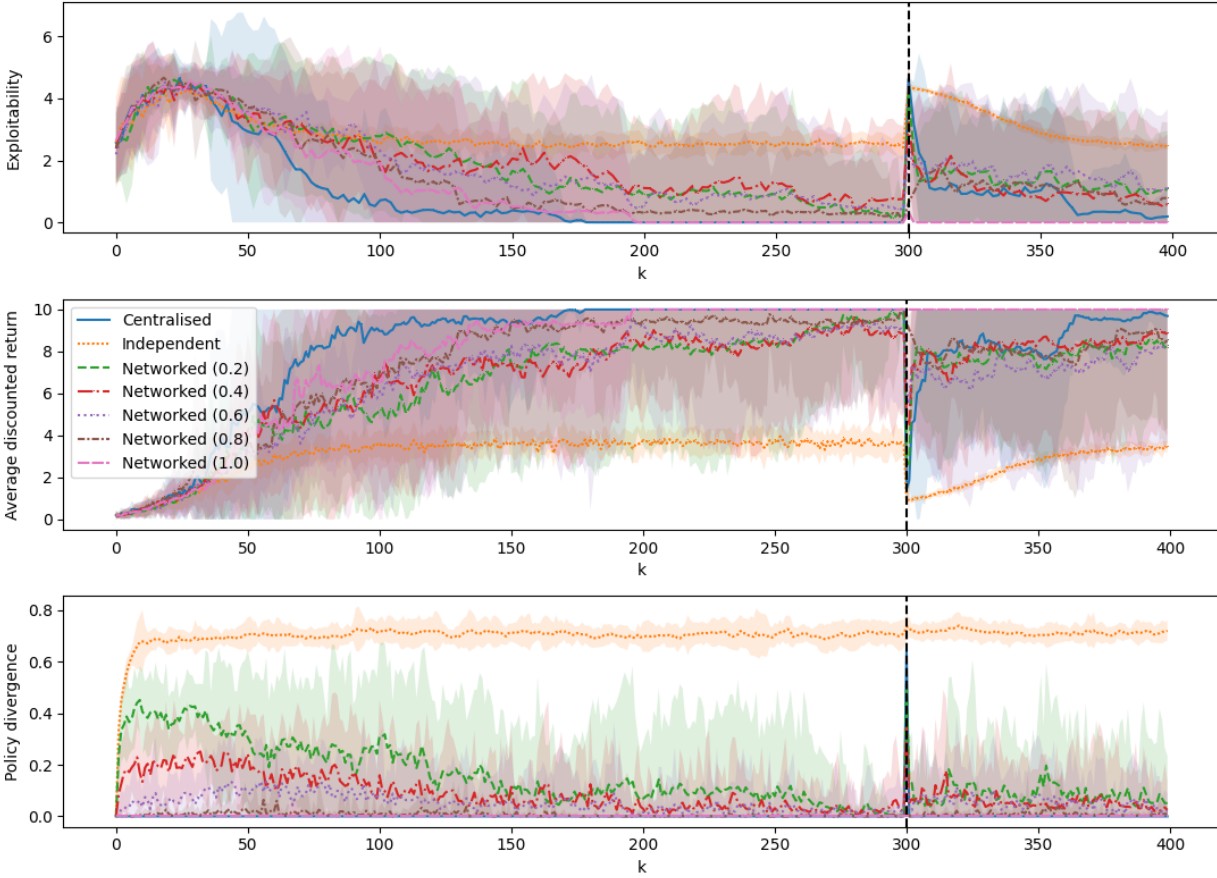

Figure 7: 'Target agreement' game, testing robustness to a five-times increase in population. The networked architectures are quickly able to spread the learnt policies to the newly arrived agents such that learning progress is minimally disturbed, whereas convergence is significantly impacted in the independent case. The largest broadcast radius (1.0, pink), in particular, suffers no disturbance at all, being more robust than the centralised case, which takes a significant amount of time to return to equilibrium.

implementation, since the decentralised agents might be sensitive to failures in synchronising these loops. However, in practice, we show that our networked architecture provides redundancy and robustness (which the independent-learning algorithm lacks) in case of learning failures that may result from the necessities of synchronisation (see Sec. 7.4.2). We have also shown that networked communication in combination with the replay buffer allows us to reduce the hyperparameter $M_{td}$ to 1, essentially removing the inner 'waiting' loop. Nevertheless, our algorithm still features multiple loops, and future work lies in simplifying the algorithms further to aid practical implementation, possibly by techniques such as asynchronous communication (Ma et al., 2024). Future works should also consider updating our theoretical guarantees in light of our current practical algorithmic enhancements, as well as any future modifications.

Since the MFG setting is technically non-cooperative, we have preempted objections that agents would not have incentive to communicate their policies by focusing on coordination games, i.e. where agents seek to maximise only their individual returns, but receive higher rewards when they follow the same strategy as other agents. In this case they stand to benefit by exchanging their policies with others. Future work lies in extending our networked communication algorithms to mean-field control, the cooperative counterpart to MFGs, where agents would have incentive to communicate across different types of game. Nevertheless, in real-world settings, the communication network could still be vulnerable to malfunctioning agents or

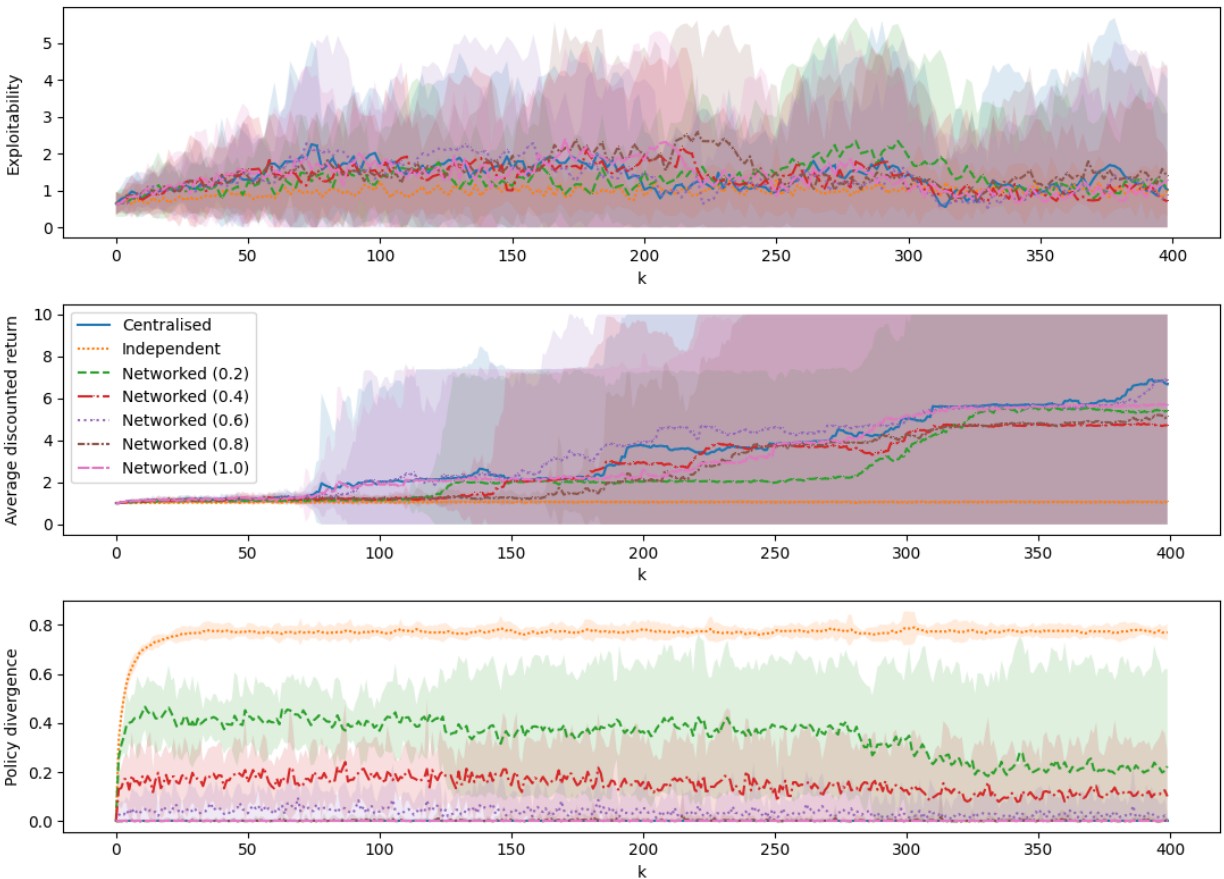

Figure 8: 'Cluster' game on the larger 16x16 grid. While the independent-learning case has similar exploitability to the other settings, we can see that it is not actually learning to increase its return at all, making this an undesirable equilibrium. (I.e. agents are moving about randomly so there is little a deviating agent can do to increase its reward, hence exploitability is low even though the agents are not in fact clustered - see Rem. 7.2.) All the networked settings perform similarly to the centralised case and outperform the return of the independent agents.

adversarial actors poisoning the equilibrium by broadcasting untrue policy information (Agrawal et al., 2024). It is outside the scope of this paper to analyse how much false information would have to be broadcast by how many agents to affect the equilibrium, but real-world applications may need to compute this and prevent it. Future research to mitigate this risk might build on work such as Piazza et al. (2024), where 'power regularisation' of information flow is proposed to limit the adverse effects of communication by misaligned agents.

While our MFG *algorithms* are designed to handle arbitrarily large numbers of agents (and theoretically perform better as $N \to \infty$), the *code* for our experiments naturally still suffers from a bottleneck of computational speed when simulating agents that in the real world would be acting and learning in parallel, since the GPU can only process JAX-vectorised elements in batches of a certain size.

**Broader Impact Statement**

We identified no specific ethical concerns regarding our work, which explores new game theoretical and machine learning algorithms in general settings.

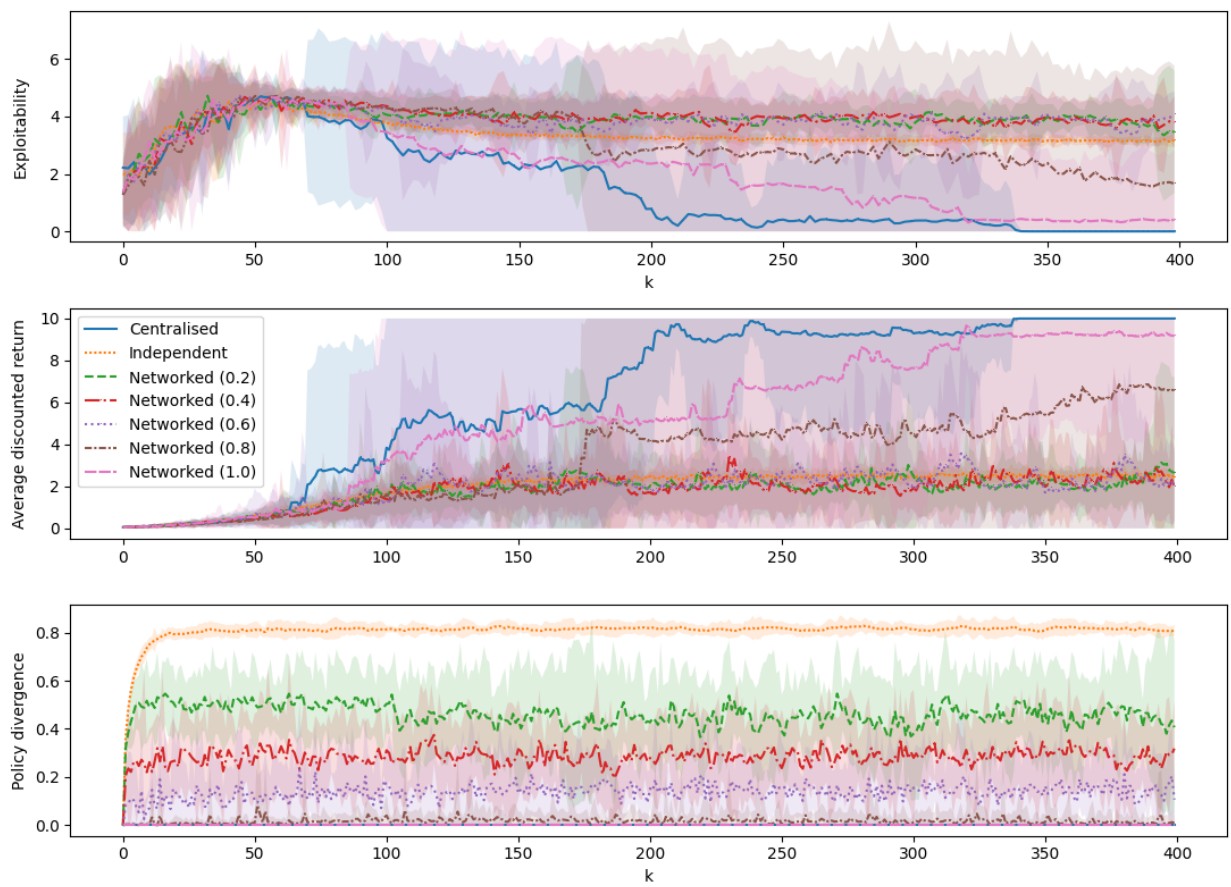

Figure 9: 'Target agreement' game on the larger 16x16 grid. There is greater differentiation in this setting than in the 8x8 grid (Fig. 3) between the different broadcast radii in the networked cases, as might be expected in a less densely populated environment. The two largest broadcast radii (1.0, pink, and 0.8, brown), which have the most connected networks, outperform the independent case in terms of both exploitability and return. However, the other broadcast radii perform similarly to the independent case.

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

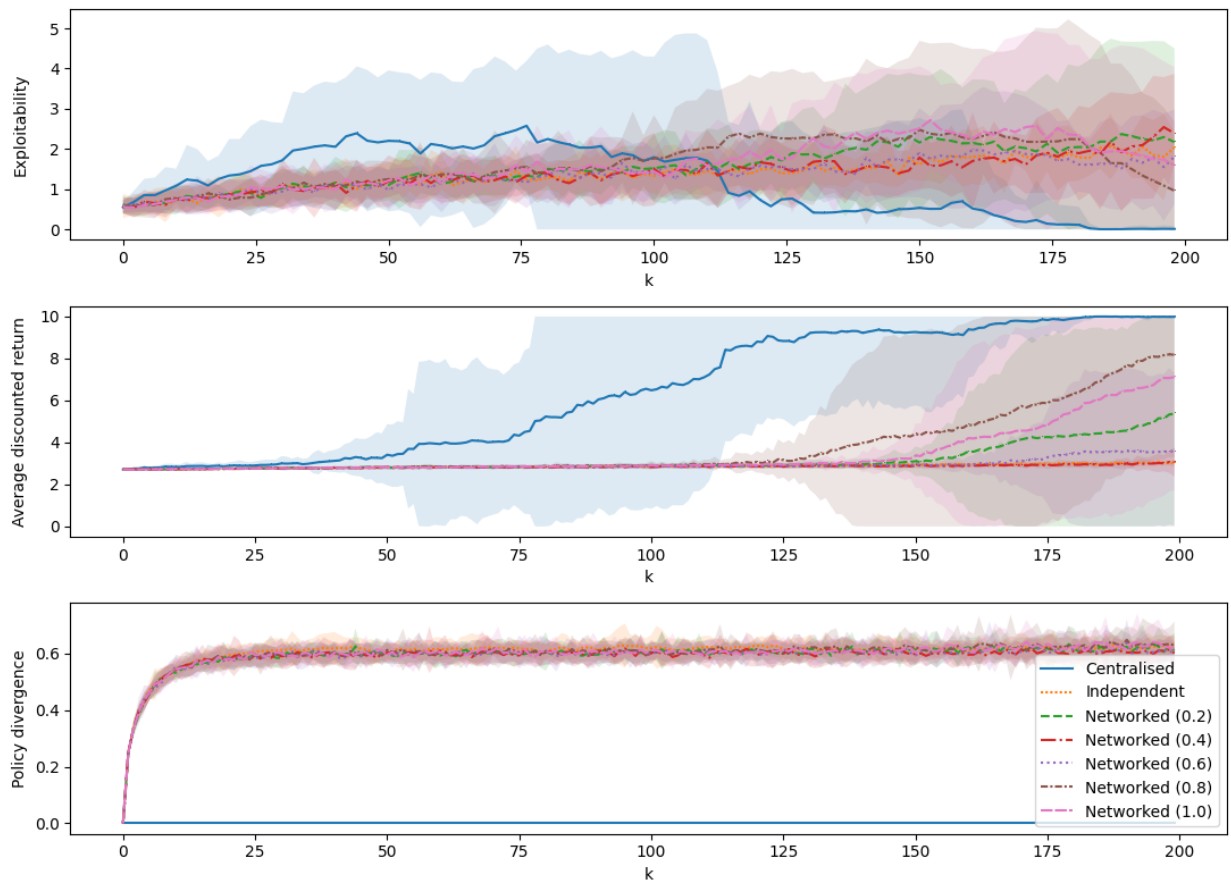

Figure 10: 'Cluster' game with $\tau_k$ fixed as 100 for all $k$; compare this to Fig. 2 where $\tau_k$ is annealed. Without the annealing scheme, the networked architecture appears to perform similarly to the independent case in terms of exploitability, but several broadcast radii outperform the independent case in terms of return, demonstrating that our networked algorithm can still help agents find 'preferable' equilibria. However, whereas with annealing the networked architecture converges similarly to the centralised case, here it performs less well.

Bernard T. Agyeman, Benjamin Decardi-Nelson, Jinfeng Liu, and Sirish L. Shah. A semi-centralized multi-agent RL framework for efficient irrigation scheduling. *Control Engineering Practice*, 155:106183, 2025. ISSN 0967-0661. doi: https://doi.org/10.1016/j.conengprac.2024.106183. URL https://www.sciencedirect.com/science/article/pii/S0967066124003423.

Anna Aksamit, Kaustav Das, Ivan Guo, Kihun Nam, and Zhou Zhou. Switching to a Green and sustainable finance setting: a mean field game approach. *arXiv preprint arXiv:2503.06967*, 2025.

Talal Algumaei, Ruben Solozabal, Reda Alami, Hakim Hacid, Merouane Debbah, and Martin Takáč. Regularization of the policy updates for stabilizing Mean Field Games. In *Pacific-Asia Conference on Knowledge Discovery and Data Mining*, pp. 361–372. Springer, 2023.

Berkay Anahtarci, Can Deha Karıksız, and Naci Saldi. Fitted Q-Learning in Mean-field Games. *ArXiv*, abs/1912.13309, 2019.

Berkay Anahtarci, Can Deha Kariksiz, and Naci Saldi. Q-learning in regularized mean-field games. *Dynamic Games and Applications*, 13(1):89–117, 2023.

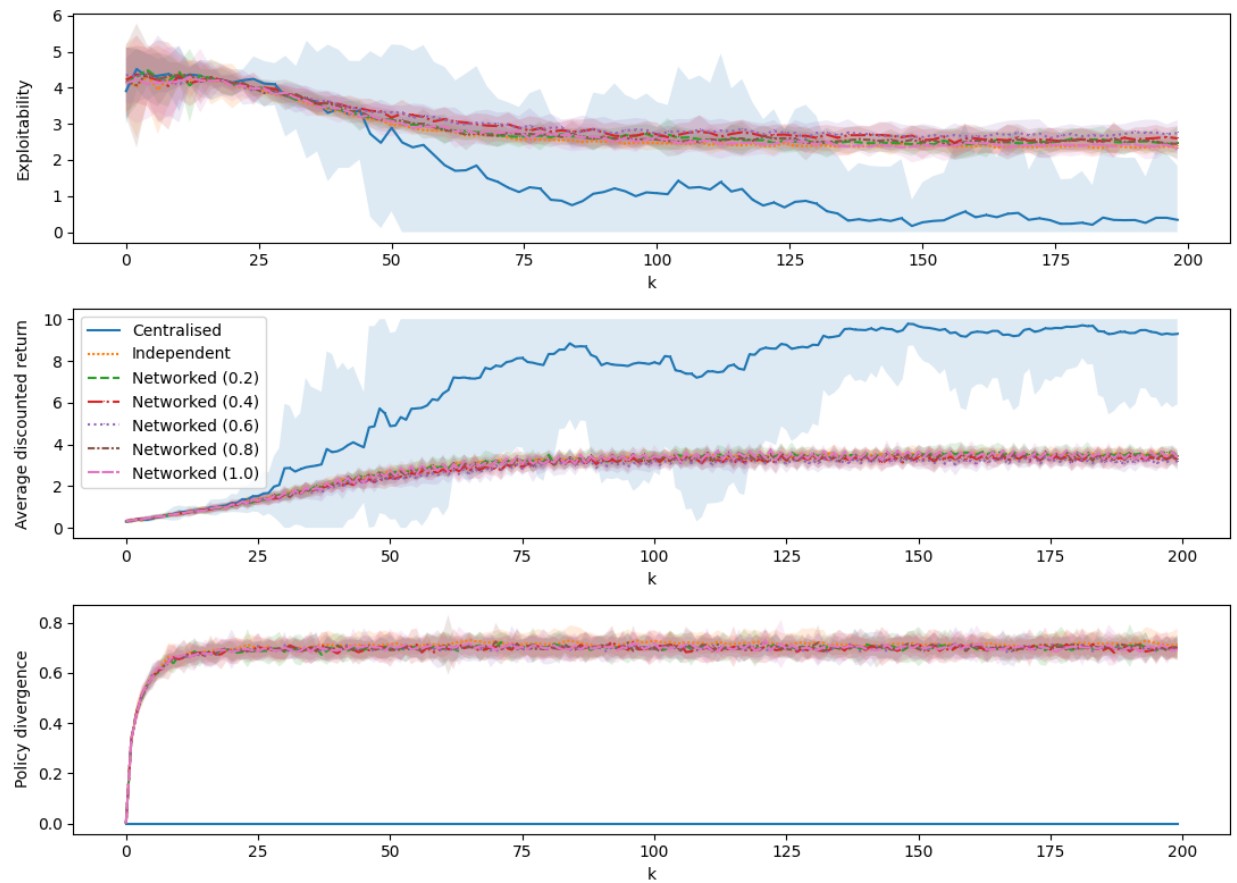

Figure 11: 'Target agreement' game with $\tau_k$ fixed as 100 for all $k$. Without our annealing scheme for the softmax temperature, the networked architecture does not outperform the independent case. Compare this to Fig. 3 which shows the benefit of annealing $\tau_k$.

David Andréen, Petra Jenning, Nils Napp, and Kirstin Petersen. Emergent structures assembled by large swarms of simple robots. In *Acadia*, pp. 54–61, 2016.

Andrea Angiuli, Jean-Pierre Fouque, and Mathieu Laurière. Unified reinforcement Q-learning for mean field game and control problems. *Mathematics of Control, Signals, and Systems*, 34(2):217–271, 2022.

Andrea Angiuli, Jean-Pierre Fouque, Mathieu Laurière, and Mengrui Zhang. Convergence of Multi-Scale Reinforcement Q-Learning Algorithms for Mean Field Game and Control Problems. *arXiv preprint arXiv:2312.06659*, 2023.

Burak Aydin, Emre Parmaksiz, and Ronnie Sircar. Fare Game: A Mean Field Model of Stochastic Intensity Control in Dynamic Ticket Pricing. *arXiv preprint arXiv:2506.13088*, 2025.

Uğur Aydın and Naci Saldi. Robustness and Approximation of Discrete-time Mean-field Games under Discounted Cost Criterion. *arXiv preprint arXiv:2310.10828*, 2023.

Yu Bai, Di Zhou, and Zhen He. Optimal Pursuit Strategies in Missile Interception: Mean Field Game Approach. *Aerospace*, 12(4), 2025. ISSN 2226-4310. doi: 10.3390/aerospace12040302. URL https://www.mdpi.com/2226-4310/12/4/302.

Julian Barreiro-Gomez and Shinkyu Park. Optimal Strategy Revision in Population Games: A Mean Field Game Theory Perspective. *arXiv preprint arXiv:2501.01389*, 2025.

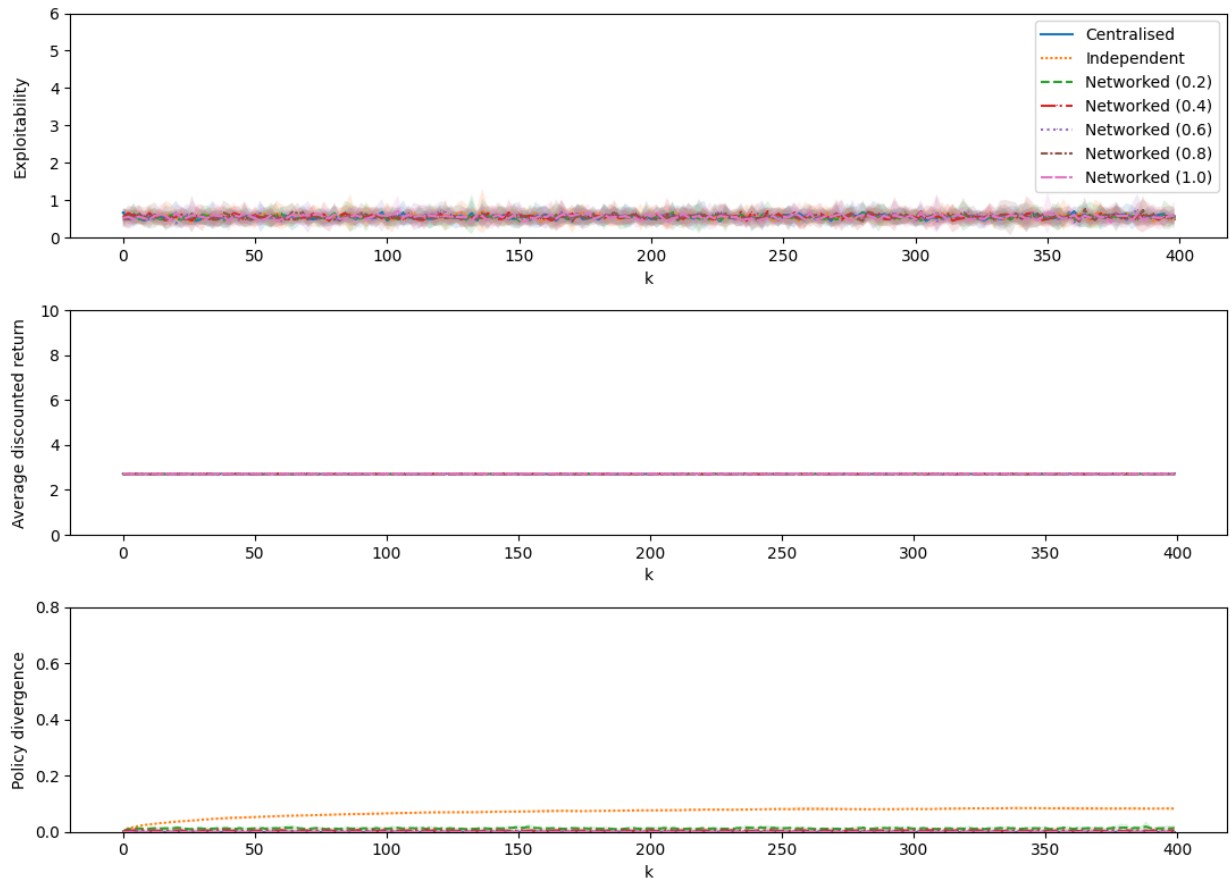

Figure 12: 'Cluster' game with our experience replay buffer removed. There is no noticeable improvement in any of the agents' returns, i.e. no noticeable learning, even after $K = 400$ iterations.

Dario Bauso and Hamidou Tembine. Crowd-Averse Cyber-Physical Systems: The Paradigm of Robust Mean-Field Games. *IEEE Transactions on Automatic Control*, 61(8):2312–2317, 2016. doi: 10.1109/TAC.2015. 2492038.

Dario Bauso, Hamidou Tembine, and Tamer Başar. Robust Mean Field Games with Application to Production of an Exhaustible Resource. *IFAC Proceedings Volumes*, 45(13):454–459, 2012. ISSN 1474-6670. doi: https://doi.org/10.3182/20120620-3-DK-2025.00135. URL https://www.sciencedirect.com/science/article/pii/S1474667015377302. 7th IFAC Symposium on Robust Control Design.

Dario Bauso, Hamidou Tembine, and Tamer Başar. Robust mean field games. *Dynamic games and applications*, 6(3):277–303, 2016.

Erhan Bayraktar and Ali D Kara. Learning with Linear Function Approximations in Mean-Field Control. *arXiv preprint arXiv:2408.00991*, 2024.

Dirk Becherer and Stefanie Hesse. Common Noise by Random Measures: Mean-Field Equilibria for Competitive Investment and Hedging. *arXiv preprint arXiv:2408.01175*, 2024.

Amani Benamor, Oussama Habachi, Inès Kammoun, and Jean-Pierre Cances. NOMA-based Power Control for Machine-Type Communications: A Mean Field Game Approach. In *2022 IEEE International Performance, Computing, and Communications Conference (IPCCC)*, pp. 338–343, 2022. doi: 10.1109/IPCCC55026.2022.9894296.

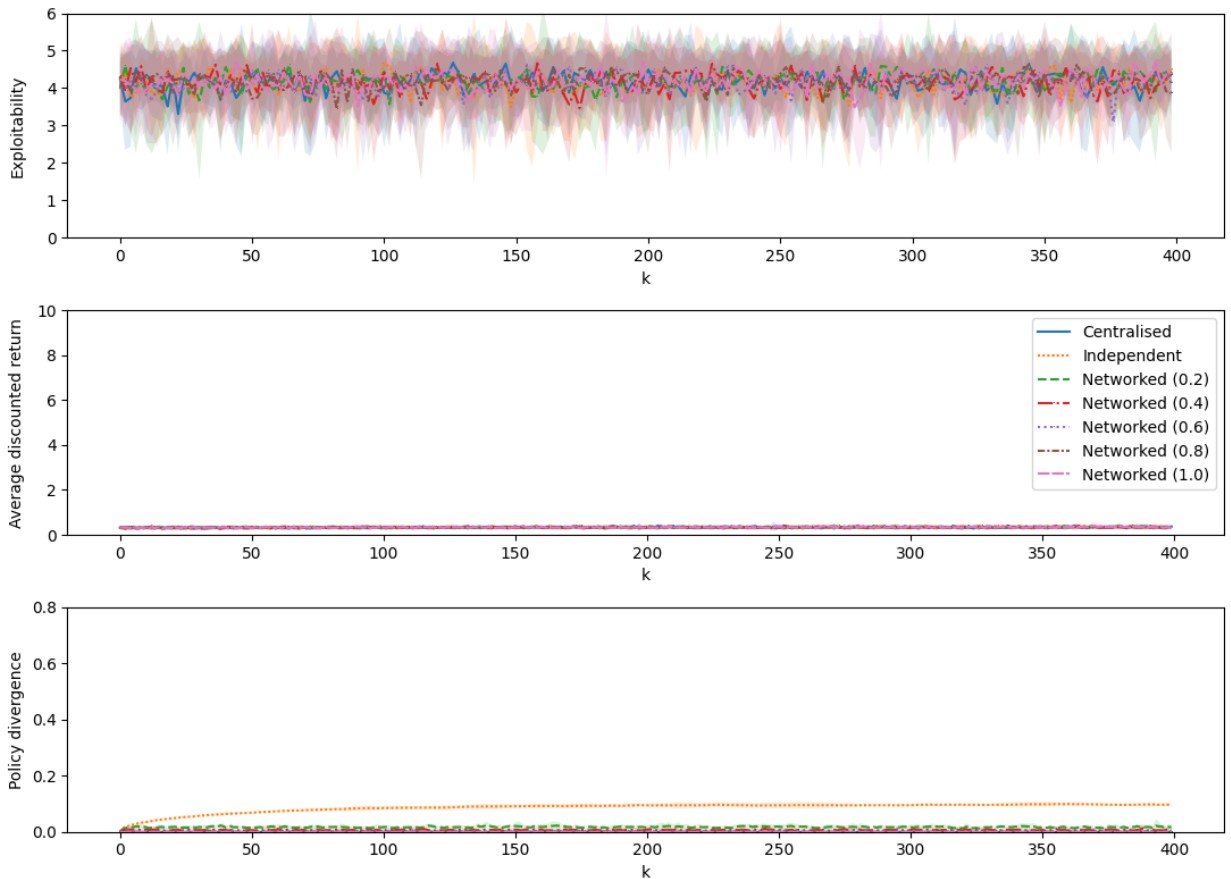

Figure 13: 'Target agreement' game with our experience replay buffer removed. There is no noticeable improvement in any of the agents' returns, i.e. no noticeable learning, even after $K = 400$ iterations.

Martino Bernasconi, E. Vittori, F. Trovò, and M. Restelli. Dealer markets: A reinforcement learning mean field game approach. *The North American Journal of Economics and Finance*, 68:101974, 2023. ISSN 1062-9408. doi: https://doi.org/10.1016/j.najef.2023.101974. URL https://www.sciencedirect.com/science/article/pii/S1062940823000979.

Rico Berner, Thilo Gross, Christian Kuehn, Jürgen Kurths, and Serhiy Yanchuk. Adaptive dynamical networks. *Physics Reports*, 1031:1–59, 2023.

Lijun Bo, Yijie Huang, and Xiang Yu. Mean Field Game of Optimal Tracking Portfolio. *arXiv preprint arXiv:2505.01858*, 2025.

Luis Briceño-Arias, Dante Kalise, and Francisco Silva. Proximal methods for stationary Mean Field Games with local couplings. *SIAM Journal on Control and Optimization*, 56:801–, 03 2018.

Cacace, Simone, Camilli, Fabio, and Goffi, Alessandro. A policy iteration method for mean field games. *ESAIM: COCV*, 27:85, 2021. doi: 10.1051/cocv/2021081. URL https://doi.org/10.1051/cocv/2021081.

Nicolas Cambier, Vincent Frémont, Vito Trianni, and Eliseo Ferrante. Embodied evolution of self-organised aggregation by cultural propagation. In Marco Dorigo, Mauro Birattari, Christian Blum, Anders L. Christensen, Andreagiovanni Reina, and Vito Trianni (eds.), *Swarm Intelligence*, pp. 351–359, Cham, 2018. Springer International Publishing.

Nicolas Cambier, Roman Miletitch, Vincent Fremont, Marco Dorigo, Eliseo Ferrante, and Vito Trianni. Language Evolution in Swarm Robotics: A Perspective. *Frontiers in Robotics and AI*, 7, 2020. ISSN 2296-9144. doi: 10.3389/frobt.2020.00012. URL `https://www.frontiersin.org/articles/10.3389/frobt.2020.00012`.

Nicolas Cambier, Dario Albani, Vincent Fremont, Vito Trianni, and Eliseo Ferrante. Cultural evolution of probabilistic aggregation in synthetic swarms. *Applied Soft Computing*, 113:108010, 2021. ISSN 1568-4946. doi: https://doi.org/10.1016/j.asoc.2021.108010. URL `https://www.sciencedirect.com/science/article/pii/S1568494621009327`.

Haoyang Cao, Xin Guo, and Mathieu Laurière. Connecting GANs, MFGs, and OT. *arXiv preprint arXiv:2002.04112*, 2020.

Zhongyuan Cao and Mathieu Laurière. Probabilistic Analysis of Graphon Mean Field Control. *arXiv preprint arXiv:2505.19664*, 2025.

Cardaliaguet, Pierre and Hadikhanloo, Saeed. Learning in mean field games: The fictitious play. *ESAIM: COCV*, 23(2):569–591, 2017. doi: 10.1051/cocv/2016004. URL `https://doi.org/10.1051/cocv/2016004`.

E. Carlini and F. J. Silva. A Fully Discrete Semi-Lagrangian Scheme for a First Order Mean Field Game Problem. *SIAM Journal on Numerical Analysis*, 52(1):45–67, 2014. doi: 10.1137/120902987. URL `https://doi.org/10.1137/120902987`.

Elisabetta Carlini and Valentina Coscetti. A semi-Lagrangian scheme for First-Order Mean Field Games based on monotone operators. *arXiv preprint arXiv:2506.10509*, 2025.

René Carmona and Mathieu Laurière. Deep learning for mean field games and mean field control with applications to finance. *arXiv preprint arXiv:2107.04568*, 7, 2021.

René Carmona, Mathieu Laurière, and Zongjun Tan. Model-Free Mean-Field Reinforcement Learning: Mean-Field MDP and Mean-Field Q-Learning. *The Annals of Applied Probability*, 33(6B):5334–5381, 2023.

Alekos Cecchin, Markus Fischer, Claudio Fontana, and Giacomo Lanaro. Weak equilibria of a mean-field market model under asymmetric information. *arXiv preprint arXiv:2504.09356*, 2025.

Lu Chang, Liang Shan, Weilong Zhang, and Yuewei Dai. Hierarchical multi-robot navigation and formation in unknown environments via deep reinforcement learning and distributed optimization. *Robotics and Computer-Integrated Manufacturing*, 83:102570, 2023. ISSN 0736-5845. doi: https://doi.org/10.1016/j.rcim.2023.102570. URL `https://www.sciencedirect.com/science/article/pii/S0736584523000467`.

Fan Chen, Nicholas Martin, Po-Yu Chen, Xiaozhen Wang, Zhenjie Ren, and Francois Buet-Golfouse. Deciding Bank Interest Rates–A Major-Minor Impulse Control Mean-Field Game Perspective. *arXiv preprint arXiv:2411.14481*, 2024a.

Mingzhe Chen, Deniz Gündüz, Kaibin Huang, Walid Saad, Mehdi Bennis, Aneta Vulgarakis Feljan, and H Vincent Poor. Distributed Learning in Wireless Networks: Recent Progress and Future Challenges. *IEEE Journal on Selected Areas in Communications*, 39(12):3579–3605, 2021.

Tian Chen, Kai Du, and Zhen Wu. Partially observed mean-field game and related mean-field forward-backward stochastic differential equation. *Journal of Differential Equations*, 408:409–448, 2024b. ISSN 0022-0396. doi: https://doi.org/10.1016/j.jde.2024.07.014. URL `https://www.sciencedirect.com/science/article/pii/S0022039624004364`.

Tian Chen, Hongyu Shi, and Zhen Wu. A Progressive Maximum Principle of Fully Coupled Mean-Field System with Jumps. *Journal of Optimization Theory and Applications*, 206(3):75, 2025. doi: 10.1007/s10957-025-02760-y. URL `https://doi.org/10.1007/s10957-025-02760-y`.

Xu Chen, Shuo Liu, and Xuan Di. A Hybrid Framework of Reinforcement Learning and Physics-Informed Deep Learning for Spatiotemporal Mean Field Games. In *Proceedings of the 2023 International Conference on Autonomous Agents and Multiagent Systems*, AAMAS '23, pp. 1079–1087, Richland, SC, 2023. International Foundation for Autonomous Agents and Multiagent Systems. ISBN 9781450394321.

Xu Chen, Shuo Liu, and Xuan Di. Bridging Agent Dynamics and Population Behaviors: Scalable Learning for Mean Field Games on Graph via Neural Operators. AAAI, 2024c.

Yan Chen, Tao Li, and Nan Qiao. Sampled-data based adaptive mean field games for leader-follower stochastic multi-agent systems. *Mathematical Control and Related Fields*, pp. 0–0, 2024d.

Yufan Chen, Lan Wu, Renyuan Xu, and Ruixun Zhang. Periodic Trading Activities in Financial Markets: Mean-field Liquidation Game with Major-Minor Players. *arXiv preprint arXiv:2408.09505*, 2024e.

Salah Eddine Choutri, Boualem Djehiche, Prajwal Chauhan, and Saif Eddin Jabari. Backpressure-based Mean-field Type Game for Scheduling in Multi-Hop Wireless Sensor Networks. *arXiv preprint arXiv:2506.03059*, 2025.

Dantong Chu, Kenneth Tsz Hin Ng, Sheung Chi Phillip Yam, and Harry Zheng. Mean field analysis of two-party governance: Competition versus cooperation among leaders. *Automatica*, 173:112028, 2025.

Kai Cui and Heinz Koeppl. Approximately Solving Mean Field Games via Entropy-Regularized Deep Reinforcement Learning. In *International Conference on Artificial Intelligence and Statistics*, pp. 1909–1917. PMLR, 2021.

Kai Cui, Anam Tahir, Gizem Ekinci, Ahmed Elshamanhory, Yannick Eich, Mengguang Li, and Heinz Koeppl. A Survey on Large-Population Systems and Scalable Multi-Agent Reinforcement Learning. *arXiv preprint arXiv:2209.03859*, 2022.

Kai Cui, Christian Fabian, and Heinz Koeppl. Multi-Agent Reinforcement Learning via Mean Field Control: Common Noise, Major Agents and Approximation Properties. *arXiv preprint arXiv:2303.10665*, 2023a.

Kai Cui, Sascha Hauck, Christian Fabian, and Heinz Koeppl. Learning Decentralized Partially Observable Mean Field Control for Artificial Collective Behavior. *arXiv preprint arXiv:2307.06175*, 2023b.

Kai Cui, Gökçe Dayanıklı, Mathieu Laurière, Matthieu Geist, Olivier Pietquin, and Heinz Koeppl. Learning Discrete-Time Major-Minor Mean Field Games. In *Proceedings of the AAAI Conference on Artificial Intelligence*, volume 38, pp. 9616–9625, 2024.

Constantinos Daskalakis, Paul W. Goldberg, and Christos H. Papadimitriou. The Complexity of Computing a Nash Equilibrium. In *Proceedings of the Thirty-Eighth Annual ACM Symposium on Theory of Computing*, STOC '06, pp. 71–78, New York, NY, USA, 2006. Association for Computing Machinery. ISBN 1595931341. doi: 10.1145/1132516.1132527. URL https://doi.org/10.1145/1132516.1132527.

Murad Dawood, Sicong Pan, Nils Dengler, Siqi Zhou, Angela P Schoellig, and Maren Bennewitz. Safe Multi-Agent Reinforcement Learning for Formation Control without Individual Reference Targets. *arXiv preprint arXiv:2312.12861*, 2023.

Gokce Dayanikli and Mathieu Lauriere. Cooperation, competition, and common pool resources in mean field games. *arXiv preprint arXiv:2504.09043*, 2025.

Gökçe Dayanikli, Mathieu Laurière, and Jiacheng Zhang. Deep Learning for Population-Dependent Controls in Mean Field Control Problems with Common Noise. In *Proceedings of the 23rd International Conference on Autonomous Agents and Multiagent Systems*, pp. 2231–2233, 2024.

Gianmarco Del Sarto, Marta Leocata, and Giulia Livieri. A Mean Field Game approach for pollution regulation of competitive firms. *arXiv preprint arXiv:2407.12754*, 2024.

Shawon Dey and Hao Xu. Intelligent Distributed Charging Control for Large Scale Electric Vehicles: A Multi-Cluster Mean Field Game Approach. In *Proceedings of Cyber-Physical Systems and Internet of Things Week 2023*, CPS-IoT Week '23, pp. 146–151, New York, NY, USA, 2023. Association for Computing Machinery. ISBN 9798400700491. doi: 10.1145/3576914.3587709. URL https://doi.org/10.1145/3576914.3587709.

Shawon Dey and Hao Xu. Extended mean field game theoretical optimal distributed control for large scale multi-agent systems: An efficiency-complexity tradeoff. *Information Sciences*, 719:122432, 2025. ISSN 0020-0255. doi: https://doi.org/10.1016/j.ins.2025.122432. URL https://www.sciencedirect.com/science/article/pii/S002002552500564X.

Adam Eck, Leen-Kiat Soh, and Prashant Doshi. Decision making in open agent systems. *AI Mag.*, 44(4):508–523, dec 2023. ISSN 0738-4602. doi: 10.1002/aaai.12131. URL https://doi.org/10.1002/aaai.12131.

A. E. Eiben and J. E. Smith. *What Is an Evolutionary Algorithm?*, pp. 25–48. Springer Berlin Heidelberg, Berlin, Heidelberg, 2015. ISBN 978-3-662-44874-8. doi: 10.1007/978-3-662-44874-8_3. URL https://doi.org/10.1007/978-3-662-44874-8_3.

Romuald Elie, Julien Pérolat, Mathieu Laurière, Matthieu Geist, and Olivier Pietquin. On the Convergence of Model Free Learning in Mean Field Games. *Proceedings of the AAAI Conference on Artificial Intelligence*, 34(05):7143–7150, Apr. 2020. doi: 10.1609/aaai.v34i05.6203. URL https://ojs.aaai.org/index.php/AAAI/article/view/6203.

Yousef Emami, Hao Gao, Kai Li, Luis Almeida, Eduardo Tovar, and Zhu Han. Age of Information Minimization using Multi-agent UAVs based on AI-Enhanced Mean Field Resource Allocation. *IEEE Transactions on Vehicular Technology*, pp. 1–14, 2024. doi: 10.1109/TVT.2024.3394235.

Olav Ersland, Espen Robstad Jakobsen, and Alessio Porretta. Long time behaviour of Mean Field Games with fractional diffusion. *arXiv preprint arXiv:2505.06183*, 2025.

Salvatore Federico, Fausto Gozzi, and Andrzej Święch. On Mean Field Games in Infinite Dimension. *arXiv preprint arXiv:2411.14604*, 2024.

F. A. Fedorov. Studying the well-posedness of the boundary value problem for a system of riccati type equations based on the concept of mean field games. *Moscow University Computational Mathematics and Cybernetics*, 49(2):150–164, 2025. doi: 10.3103/S0278641925700086. URL https://doi.org/10.3103/S0278641925700086.

William Fedus, Prajit Ramachandran, Rishabh Agarwal, Yoshua Bengio, Hugo Larochelle, Mark Rowland, and Will Dabney. Revisiting Fundamentals of Experience Replay. In *Proceedings of the 37th International Conference on Machine Learning*, ICML'20. JMLR.org, 2020.

Chen Feng and Andrew L Liu. Decentralized Integration of Grid Edge Resources into Wholesale Electricity Markets via Mean-field Games. *arXiv preprint arXiv:2503.07984*, 2025.

Iñaki Fernández Pérez and Stéphane Sanchez. Influence of Local Selection and Robot Swarm Density on the Distributed Evolution of GRNs. In Paul Kaufmann and Pedro A. Castillo (eds.), *Applications of Evolutionary Computation*, pp. 567–582, Cham, 2019. Springer International Publishing. ISBN 978-3-030-16692-2.

Iñaki Fernández Pérez, Amine Boumaza, and François Charpillet. Maintaining Diversity in Robot Swarms with Distributed Embodied Evolution. In Marco Dorigo, Mauro Birattari, Christian Blum, Anders L. Christensen, Andreagiovanni Reina, and Vito Trianni (eds.), *Swarm Intelligence*, pp. 395–402, Cham, 2018. Springer International Publishing. ISBN 978-3-030-00533-7.

Rita Ferreira, Diogo Gomes, and Melih Ucer. Solving Mean-Field Games with Monotonicity Methods in Banach Spaces. *arXiv preprint arXiv:2506.21212*, 2025.

Jean-Pierre Fouque and Zhaoyu Zhang. Deep Learning Methods for Mean Field Control Problems With Delay. *Frontiers in Applied Mathematics and Statistics*, 6, 2020. ISSN 2297-4687. doi: 10.3389/fams.2020.00011. URL https://www.frontiersin.org/articles/10.3389/fams.2020.00011.

J Frédéric Bonnans, Pierre Lavigne, and Laurent Pfeiffer. Generalized conditional gradient and learning in potential mean field games. *arXiv e-prints*, pp. arXiv–2109, 2021.

Guanxing Fu and Ulrich Horst. Mean Field Portfolio Games with Epstein-Zin Preferences. *arXiv preprint arXiv:2505.07231*, 2025.

Zuyue Fu, Zhuoran Yang, Yongxin Chen, and Zhaoran Wang. Actor-critic provably finds Nash equilibria of linear-quadratic mean-field games. *arXiv preprint arXiv:1910.07498*, 2019.

Sriram Ganapathi Subramanian, Pascal Poupart, Matthew E Taylor, and Nidhi Hegde. Multi Type Mean Field Reinforcement Learning. In *Proceedings of the 19th International Conference on Autonomous Agents and MultiAgent Systems*, pp. 411–419, 2020.

Sriram Ganapathi Subramanian, Matthew E Taylor, Mark Crowley, and Pascal Poupart. Partially Observable Mean Field Reinforcement Learning. In *Proceedings of the 20th International Conference on Autonomous Agents and MultiAgent Systems*, pp. 537–545, 2021.

Hao Gao, Yongkang Liu, Emrah Akin Sisbot, Yashar Zeinali Farid, Kentaro Oguchi, and Zhu Han. Hierarchical Federated Learning with Mean Field Game Device Selection for Connected Vehicle Applications. In *2023 IEEE Intelligent Vehicles Symposium (IV)*, pp. 1–6, 2023. doi: 10.1109/IV55152.2023.10186687.

Yuzhao Gao, Yiming Nie, and Hongliang Wang. A Scalable Multi-agent Reinforcement Learning Approach Based on Value Function Decomposition. In Yi Qu, Mancang Gu, Yifeng Niu, and Wenxing Fu (eds.), *Proceedings of 3rd 2023 International Conference on Autonomous Unmanned Systems (3rd ICAUS 2023)*, pp. 88–96, Singapore, 2024. Springer Nature Singapore. ISBN 978-981-97-1087-4.

Nicolas Garcia, Ronnie Sircar, and H Mete Soner. Mean Field Games of Control and Cryptocurrency Mining. *arXiv preprint arXiv:2504.15526*, 2025.

Matthieu Geist, Julien Pérolat, Mathieu Laurière, Romuald Elie, Sarah Perrin, Olivier Bachem, Rémi Munos, and Olivier Pietquin. Concave Utility Reinforcement Learning: the Mean-Field Game Viewpoint. *arXiv preprint arXiv:2106.03787*, 2021.

Maximilien Germain, Joseph Mikael, and Xavier Warin. Numerical resolution of McKean-Vlasov FBSDEs using neural networks. *Methodology and Computing in Applied Probability*, 24(4):2557–2586, 2022.

Ramen Ghosh. Federated Mean-Field Learning with Fairness Constraints: An Optimal Transport Game-Theoretic Approach. 2025a.

Ramen Ghosh. Mean-Field Games for Coordinated Exploration in Dynamic Environments. 2025b.

A. I. Glukhov, M. A. Shishlenin, and N. V. Trusov. Modelling the Dynamics of Social Protests: Mean-Field Games and Inverse Problems. *Differencial'nye uravneniya*, 61(6):802–822, 2025. ISSN 0374-0641. URL https://clinpractice.ru/0374-0641/article/view/685643.

Jorge Gomes and Anders L. Christensen. Generic Behaviour Similarity Measures for Evolutionary Swarm Robotics. In *Proceedings of the 15th Annual Conference on Genetic and Evolutionary Computation*, GECCO '13, pp. 199–206, New York, NY, USA, 2013. Association for Computing Machinery. ISBN 9781450319638. doi: 10.1145/2463372.2463398. URL https://doi.org/10.1145/2463372.2463398.

P Jameson Graber. A "trembling hand perfect" equilibrium for a certain class of mean field games. *arXiv preprint arXiv:2506.11868*, 2025.

Sergio Grammatico, Basilio Gentile, Francesca Parise, and John Lygeros. A Mean Field control approach for demand side management of large populations of Thermostatically Controlled Loads. In *2015 European Control Conference (ECC)*, pp. 3548–3553, 2015a. doi: 10.1109/ECC.2015.7331083.

Sergio Grammatico, Francesca Parise, and John Lygeros. Constrained linear quadratic deterministic mean field control: Decentralized convergence to Nash equilibria in large populations of heterogeneous agents. In *2015 54th IEEE Conference on Decision and Control (CDC)*, pp. 4412–4417, 2015b. doi: 10.1109/CDC.2015.7402908.

Sergio Grammatico, Francesca Parise, Marcello Colombino, and John Lygeros. Decentralized Convergence to Nash Equilibria in Constrained Deterministic Mean Field Control. *IEEE Transactions on Automatic Control*, 61(11):3315–3329, 2016. doi: 10.1109/TAC.2015.2513368.

Luca Grosset and Elena Sartori. Mean-Field Modeling of Green Technology Adoption: A Competition for Incentives. *Mathematics*, 13(5), 2025. ISSN 2227-7390. doi: 10.3390/math13050691. URL https://www.mdpi.com/2227-7390/13/5/691.

Xin Guo, Anran Hu, Renyuan Xu, and Junzi Zhang. Learning Mean-Field Games. In H. Wallach, H. Larochelle, A. Beygelzimer, F. d'Alché-Buc, E. Fox, and R. Garnett (eds.), *Advances in Neural Information Processing Systems*, volume 32. Curran Associates, Inc., 2019a. URL https://proceedings.neurips.cc/paper_files/paper/2019/file/030e65da2b1c944090548d36b244b28d-Paper.pdf.

Xin Guo, Anran Hu, Renyuan Xu, and Junzi Zhang. Learning Mean-Field Games. In H. Wallach, H. Larochelle, A. Beygelzimer, F. d'Alché-Buc, E. Fox, and R. Garnett (eds.), *Advances in Neural Information Processing Systems*, volume 32. Curran Associates, Inc., 2019b. URL https://proceedings.neurips.cc/paper_files/paper/2019/file/030e65da2b1c944090548d36b244b28d-Paper.pdf.

Xin Guo, Renyuan Xu, and Thaleia Zariphopoulou. Entropy Regularization for Mean Field Games with Learning. *Math. Oper. Res.*, 47(4):3239–3260, nov 2022. ISSN 0364-765X. doi: 10.1287/moor.2021.1238. URL https://doi.org/10.1287/moor.2021.1238.

Xin Guo, Anran Hu, Renyuan Xu, and Junzi Zhang. A General Framework for Learning Mean-Field Games. *Mathematics of Operations Research*, 48(2):656–686, 2023.

Evert Haasdijk, Nicolas Bredeche, and Agoston E Eiben. Combining environment-driven adaptation and task-driven optimisation in evolutionary robotics. *PloS one*, 9(6):e98466, 2014.

Emma Hart, Andreas Steyven, and Ben Paechter. Improving Survivability in Environment-Driven Distributed Evolutionary Algorithms through Explicit Relative Fitness and Fitness Proportionate Communication. In *Proceedings of the 2015 Annual Conference on Genetic and Evolutionary Computation*, GECCO '15, pp. 169–176, New York, NY, USA, 2015. Association for Computing Machinery. ISBN 9781450334723. doi: 10.1145/2739480.2754688. URL https://doi.org/10.1145/2739480.2754688.

Jun He and Andrew L Liu. A Hybrid Mean Field Framework for Aggregators Participating in Wholesale Electricity Markets. *arXiv preprint arXiv:2507.03240*, 2025.

Jehad Hedel and Nga Nguyen. Price Coordination for Electric Vehicle Fleet Using Mean Field Game Theory. In *2024 56th North American Power Symposium (NAPS)*, pp. 1–6, 2024. doi: 10.1109/NAPS61145.2024.10741829.

Jehad Hedel and Nga Nguyen. Optimal Charging Control for Electric Vehicle Fleet Using Mean Field Game Theory. In *2025 IEEE Texas Power and Energy Conference (TPEC)*, pp. 1–6, 2025. doi: 10.1109/TPEC63981.2025.10907167.

Felix Höfer, H Mete Soner, and Atilla Yılmaz. Markov Perfect Equilibria in Discrete Finite-Player and Mean-Field Games. *arXiv preprint arXiv:2507.04540*, 2025.

Jiří Horyna, Roland Jung, Stephan Weiss, Eliseo Ferrante, and Martin Saska. Swarming Without an Anchor (SWA): Robot Swarms Adapt Better to Localization Dropouts Then a Single Robot. *IEEE Robotics and Automation Letters*, 10(6):6207–6214, 2025. doi: 10.1109/LRA.2025.3562786.

Anran Hu and Junzi Zhang. MF-OML: Online Mean-Field Reinforcement Learning with Occupation Measures for Large Population Games. *arXiv preprint arXiv:2405.00282*, 2024. URL https://arxiv.org/abs/2405.00282.

Tianfeng Hu, Zhiqun hu, Zhaoming Lu, and Xiangming Wen. Dynamic traffic signal control using mean field multi-agent reinforcement learning in large scale road-networks. *IET Intelligent Transport Systems*, 04 2023. doi: 10.1049/itr2.12364.

Tianjiao Hua and Peng Luo. Extended mean field games with terminal constraint via decoupling fields. *arXiv preprint arXiv:2506.07485*, 2025.

Han Huang and Rongjie Lai. Unsupervised Solution Operator Learning for Mean-Field Games via Sampling-Invariant Parametrizations. *arXiv preprint arXiv:2401.15482*, 2024.

Han Huang and Rongjie Lai. Unsupervised solution operator learning for mean-field games. *Journal of Computational Physics*, 537:114057, 2025. ISSN 0021-9991. doi: https://doi.org/10.1016/j.jcp.2025.114057. URL https://www.sciencedirect.com/science/article/pii/S0021999125003407.

Hui Huang and Jethro Warnett. Well-posedness and mean-field limit estimate of a consensus-based algorithm for multiplayer games. *arXiv preprint arXiv:2505.13632*, 2025.

Jianhui Huang and Minyi Huang. Robust Mean Field Linear-Quadratic-Gaussian Games with Unknown $L^2$-Disturbance. *SIAM Journal on Control and Optimization*, 55(5):2811–2840, 2017. doi: 10.1137/15M1014437. URL https://doi.org/10.1137/15M1014437.

Jiawei Huang, Niao He, and Andreas Krause. Model-Based RL for Mean-Field Games is not Statistically Harder than Single-Agent RL. *arXiv preprint arXiv:2402.05724*, 2024a.

Jiawei Huang, Batuhan Yardim, and Niao He. On the Statistical Efficiency of Mean-Field Reinforcement Learning with General Function Approximation . In Sanjoy Dasgupta, Stephan Mandt, and Yingzhen Li (eds.), *Proceedings of The 27th International Conference on Artificial Intelligence and Statistics*, volume 238 of *Proceedings of Machine Learning Research*, pp. 289–297. PMLR, 02–04 May 2024b. URL https://proceedings.mlr.press/v238/huang24a.html.

Kuang Huang, Xuan Di, Qiang Du, and Xi Chen. A game-theoretic framework for autonomous vehicles velocity control: Bridging microscopic differential games and macroscopic mean field games. *Discrete and Continuous Dynamical Systems - B*, 25(12):4869–4903, 2020. ISSN 1531-3492. doi: 10.3934/dcdsb.2020131.

Minyi Huang, Roland P. Malhamé, and Peter E. Caines. Large population stochastic dynamic games: closed-loop McKean-Vlasov systems and the Nash certainty equivalence principle. *Communications in Information & Systems*, 6(3):221 – 252, 2006.

Daisuke Inoue, Yuji Ito, Takahito Kashiwabara, Norikazu Saito, and Hiroaki Yoshida. Partially Centralized Model-Predictive Mean Field Games for controlling multi-agent systems. *IFAC Journal of Systems and Control*, 24:100217, 2023. ISSN 2468-6018. doi: https://doi.org/10.1016/j.ifacsc.2023.100217. URL https://www.sciencedirect.com/science/article/pii/S2468601823000032.

A. Jadbabaie, Jie Lin, and A.S. Morse. Coordination of groups of mobile autonomous agents using nearest neighbor rules. *IEEE Transactions on Automatic Control*, 48(6):988–1001, 2003. doi: 10.1109/TAC.2003.812781.

Max Jaderberg, Valentin Dalibard, Simon Osindero, Wojciech M Czarnecki, Jeff Donahue, Ali Razavi, Oriol Vinyals, Tim Green, Iain Dunning, Karen Simonyan, et al. Population based training of neural networks. *arXiv preprint arXiv:1711.09846*, 2017.

Bhavini Jeloka, Yue Guan, and Panagiotis Tsiotras. Learning Large-Scale Competitive Team Behaviors with Mean-Field Interactions. *arXiv preprint arXiv:2504.21164*, 2025.

Jiechuan Jiang, Kefan Su, and Zongqing Lu. Fully Decentralized Cooperative Multi-Agent Reinforcement Learning: A Survey. *arXiv preprint arXiv:2401.04934*, 2024.

Yuhan Kang, Hao Gao, and Zhu Han. *Mean Field Game Guided Deep Reinforcement Learning*, pp. 75–90. Springer Nature Switzerland, Cham, 2025a. ISBN 978-3-031-91859-9. doi: 10.1007/978-3-031-91859-9_5. URL https://doi.org/10.1007/978-3-031-91859-9_5.

Yuhan Kang, Hao Gao, and Zhu Han. *Opinion Evolution in Social Networks: Use Generative Adversarial Networks to Solve Mean Field Game*, pp. 29–47. Springer Nature Switzerland, Cham, 2025b. ISBN 978-3-031-91859-9. doi: 10.1007/978-3-031-91859-9_3. URL https://doi.org/10.1007/978-3-031-91859-9_3.

Yuhan Kang, Hao Gao, and Zhu Han. *Incentive Mechanism Design in Satellite-Based Federated Learning Using Mean Field Evolutionary Approach*, pp. 91–114. Springer Nature Switzerland, Cham, 2025c. ISBN 978-3-031-91859-9. doi: 10.1007/978-3-031-91859-9_6. URL https://doi.org/10.1007/978-3-031-91859-9_6.

Marcin Korecki, Damian Dailisan, and Dirk Helbing. How Well Do Reinforcement Learning Approaches Cope With Disruptions? The Case of Traffic Signal Control. *IEEE Access*, 11:36504–36515, 2023. doi: 10.1109/ACCESS.2023.3266644.

Georgios Kotsalis, Guanghui Lan, and Tianjiao Li. Simple and Optimal Methods for Stochastic Variational Inequalities, II: Markovian Noise and Policy Evaluation in Reinforcement Learning. *SIAM Journal on Optimization*, 32(2):1120–1155, 2022. doi: 10.1137/20M1381691. URL https://doi.org/10.1137/20M1381691.

Razvan-Andrei Lascu and Mateusz B Majka. Non-convex entropic mean-field optimization via Best Response flow. *arXiv preprint arXiv:2505.22760*, 2025.

Jean-Michel Lasry and Pierre-Louis Lions. Mean Field Games. *Japanese Journal of Mathematics*, 2(1): 229–260, 2007.

Mathieu Laurière. Numerical Methods for Mean Field Games and Mean Field Type Control. *Mean field games*, 78(221-282), 2021.

Mathieu Laurière, Sarah Perrin, Matthieu Geist, and Olivier Pietquin. Learning Mean Field Games: A Survey. *arXiv preprint arXiv:2205.12944*, 2022a.

Mathieu Laurière, Sarah Perrin, Sertan Girgin, Paul Muller, Ayush Jain, Theophile Cabannes, Georgios Piliouras, Julien Perolat, Romuald Elie, Olivier Pietquin, and Matthieu Geist. Scalable Deep Reinforcement Learning Algorithms for Mean Field Games. In Kamalika Chaudhuri, Stefanie Jegelka, Le Song, Csaba Szepesvari, Gang Niu, and Sivan Sabato (eds.), *Proceedings of the 39th International Conference on Machine Learning*, volume 162 of *Proceedings of Machine Learning Research*, pp. 12078–12095. PMLR, 17–23 Jul 2022b. URL https://proceedings.mlr.press/v162/lauriere22a.html.

Stéphane Le Ménec. Swarm Guidance Based on Mean Field Game Concepts. *International Game Theory Review*, pp. 2440008, 2024.

Kiyeob Lee, Desik Rengarajan, Dileep Kalathil, and Srinivas Shakkottai. Reinforcement Learning for Mean Field Games with Strategic Complementarities . In Arindam Banerjee and Kenji Fukumizu (eds.), *Proceedings of The 24th International Conference on Artificial Intelligence and Statistics*, volume 130 of *Proceedings of Machine Learning Research*, pp. 2458–2466. PMLR, 13–15 Apr 2021. URL https://proceedings.mlr.press/v130/lee21b.html.

Taeyoung Lee et al. Mean Field Game and Control for Switching Hybrid Systems. *arXiv preprint arXiv:2412.10522*, 2024.

Yangqi Lei, Quan Quan, and Zhikun She. Mean-Field-Based Density Control for Swarm Robotics Passing-Through a Virtual Tube. *IEEE Control Systems Letters*, 8:3500–3505, 2024. doi: 10.1109/LCSYS.2025.3550036.

Juan Li, Yanwei Li, and Wenliang Wang. Mean-field backward stochastic differential equations with random terminal time. *Journal of Mathematical Analysis and Applications*, 553(1):129830, 2026. ISSN 0022-247X. doi: https://doi.org/10.1016/j.jmaa.2025.129830. URL https://www.sciencedirect.com/science/article/pii/S0022247X25006110.

Min Li, Tianyang Nie, Shujun Wang, and Ke Yan. Incomplete Information Mean-Field Games and Related Riccati Equations. *Journal of Optimization Theory and Applications*, pp. 1–22, 2024a.

Na Li, Yilin Wei, and Qingfeng Zhu. Stochastic Linear-Quadratic Mean-Field Games of Controls for Delayed Systems with Jump Diffusion. *Journal of Optimization Theory and Applications*, 206(3):66, 2025a. doi: 10.1007/s10957-025-02730-4. URL https://doi.org/10.1007/s10957-025-02730-4.

Yueheng Li, Guangming Xie, and Zongqing Lu. Revisiting Cooperative Off-Policy Multi-Agent Reinforcement Learning. In *Forty-second International Conference on Machine Learning*.

Yunpeng Li, Antonis Dimakis, and Costas A Courcoubetis. On the Effect of Time Preferences on the Price of Anarchy. *arXiv preprint arXiv:2504.20774*, 2025b.

Yunpeng Li, Antonis Dimakis, and Costas A Courcoubetis. Repositioning, Ride-matching, and Abandonment in On-demand Ride-hailing Platforms: A Mean Field Game Approach. *arXiv preprint arXiv:2504.02346*, 2025c.

Zongxi Li, A Max Reppen, and Ronnie Sircar. A Mean Field Games Model for Cryptocurrency Mining. *Management Science*, 70(4):2188–2208, 2024b.

Long-Ji Lin. Self-Improving Reactive Agents Based on Reinforcement Learning, Planning and Teaching. *Mach. Learn.*, 8(3–4):293–321, may 1992. ISSN 0885-6125. doi: 10.1007/BF00992699. URL https://doi.org/10.1007/BF00992699.

Yulong Lu and Pierre Monmarché. Convergence of time-averaged mean field gradient descent dynamics for continuous multi-player zero-sum games. *arXiv preprint arXiv:2505.07642*, 2025.

Zhuangzhuang Ma, Lei Shi, Kai Chen, Jinliang Shao, and Yuhua Cheng. Multi-Agent Bipartite Flocking Control over Cooperation-Competition Networks with Asynchronous Communications. *IEEE Transactions on Signal and Information Processing over Networks*, pp. 1–12, 2024. doi: 10.1109/TSIPN.2024.3384817.

Lorenzo Magnino, Yuchen Zhu, and Mathieu Lauriere. Learning to Stop: Deep Learning for Mean Field Optimal Stopping. In *Forty-second International Conference on Machine Learning*, 2025.

Weichao Mao, Haoran Qiu, Chen Wang, Hubertus Franke, Zbigniew T. Kalbarczyk, Ravishankar K. Iyer, and Tamer Başar. A mean-field game approach to cloud resource management with function approximation. In *Proceedings of the 36th Conference on Advances in Neural Information Processing Systems (NIPS 2022)*, volume 36, pp. 1–12, New Orleans, LA, USA, 2022. Curran Associates, Inc.

E. Everardo Martinez-Garcia, Fernando Luque-Vásquez, and J. Adolfo Minjárez-Sosa. Statistical Estimation of Mean-Field Equilibria in a Class of Discounted Mean-Field Games. *Applied Mathematics & Optimization*, 91(3):73, 2025. doi: 10.1007/s00245-025-10273-3. URL https://doi.org/10.1007/s00245-025-10273-3.

Stephen Mcaleer, JB Lanier, Roy Fox, and Pierre Baldi. Pipeline PSRO: A Scalable Approach for Finding Approximate Nash Equilibria in Large Games. In H. Larochelle, M. Ranzato, R. Hadsell, M.F. Balcan, and H. Lin (eds.), *Advances in Neural Information Processing Systems*, volume 33, pp. 20238–20248. Curran Associates, Inc., 2020. URL https://proceedings.neurips.cc/paper_files/paper/2020/file/e9bcd1b063077573285ae1a41025f5dc-Paper.pdf.

Emily Meigs, Francesca Parise, Asuman E. Ozdaglar, and Daron Acemoglu. Optimal dynamic information provision in traffic routing. *CoRR*, abs/2001.03232, 2020. URL https://arxiv.org/abs/2001.03232.

David Mguni, Joel Jennings, and Enrique Munoz de Cote. Decentralised Learning in Systems With Many, Many Strategic Agents. *Proceedings of the AAAI Conference on Artificial Intelligence*, 32(1), Apr. 2018. doi: 10.1609/aaai.v32i1.11586. URL https://ojs.aaai.org/index.php/AAAI/article/view/11586.

Li Miao, Shuai Li, Xiangjuan Wu, and Bingjie Liu. Mean-Field Stackelberg Game-Based Security Defense and Resource Optimization in Edge Computing. *Applied Sciences*, 14(9), 2024. ISSN 2076-3417. doi: 10.3390/app14093538. URL https://www.mdpi.com/2076-3417/14/9/3538.

Rajesh Mishra, Sriram Vishwanath, and Deepanshu Vasal. Model-free Reinforcement Learning for Mean Field Games. *IEEE Transactions on Control of Network Systems*, pp. 1–11, 2023. doi: 10.1109/TCNS. 2023.3264934.

Rajesh K Mishra, Deepanshu Vasal, and Sriram Vishwanath. Model-free Reinforcement Learning for Non-stationary Mean Field Games. In *2020 59th IEEE Conference on Decision and Control (CDC)*, pp. 1032–1037, 2020. doi: 10.1109/CDC42340.2020.9304340.

Zhaobin Mo, Xu Chen, Xuan Di, Elisa Iacomini, Chiara Segala, Michael Herty, and Mathieu Lauriere. A game-theoretic framework for generic second-order traffic flow models using mean field games and adversarial inverse reinforcement learning. *Transportation Science*, 58(6):1403–1426, 2024.

Benjamin Moll and Lenya Ryzhik. Mean Field Games without Rational Expectations. *arXiv preprint arXiv:2506.11838*, 2025.

Jun Moon and Tamer Başar. Linear Quadratic Risk-Sensitive and Robust Mean Field Games. *IEEE Transactions on Automatic Control*, 62(3):1062–1077, 2017. doi: 10.1109/TAC.2016.2579264.

Behrang Monajemi Nejad, Sid Ahmed Attia, and Jorg Raisch. Max-consensus in a max-plus algebraic setting: The case of fixed communication topologies. In *2009 XXII International Symposium on Information, Communication and Automation Technologies*, pp. 1–7, 2009. doi: 10.1109/ICAT.2009.5348437.

Zijia Niu, Wang Yao, Yuxin Jin, Sanjin Huang, Xiao Zhang, and Langyu Qian. Integrated Task Assignment and Trajectory Planning for a Massive Number of Agents Based on Bilayer-Coupled Mean Field Games. *IEEE Transactions on Automation Science and Engineering*, 22:1833–1852, 2025. doi: 10.1109/TASE. 2024.3370619.

Manfred Opper and Sebastian Reich. On a mean-field Pontryagin minimum principle for stochastic optimal control. *arXiv preprint arXiv:2506.10506*, 2025.

Daniel Jarne Ornia, Pedro J. Zufiria, and Manuel Mazo Jr. Mean Field Behavior of Collaborative Multiagent Foragers. *IEEE Transactions on Robotics*, 38(4):2151–2165, 2022. doi: 10.1109/TRO.2022.3152691.

James Orr and Ayan Dutta. Multi-Agent Deep Reinforcement Learning for Multi-Robot Applications: A Survey. *Sensors*, 23(7), 2023. ISSN 1424-8220. doi: 10.3390/s23073625. URL https://www.mdpi.com/1424-8220/23/7/3625.

Yohance AP Osborne and Iain Smears. Rates of convergence of finite element approximations of second-order mean field games with nondifferentiable Hamiltonians. *arXiv preprint arXiv:2506.03039*, 2025.

Naman Krishna Pande, Arun Kumar, and Arvind Kumar Gupta. Generative adversarial modelling of traffic flow via second-order mean field games with stochastic driving attributes. 2025.

Francesca Parise, Sergio Grammatico, Basilio Gentile, and John Lygeros. Network Aggregative Games and Distributed Mean Field Control via Consensus Theory. *arXiv preprint arXiv:1506.07719*, 2015.

Bhrij Patel, Wesley A Suttle, Alec Koppel, Vaneet Aggarwal, Brian M Sadler, Amrit Singh Bedi, and Dinesh Manocha. Global Optimality without Mixing Time Oracles in Average-reward RL via Multi-level Actor-Critic. *arXiv preprint arXiv:2403.11925*, 2024.

Julien Perolat, Sarah Perrin, Romuald Elie, Mathieu Laurière, Georgios Piliouras, Matthieu Geist, Karl Tuyls, and Olivier Pietquin. Scaling up Mean Field Games with Online Mirror Descent. *arXiv preprint arXiv:2103.00623*, 2021.

Julien Pérolat, Sarah Perrin, Romuald Elie, Mathieu Laurière, Georgios Piliouras, Matthieu Geist, Karl Tuyls, and Olivier Pietquin. Scaling Mean Field Games by Online Mirror Descent. In *Proceedings of the 21st International Conference on Autonomous Agents and Multiagent Systems*, AAMAS '22, pp. 1028–1037, Richland, SC, 2022. International Foundation for Autonomous Agents and Multiagent Systems. ISBN 9781450392136.

Sarah Perrin, Julien Pérolat, Mathieu Laurière, Matthieu Geist, Romuald Elie, and Olivier Pietquin. Fictitious Play for Mean Field Games: Continuous Time Analysis and Applications. In *Proceedings of the 34th International Conference on Neural Information Processing Systems*, NIPS'20, Red Hook, NY, USA, 2020. Curran Associates Inc. ISBN 9781713829546.

Sarah Perrin, Mathieu Laurière, Julien Pérolat, Matthieu Geist, Romuald Élie, and Olivier Pietquin. Mean Field Games Flock! The Reinforcement Learning Way. In *IJCAI*, 2021.

Sarah Perrin, Mathieu Laurière, Julien Pérolat, Romuald Élie, Matthieu Geist, and Olivier Pietquin. Generalization in mean field games by learning master policies. In *Proceedings of the AAAI Conference on Artificial Intelligence*, volume 36, pp. 9413–9421, 2022.

Nancirose Piazza, Vahid Behzadan, and Stefan Sarkadi. The Power in Communication: Power Regularization of Communication for Autonomy in Cooperative Multi-Agent Reinforcement Learning. *arXiv preprint arXiv:2404.06387*, 2024.

Philipp Plank and Yufei Zhang. Policy Optimization for Continuous-time Linear-Quadratic Graphon Mean Field Games. *arXiv preprint arXiv:2506.05894*, 2025.

Abraham Prieto, Francisco Bellas, Pedro Trueba, and Richard J Duro. Real-time optimization of dynamic problems through distributed embodied evolution. *Integrated Computer-Aided Engineering*, 23(3):237–253, 2016.

Shreevatsa Rajagopalan and Devavrat Shah. Distributed Averaging in Dynamic Networks. In *Proceedings of the ACM SIGMETRICS International Conference on Measurement and Modeling of Computer Systems*, SIGMETRICS '10, pp. 369–370, New York, NY, USA, 2010. Association for Computing Machinery. ISBN 9781450300384. doi: 10.1145/1811039.1811091. URL https://doi.org/10.1145/1811039.1811091.

Navid Rashedi, Mohammad Amin Tajeddini, and Hamed Kebriaei. Markov game approach for multi-agent competitive bidding strategies in electricity market. *IET Generation, Transmission & Distribution*, 10: 3756–3763(7), November 2016. ISSN 1751-8687. URL https://digital-library.theiet.org/content/journals/10.1049/iet-gtd.2016.0075.

Lu Ren, Yuxin Jin, Zijia Niu, Wang Yao, and Xiao Zhang. Hierarchical Cooperation in LQ Multi-Population Mean Field Game With Its Application to Opinion Evolution. *IEEE Transactions on Network Science and Engineering*, 11(5):5008–5022, 2024. doi: 10.1109/TNSE.2024.3418832.

Naci Saldi, Tamer Başar, and Maxim Raginsky. Markov–Nash Equilibria in Mean-Field Games with Discounted Cost. *SIAM Journal on Control and Optimization*, 56(6):4256–4287, 2018. doi: 10.1137/17M1112583. URL https://doi.org/10.1137/17M1112583.

Ali Shavandi and Majid Khedmati. A multi-agent deep reinforcement learning framework for algorithmic trading in financial markets. *Expert Systems with Applications*, 208:118124, 2022. ISSN 0957-4174. doi: https://doi.org/10.1016/j.eswa.2022.118124. URL https://www.sciencedirect.com/science/article/pii/S0957417422013082.

Shigen Shen, Chenpeng Cai, Yizhou Shen, Xiaoping Wu, Wenlong Ke, and Shui Yu. MFGD3QN: Enhancing Edge Intelligence Defense against DDoS with Mean-Field Games and Dueling Double Deep Q-network. *IEEE Internet of Things Journal*, pp. 1–1, 2024. doi: 10.1109/JIOT.2024.3387090.

Hamid Shiri, Jihong Park, and Mehdi Bennis. Massive Autonomous UAV Path Planning: A Neural Network Based Mean-Field Game Theoretic Approach. In *2019 IEEE Global Communications Conference (GLOBECOM)*, pp. 1–6. IEEE, 2019.

Yu Si and Jingtao Shi. Backward Linear-Quadratic Mean Field Stochastic Differential Games: A Direct Method. *arXiv preprint arXiv:2411.18891*, 2024.

Yu Si and Jingtao Shi. Decentralized Strategies for Backward Linear-Quadratic Mean Field Games and Teams. *Optimal Control Applications and Methods*, 2025a.

Yu Si and Jingtao Shi. General Linear-Quadratic Mean Field Stochastic Differential Game with Common Noise: a Direct Method. *arXiv preprint arXiv:2506.16779*, 2025b.

Rajeshwari Sissodia, ManMohan Singh Rauthan, Varun Barthwal, and Vinay Dwivedi. Evolutionary Algorithms for Optimization and Swarm Intelligence-Based Optimization. In *Optimization Tools and Techniques for Enhanced Computational Efficiency*, pp. 17–42. IGI Global Scientific Publishing, 2025.

Javad Soleimani, Reza Farhangi, and Gunes Karabulut Kurt. Distributed Critic-Based Neuro-Fuzzy Learning in Swarm Autonomous Vehicles. In *2024 IEEE 100th Vehicular Technology Conference (VTC2024-Fall)*, pp. 1–6, 2024. doi: 10.1109/VTC2024-Fall63153.2024.10757965.

Kefan Su and Zongqing Lu. Divergence-Regularized Multi-Agent Actor-Critic. In Kamalika Chaudhuri, Stefanie Jegelka, Le Song, Csaba Szepesvari, Gang Niu, and Sivan Sabato (eds.), *Proceedings of the 39th International Conference on Machine Learning*, volume 162 of *Proceedings of Machine Learning Research*, pp. 20580–20603. PMLR, 17–23 Jul 2022. URL `https://proceedings.mlr.press/v162/su22b.html`.

Jayakumar Subramanian and Aditya Mahajan. Reinforcement Learning in Stationary Mean-Field Games. In *Proceedings of the 18th International Conference on Autonomous Agents and MultiAgent Systems*, AAMAS '19, pp. 251–259, Richland, SC, 2019. International Foundation for Autonomous Agents and Multiagent Systems. ISBN 9781450363099.

Sriram Ganapathi Subramanian, Matthew E Taylor, Mark Crowley, and Pascal Poupart. Decentralized Mean Field Games. In *Proceedings of the AAAI Conference on Artificial Intelligence*, volume 36, pp. 9439–9447, 2022.

Wei Sun and Theodore B. Trafalis. Risk-aware controller implementation for risk-sensitive mean field games through a game-theoretic differential dynamic programming approach. *International Journal of Control*, 0(0):1–9, 2025. doi: 10.1080/00207179.2025.2491820. URL `https://doi.org/10.1080/00207179.2025.2491820`.

Richard S Sutton and Andrew G Barto. *Reinforcement Learning: An Introduction*. MIT press, 2018.

Huaze Tang, Yuanquan Hu, Fanfan Zhao, Junji Yan, Ting Dong, and Wenbo Ding. M3ARL: Moment-Embedded Mean-Field Multi-Agent Reinforcement Learning for Continuous Action Space. In *ICASSP 2024 - 2024 IEEE International Conference on Acoustics, Speech and Signal Processing (ICASSP)*, pp. 7250–7254, 2024. doi: 10.1109/ICASSP48485.2024.10448058.

Rinel Foguen Tchuendom, Roland Malhamé, and Peter E. Caines. On a class of linear quadratic Gaussian quantilized mean field games. *Automatica*, 170:111878, 2024. ISSN 0005-1098. doi: https://doi.org/10.1016/j.automatica.2024.111878. URL `https://www.sciencedirect.com/science/article/pii/S0005109824003728`.

Rinel Foguen Tchuendom, Dena Firoozi, and Michèle Breton. Ranking quantilized mean-field games with an application to early-stage venture investments. *arXiv preprint arXiv:2507.00853*, 2025.

Hamidou Tembine, Raul Tempone, and Pedro Vilanova. Mean-Field Learning: a Survey. *arXiv preprint arXiv:1210.4657*, 2012.

Amoolya Tirumalai and John S. Baras. A Robust Mean-field Game of Boltzmann-Vlasov-like Traffic Flow. In *2022 American Control Conference (ACC)*, pp. 556–561, 2022. doi: 10.23919/ACC53348.2022.9867331.

Noureddine Toumi, Roland Malhame, and Jerome Le Ny. A mean field game approach for a class of linear quadratic discrete choice problems with congestion avoidance. *Automatica*, 160:111420, 2024. ISSN 0005-1098. doi: https://doi.org/10.1016/j.automatica.2023.111420. URL `https://www.sciencedirect.com/science/article/pii/S0005109823005873`.

Torsten Trimborn, Martin Frank, and Stephan Martin. Mean field limit of a behavioral financial market model. *Physica A: Statistical Mechanics and its Applications*, 505:613–631, 2018. ISSN 0378-4371. doi: https://doi.org/10.1016/j.physa.2018.03.079. URL `https://www.sciencedirect.com/science/article/pii/S0378437118303984`.

Pedro Trueba, Abraham Prieto, Francisco Bellas, and Richard J. Duro. Embodied Evolution for Collective Indoor Surveillance and Location. In *Proceedings of the Companion Publication of the 2015 Annual Conference on Genetic and Evolutionary Computation*, GECCO Companion '15, pp. 1241–1242, New York, NY, USA, 2015. Association for Computing Machinery. ISBN 9781450334884. doi: 10.1145/2739482. 2768490. URL `https://doi.org/10.1145/2739482.2768490`.

Nino Vieillard, Olivier Pietquin, and Matthieu Geist. Munchausen Reinforcement Learning. In H. Larochelle, M. Ranzato, R. Hadsell, M.F. Balcan, and H. Lin (eds.), *Advances in Neural Information Processing Systems*, volume 33, pp. 4235–4246. Curran Associates, Inc., 2020. URL `https://proceedings.neurips.cc/paper_files/paper/2020/file/2c6a0bae0f071cbbf0bb3d5b11d90a82-Paper.pdf`.

Oriol Vinyals, Igor Babuschkin, Wojciech M. Czarnecki, Michaël Mathieu, Andrew Dudzik, Junyoung Chung, David H. Choi, Richard Powell, Timo Ewalds, Petko Georgiev, Junhyuk Oh, Dan Horgan, Manuel Kroiss, Ivo Danihelka, Aja Huang, L. Sifre, Trevor Cai, John P. Agapiou, Max Jaderberg, Alexander Sasha Vezhnevets, Rémi Leblond, Tobias Pohlen, Valentin Dalibard, David Budden, Yury Sulsky, James Molloy, Tom Le Paine, Caglar Gulcehre, Ziyun Wang, Tobias Pfaff, Yuhuai Wu, Roman Ring, Dani Yogatama, Dario Wünsch, Katrina McKinney, Oliver Smith, Tom Schaul, Timothy P. Lillicrap, Koray Kavukcuoglu, Demis Hassabis, Chris Apps, and David Silver. Grandmaster level in StarCraft II using multi-agent reinforcement learning. *Nature*, pp. 1–5, 2019.

Hoi-To Wai, Zhuoran Yang, Zhaoran Wang, and Mingyi Hong. Multi-Agent Reinforcement Learning via Double Averaging Primal-Dual Optimization. In *Proceedings of the 32nd International Conference on Neural Information Processing Systems*, NIPS'18, pp. 9672–9683, Red Hook, NY, USA, 2018. Curran Associates Inc.

Bing-Chang Wang. Mean field hierarchical control for production output adjustment with noisy sticky prices. *Automatica*, 176:112260, 2025. ISSN 0005-1098. doi: https://doi.org/10.1016/j.automatica.2025.112260. URL `https://www.sciencedirect.com/science/article/pii/S0005109825001529`.

Bing-Chang Wang, Juanjuan Xu, Huanshui Zhang, and Yong Liang. Linear Quadratic Mean Field Stackelberg Games: Open-loop and Feedback Solutions. *arXiv preprint arXiv:2504.09401*, 2025a.

Haibo Wang, Hongwei Gao, Pai Jiang, Matthieu De Mari, Panzer Gu, and Yinsheng Liu. Mean Field-based Dynamic Backoff Optimization for MIMO-enabled Grant-Free NOMA in Massive IoT Networks. *arXiv preprint arXiv:2410.12497*, 2024a.

Lingxiao Wang, Zhuoran Yang, and Zhaoran Wang. Breaking the Curse of Many Agents: Provable Mean Embedding Q-Iteration for Mean-Field Reinforcement Learning. In *Proceedings of the 37th International Conference on Machine Learning*, ICML'20. JMLR.org, 2020a.

Meijiao Wang, Maoning Tang, Qiuhong Shi, and Qingxin Meng. A Variational Formula of Forward-Backward Stochastic Differential System of Mean-Field Type with Observation Noise and Some Application. *Communications on Applied Mathematics and Computation*, pp. 1–18, 2024b.

Ximing Wang, Yuhua Xu, Jin Chen, Chunguo Li, Xin Liu, Dianxiong Liu, and Yifan Xu. Mean Field Reinforcement Learning Based Anti-Jamming Communications for Ultra-Dense Internet of Things in 6G. In *2020 International Conference on Wireless Communications and Signal Processing (WCSP)*, pp. 195–200, 2020b. doi: 10.1109/WCSP49889.2020.9299742.

Xu Wang, Samy Wu Fung, and Levon Nurbekyan. A primal-dual price-optimization method for computing equilibrium prices in mean-field games models. *arXiv preprint arXiv:2506.04169*, 2025b.

Yao Wang, Chungang Yang, Tong Li, Xinru Mi, Lixin Li, and Zhu Han. A Survey On Mean-Field Game for Dynamic Management and Control in Space-Air-Ground Network. *IEEE Communications Surveys & Tutorials*, pp. 1–1, 2024c. doi: 10.1109/COMST.2024.3393369.

Samuel Wiggins, Yuan Meng, Rajgopal Kannan, and Viktor Prasanna. Characterizing Speed Performance of Multi-Agent Reinforcement Learning. *arXiv preprint arXiv:2309.07108*, 2023.

Peiliang Wu, Liqiang Tian, Qian Zhang, Bingyi Mao, and Wenbai Chen. MARRGM: Learning Framework for Multi-agent Reinforcement Learning via Reinforcement Recommendation and Group Modification. *IEEE Robotics and Automation Letters*, pp. 1–8, 2024a. doi: 10.1109/LRA.2024.3389813.

Xuesong Wu, Tianshuai Zheng, Runfang Wu, Jie Ren, Junyan Guo, and Ye Du. Hi-SAM: A high-scalable authentication model for satellite-ground Zero-Trust system using mean field game. *arXiv preprint arXiv:2408.06185*, 2024b.

Zida Wu, Mathieu Laurière, Samuel Jia Cong Chua, Matthieu Geist, Olivier Pietquin, and Ankur Mehta. Population-aware Online Mirror Descent for Mean-Field Games by Deep Reinforcement Learning. *arXiv preprint arXiv:2403.03552*, 2024c.

Na Xiang and Jingtao Shi. Robust Incentive Stackelberg Mean Field Stochastic Linear-Quadratic Differential Game with Model Uncertainty. *arXiv preprint arXiv:2507.04585*, 2025.

Qiaomin Xie, Zhuoran Yang, Zhaoran Wang, and Andreea Minca. Learning While Playing in Mean-Field Games: Convergence and Optimality. In Marina Meila and Tong Zhang (eds.), *Proceedings of the 38th International Conference on Machine Learning*, volume 139 of *Proceedings of Machine Learning Research*, pp. 11436–11447. PMLR, 18–24 Jul 2021. URL https://proceedings.mlr.press/v139/xie21g.html.

Kun Xu, Yue Li, Jun Sun, Shuyuan Du, Xinpeng Di, Yuguang Yang, and Bo Li. Targets capture by distributed active swarms via bio-inspired reinforcement learning. *Science China Physics, Mechanics & Astronomy*, 68(1):1–12, 2025a.

Linjie Xu, Zichuan Liu, Alexander Dockhorn, Diego Perez-Liebana, Jinyu Wang, Lei Song, and Jiang Bian. Higher Replay Ratio Empowers Sample-Efficient Multi-Agent Reinforcement Learning. *arXiv preprint arXiv:2404.09715*, 2024a.

Ruimin Xu, Kaiyue Dong, Jingyu Zhang, and Ying Zhou. Linear-quadratic-Gaussian mean-field games driven by Poisson jumps with major and minor agents. *AIMS MATHEMATICS*, 10(5):11086–11110, 2025b.

Runchen Xu, Zheng Chang, Zhu Han, Sahil Garg, Georges Kaddoum, and Joel J. P. C. Rodrigues. Energy-Efficient Joint Optimization of Sensing and Computation in MEC-Assisted IoT Using Mean-Field Game. *IEEE Internet of Things Journal*, 11(23):37857–37871, 2024b. doi: 10.1109/JIOT.2024.3443701.

Yue Xu, Linjiang Zheng, Xiao Wu, Yi Tang, Weining Liu, and Dihua Sun. Joint Resource Allocation for UAV-Assisted V2X Communication With Mean Field Multi-Agent Reinforcement Learning. *IEEE Transactions on Vehicular Technology*, 74(1):1209–1223, 2025c. doi: 10.1109/TVT.2024.3466116.

Chungang Yang, Haoxiang Dai, Jiandong Li, Yue Zhang, and Zhu Han. Distributed Interference-Aware Power Control in Ultra-Dense Small Cell Networks: A Robust Mean Field Game. *IEEE Access*, 6:12608–12619, 2018a. doi: 10.1109/ACCESS.2018.2799138.

Hongyi Yang, Jingzhi Liu, Geng Li, Jianming Zhang, Ling Jiang, and Shoulian Yang. Distributed Intelligent Power Distribution Optimization Method Based on Mean Field Game Theory. In *2025 IEEE 5th International Conference on Power, Electronics and Computer Applications (ICPECA)*, pp. 818–822, 2025. doi: 10.1109/ICPECA63937.2025.10928821.

Xianjin Yang and Jingguo Zhang. Gaussian Process Policy Iteration with Additive Schwarz Acceleration for Forward and Inverse HJB and Mean Field Game Problems. *arXiv preprint arXiv:2505.00909*, 2025.

Yaodong Yang, Rui Luo, Minne Li, Ming Zhou, Weinan Zhang, and Jun Wang. Mean Field Multi-Agent Reinforcement Learning. In Jennifer Dy and Andreas Krause (eds.), *Proceedings of the 35th International Conference on Machine Learning*, volume 80 of *Proceedings of Machine Learning Research*, pp. 5571–5580. PMLR, 10–15 Jul 2018b. URL https://proceedings.mlr.press/v80/yang18d.html.

Yaoqi Yang, Bangning Zhang, Daoxing Guo, Renhui Xu, Neeraj Kumar, and Weizheng Wang. Mean Field Game and Broadcast Encryption-Based Joint Data Freshness Optimization and Privacy Preservation for Mobile Crowdsensing. *IEEE Transactions on Vehicular Technology*, 72(11):14860–14874, 2023. doi: 10.1109/TVT.2023.3282694.

Zeyu Yang and Yongsheng Song. On Discounted Infinite-Time Mean Field Games. *arXiv preprint arXiv:2505.15131*, 2025.

Batuhan Yardim and Niao He. Exploiting Approximate Symmetry for Efficient Multi-Agent Reinforcement Learning. *arXiv preprint arXiv:2408.15173*, 2024.

Batuhan Yardim, Semih Cayci, Matthieu Geist, and Niao He. Policy Mirror Ascent for Efficient and Independent Learning in Mean Field Games. In *International Conference on Machine Learning*, pp. 39722–39754. PMLR, 2023.

Batuhan Yardim, Artur Goldman, and Niao He. When is Mean-Field Reinforcement Learning Tractable and Relevant? *arXiv preprint arXiv:2402.05757*, 2024.

Batuhan Yardim, Semih Cayci, and Niao He. A Variational Inequality Approach to Independent Learning in Static Mean-Field Games. *ACM/IMS Journal of Data Science*, 2025.

Bora Yongacoglu, Gürdal Arslan, and Serdar Yüksel. Independent Learning and Subjectivity in Mean-Field Games. In *2022 IEEE 61st Conference on Decision and Control (CDC)*, pp. 2845–2850, 2022. doi: 10.1109/CDC51059.2022.9992399.

Bora Yongacoglu, Gürdal Arslan, and Serdar Yüksel. Mean-field games with finitely many players: independent learning and subjectivity. *Journal of Machine Learning Research*, 25(419):1–69, 2024.

Hidekazu Yoshioka, Motoh Tsujimura, and Yumi Yoshioka. Numerical analysis of an extended mean field game for harvesting common fishery resource. *Computers & Mathematics with Applications*, 165:88–105, 2024. ISSN 0898-1221. doi: https://doi.org/10.1016/j.camwa.2024.04.003. URL https://www.sciencedirect.com/science/article/pii/S0898122124001615.

Chenyu You, Mengru Cai, Shan Yin, Honglei Wang, and Shanguo Huang. Latency-Aware Mean Field Game-Based Task Offloading Strategy in Metro Optical Networks. In *2024 Asia Communications and Photonics Conference (ACP) and International Conference on Information Photonics and Optical Communications (IPOC)*, pp. 1–4, 2024. doi: 10.1109/ACP/IPOC63121.2024.10809790.

Hanfei Yu, Jian Li, Yang Hua, Xu Yuan, and Hao Wang. Cheaper and faster: Distributed deep reinforcement learning with serverless computing. *Proceedings of the AAAI Conference on Artificial Intelligence*, 38(15): 16539–16547, Mar. 2024a. doi: 10.1609/aaai.v38i15.29592. URL https://ojs.aaai.org/index.php/AAAI/article/view/29592.

Jiajia Yu, Xiuyuan Cheng, Jian-Guo Liu, and Hongkai Zhao. Convergence Analysis and Acceleration of Fictitious Play for General Mean-Field Games via the Best Response. *arXiv preprint arXiv:2411.07989*, 2024b.

Xiang Yu and Fengyi Yuan. Time-inconsistent mean-field stopping problems: A regularized equilibrium approach. *arXiv preprint arXiv:2311.00381*, 2023.

Muhammad Aneeq Uz Zaman, Alec Koppel, Sujay Bhatt, and Tamer Basar. Oracle-free Reinforcement Learning in Mean-Field Games along a Single Sample Path. In *International Conference on Artificial Intelligence and Statistics*, pp. 10178–10206. PMLR, 2023.

Muhammad Aneeq Uz Zaman, Mathieu Lauriere, Alec Koppel, and Tamer Başar. Robust cooperative multi-agent reinforcement learning: A mean-field type game perspective. In *6th Annual Learning for Dynamics & Control Conference*, pp. 770–783. PMLR, 2024.

Sihan Zeng, Sujay Bhatt, Alec Koppel, and Sumitra Ganesh. A Single-Loop Finite-Time Convergent Policy Optimization Algorithm for Mean Field Games (and Average-Reward Markov Decision Processes). *arXiv e-prints*, pp. arXiv–2408, 2024.

Chenyu Zhang, Xu Chen, and Xuan Di. Stochastic Semi-Gradient Descent for Learning Mean Field Games with Population-Aware Function Approximation. *arXiv preprint arXiv:2408.08192*, 2024.

Jingguo Zhang and Lianhai Ren. A mean field game model of green economy. *Digital Finance*, pp. 1–36, 2024.

Kaiqing Zhang, Zhuoran Yang, Han Liu, Tong Zhang, and Tamer Basar. Fully Decentralized Multi-Agent Reinforcement Learning with Networked Agents. In Jennifer Dy and Andreas Krause (eds.), *Proceedings of the 35th International Conference on Machine Learning*, volume 80 of *Proceedings of Machine Learning Research*, pp. 5872–5881. PMLR, 10–15 Jul 2018. URL `https://proceedings.mlr.press/v80/zhang18n.html`.

Kaiqing Zhang, Yang Liu, Ji Liu, Mingyan Liu, and Tamer Basar. Distributed learning of average belief over networks using sequential observations. *Automatica*, 115:108857, 2020. ISSN 0005-1098. doi: https://doi.org/10.1016/j.automatica.2020.108857. URL `https://www.sciencedirect.com/science/article/pii/S0005109820300558`.

Kaiqing Zhang, Zhuoran Yang, and Tamer Başar. Decentralized Multi-Agent Reinforcement Learning with Networked Agents: Recent Advances. *Frontiers of Information Technology & Electronic Engineering*, 22 (6):802–814, 2021a.

Kaiqing Zhang, Zhuoran Yang, and Tamer Başar. *"Multi-Agent Reinforcement Learning: A Selective Overview of Theories and Algorithms"*, pp. 321–384. Springer International Publishing, Cham, 2021b. ISBN 978-3-030-60990-0. doi: 10.1007/978-3-030-60990-0_12. URL `https://doi.org/10.1007/978-3-030-60990-0_12`.

Shangtong Zhang and Richard S Sutton. A deeper look at experience replay. *arXiv preprint arXiv:1712.01275*, 2017.

Lianmin Zheng, Jiacheng Yang, Han Cai, Ming Zhou, Weinan Zhang, Jun Wang, and Yong Yu. MAgent: A Many-Agent Reinforcement Learning Platform for Artificial Collective Intelligence. In *Proceedings of the AAAI conference on artificial intelligence*, volume 32, 2018.

Zejian Zhou, Lijun Qian, and Hao Xu. Decentralized Multi-agent Reinforcement Learning for Large-scale Mobile Wireless Sensor Network Control Using Mean Field Games. In *2024 33rd International Conference on Computer Communications and Networks (ICCCN)*, pp. 1–6, 2024. doi: 10.1109/ICCCN61486.2024.10637582.

