# OpenReview forum: "Networked Communication for Decentralised Agents in Mean-Field Games"
_TMLR — Rejected by TMLR_

### Review · Reviewer_ChYz · 2025-09-02

**Summary Of Contributions:**

The authors introduce networked communication to the mean-field game framework. They also show that the sample guarantees of three theoretical algorithms do not translate to practical convergence times. Furthermore, their communication scheme is helpful in practical settings where the theoretical hyperparameters are not observed.

**Additional Comments:**

It is clear that the authors have put in a substantial amount of work into this paper, but it is frustrating to read as how the content within it is related to each other is barely explained. This makes me feel uncomfortable recommending acceptance for the TMLR audience, as I find, in its current form, few will be able to appreciate the authors' work and contributions. The amount of edits that would be required to bring this submission in line with what should be accepted for TMLR is prohibitive. The authors should resubmit elsewhere that solely focuses on MFG if they will not provide the necessary background.

**Audience:**

No

**Audience Explanation:**

It is borderline incomprehensible in its current format, with little connection to broader impacts. See Requested Changes.

**Claims And Evidence:**

Yes

**Claims Explanation:**

The authors provide theoretical proofs and some experimental results in simple gridworld tasks.

**Requested Changes:**

The paper seems to just start with very little introduction or explanation for why the audience should care. The issues begin with the abstract: it is a completely disorganized summary of the paper, where few will gain a good grip of what has been done. Never does it address why these findings or contributions are significant or have a practical application. For instance, this sentence, " We provide the order of
the difference in these bounds in terms of network structure and number of communication rounds, and also contribute a policy-update stability guarantee," is very confusing in its current written form. The entire abstract is just a laundry list of the contributions, but no overall connection, theme, or big picture.

Why care about MFG? It sounds extremely limited and niche if it assumes an infinite number of symmetric and anonymous agents that have identical reward and transition functions. You could also better explain terms such as MFG-NE. Why not position your paper (and abstract) by leading with MARL? Talk about the problem first, and then why your solution is so great.

You should also not assume your audience or readers are familiar with MFG. It is more reasonable to assume familiarity with MARL or RL, but it never hurts to include a section dedicated to basic background to supplement the material. Especially when this is already a long journal submission. There is really no excuse for not having more of these terms defined or even a diagram illustrating how any of this works on a high level.

I am confused why MFG-NE is said to have been used to find approx solutions for a wide variety of real-world problems, but on the same page, it is said that for such large, complex many-agent systems in the real world, it may be infeasible to find MFG-NEs analytically

When citing articles, put your citation closer to the example application: autonomous vehicles [your citations here], traffic signal control [more of your citations here], etc. not all at the end -> otherwise your reader/reviewer needs to look through 15-20 articles to find what points to where

pg. 2: why rattle off various phrases without connecting them: "i.e. similar to the above, the population is not arbitrarily reset by an external controller; model-free learning; decentralisation; fast practical convergence (Huang & Lai, 2025); and robustness to unexpected failures of decentralised learners or changes in population size (Korecki et al., 2023)."

For some reason, I am rather bothered by Footnote 1: I really dislike the referencing to a massive body of literature. It just seems bloated. Is that all the works that use this same definition? Which is the work that presented this technique? Cite that article. Then, in a related work section, you list all of these as other works that have used this method.

Your introduction (e.g., bottom of page 2 and top of page 3) is more like a related works section than an introduction. Again, this makes it harder to realize what your contributions are and why they matter. Introduction, in my opinion, starts at the bottom of page 3: "Almost all prior work relies on a centralised node to learn on behalf of all the agents." You should have led with that. Maybe include a diagram depicting this and showing the problem(s).

(Contribution #2, page 4) Exactly how novel is it to include an experience replay buffer? This seems obvious to do.

Why is anonymity important in the mean-field paradigm? This seems like something that should be explained in the paper.

When might we realistically have an infinite population that needs to be modeled in this way, for which we need to learn a single MFG-NE policy that is to be followed by the whole infinite population? Seems unrealistic and impractical

"Our algorithms are tabular rather than neural network-based" (page 7). Is this a potential shortcoming? I cannot picture how this can scale to complex tasks.

What are these constants in Assumption 3.9?

Why do we need the single-step population update operator (Definition 3.10)?

Why do we need Lemma 3.11?

(I can continue)

I cannot follow the logic behind any of the theorems, definitions, assumptions, etc. on pages 8-10. You *need* to explain your reasoning. Do not just list everything. This is completely inexcusable. For instance, see "Conservative Q-Learning for Offline Reinforcement Learning" by Kumar et al. (https://arxiv.org/abs/2006.04779) for ideas on how to write theoretical material.

---

> ### Author Response · Authors · 2025-10-20
> **Review response**
>
> Thank you very much for your time in reading our paper, and for your detailed review.
>
> You have raised many points and indicate that there are more that you did not include, but you think that ultimately a prohibitive number of edits would be required to make the work acceptable and its contributions clear. The core of your review seems to be that you found it difficult to follow the paper and to understand the area of the work and the value, both of our work and the area as a whole. You fear the rest of the TMLR audience will find the same.
>
> We hope that the enthusiastic reception of the other two reviewers will help to change your mind. Reviewer TRCd said that the paper “is very well written”, “is very nicely written in great detail”, is “very well organized and properly structured” and that the “theoretical results are clearly presented [and] properly introduced”. They also said that the paper is “strongly motivated” and “does an exceptional work on discussing related literature, with a very thorough and in-depth analysis of existing algorithms in the context of mean field games”. Reviewer PXH7 agrees that our work is “well-motivated in the context of existing literature and addresses key knowledge gaps”, and “offers a comprehensive introduction to mean-field games (MFGs), standard theoretical and empirical approaches to solving them, and challenges in the practical implementation of algorithms to find MFG Nash equilibria (MFG-NE)”.
>
> You ask “why care about MFG?”, saying it sounds like an “extremely limited and niche” area, despite it being a well-established and very active field of research. You are also concerned with the novelty and significance of elements of our contribution. Questioning the validity of an entire area with which you are not familiar feels like a somewhat unfair and speculative criticism of our work; the work should be evaluated relative to the area in which it sits, rather than needing to appeal to all readers who are not familiar with the area. Moreover, we respectfully remind you that the TMLR acceptance criteria (https://jmlr.org/tmlr/acceptance-criteria.html) state “ we explicitly avoid these terms (“significant”, “impactful”, “novel”), and focus instead on the notion of “interest”.” They say that if “there is *something* to be learned by *some* researchers *in their area* from their work, then the criterion of interest is considered satisfied”. We hope that the supportiveness and interest of the other reviewers convinces you that this applies in our case. In any case these reviewers say that our work is “creative and effective” (PXH7), “conceptually novel”, “of great theoretical interest” and “could inspire further research” (TRCd).
>
> We hope that the other reviewers’ understanding of and enthusiasm for our work persuades you that it does in fact fulfil TMLR’s acceptance criteria of being interesting to “some individuals in TMLR's audience”. In light of this, are there any more specific parts of your review that you would still like us to address or clarify, to make you feel more comfortable recommending acceptance?
>
> Thank you very much again for your efforts and consideration; we are really grateful for your time.

---

> > ### Comment · Reviewer_ChYz · 2025-10-20
> >
> > I am also able to review the other reviewers' comments. However, this does not discount my review, and also does not dismiss my questions. Please answer the questions.
> >
> > "You ask “why care about MFG?”, saying it sounds like an “extremely limited and niche” area, despite it being a well-established and very active field of research. You are also concerned with the novelty and significance of elements of our contribution. Questioning the validity of an entire area with which you are not familiar feels like a somewhat unfair and speculative criticism of our work; the work should be evaluated relative to the area in which it sits, rather than needing to appeal to all readers who are not familiar with the area."
> >
> > Then you should be able to answer that question. This is a very common question that should be addressed in research articles. I have reviewed for many respected venues and this is a standard point to address in a paper, despite how widespread or active the field is. I have seen papers justify the existence of reinforcement learning or simply state how important it is before making their contribution. In fact, the first sentence of the Introduction in the CQL paper I linked (published at NeurIPS) follows this practice.
> >
> > I also would like to refer you to the other acceptance criteria (https://jmlr.org/tmlr/acceptance-criteria.html):
> >
> > "Are the claims made in the submission supported by accurate, convincing and clear evidence?
> >
> > This is the most important criterion. This implies assessing the technical soundness as well as the clarity of the narrative and arguments presented. "
> >
> > So far, the clarity of the narrative and arguments presented is greatly lacking. I asked questions, and you have not answered.

---

> > > ### Author Response · Authors · 2025-10-27
> > > **Response part 1**
> > >
> > > Apologies, your original review suggested that no number of edits would be sufficient to change your mind, so we wanted to begin with a more general response, but we are happy to answer your questions below. Moreover, in your original review you responded ‘yes’ to the question ‘Are the claims made in the submission supported by accurate, convincing and clear evidence?’, which is why we focused on ‘Would at least some individuals in TMLR's audience be interested in knowing the findings of this paper?’, to which you replied ‘no’.
> > >
> > > ***Why care about MFG? … so great.*** The reason that we currently lead with MFGs instead of MARL is that MFGs are a game theoretical framework that has an independent scientific lineage to MARL and is inspired by theoretical physics, whereas MARL is a learning approach. MFGs are a limit theory for strategic interactions in populations that are so large that the individual identities of agents are not significant (hence anonymity), and encompasses PDEs, stochastic control and others. While RL can be used as a learning method within MFGs (as in our case), other (model-based) methods have traditionally been used, and the MFG framework has considered structural questions regarding the existence, uniqueness and stability of equilibria even without running a learning algorithm.
> > >
> > > Anonymous agents are a modelling choice that fits many systems, such as crowd motion, traffic, epidemics, electric vehicle charging and energy demand response. Nevertheless, variants have been developed beyond the classical symmetric/anonymous assumptions, including heterogeneous-type/multi-population MFGs, major-minor MFGs and graphon MFGs. An infinite population is assumed because it allows analysis to take place in the mathematical limit. As we discuss in the paper, the solution for the infinite population can be used as an approximate solution for the finite population, with the error decreasing as the size of the population increases. It might be very difficult to directly find a NE for a million agents, but this can be circumvented by finding the solution for the infinite population and then applying that back to the million agents. Moreover, once the solution is found it does not depend on the size of the deployed population, so some of the million agents could leave without requiring the rest of the population to compute a new solution, or the population could grow to 10 million and the original solution would work even better than before.
> > >
> > > MFGs have certainly been used as an approach to circumvent the scalability issues faced by MARL - this rationale is given in the second paragraph of our introduction. In the past we have indeed structured our works as you suggest, leading with MARL, but then found that readers with a MFG background thought this was alienating, as it diminished the fact that MFGs are of independent interest and not born out of MARL. Nevertheless, we are happy to swap the content of our first and second paragraphs if you think this would make a motivation for our work clearer.
> > >
> > > ***The paper seems to just … big picture.*** Similarly, we are happy to add lines to our abstract motivating our work in terms of MARL if you think this is preferable. The existing abstract is organised to follow the structure of the paper, and is intended to give a logical sequence of how to understand our contributions in relation to each other (algorithms -> theoretical analysis -> empirical modifications and studies). We respectfully remind you that significance and practical applications are not acceptance criteria for TMLR, which is why we do not highlight these in the abstract, but we are happy to do so if you prefer. We are also happy to simplify the sentence that you find confusing - it is meant to encapsulate that we don’t just bound the sample guarantees of our algorithm between those of the other architectures, but also prove how the different bounds relate to each other in terms of the communication network.
> > >
> > > ***You should also not …high level.*** In our preliminaries in Sec. 3 we incrementally build up definitions of and intuition around MFGs and MFG-NE. We also give Fig. 1 on page 5 showing a diagram of the connection between finite- and infinite-player games. Would you prefer some of these definitions to appear earlier (they are already given informally in the first paragraph of the introduction), and a diagram of something else, e.g. the fixed point of the coupled equations leading to the MFG-NE?
> > >
> > > *Continued in the next comment...*

---

> > > > ### Author Response · Authors · 2025-10-27
> > > > **Response part 2 of 3**
> > > >
> > > > *... Continued from the previous comment.*
> > > >
> > > > ***I am confused … analytically.*** We are referring to non-analytical methods for finding MFG-NEs, as discussed in the paragraphs beginning at the bottom of page 2. Classical approaches to solving MFGs were analytical, and involved finding the fixed point of the coupled forward-backward equations. While these methods were mainly theoretical, some ‘simple’ real-world problems can be modelled and solved as such, for example, some problems in financial systems, electricity markets and epidemic modelling where a model of the system is considered available. Other real-world problems are more complex and/or may not have a model available, such that while the MFG-NE cannot be found analytically, they can nevertheless be found by other approaches, e.g. numerically or via a model-free method.
> > > >
> > > > ***When citing articles … points to where.*** We originally did have our citations as you suggest, but because the TMLR format uses named citations rather than numeric ones, we found that this made the list of applications difficult to parse as each line was significantly broken up. We thought delaying citations to the end made these lines easier to read, but we are happy to revert to the original structure if preferred.
> > > >
> > > > ***pg. 2: why rattle off various phrases without connecting them*** This is intended as a list of desiderata for algorithms that learn solutions to MFGs from an empirical population in a deployed environment, i.e. we don’t want to be restricted by typical assumptions from prior works which are contrary to these desiderata, such as episodic learning, centralised learning etc. This list is in turn intended to motivate the setting of the algorithms in our work.
> > > >
> > > > ***For some reason … used this method.*** This list is intended to demonstrate the prevalance of strong, classical assumptions in the MFG literature, which our approach avoids, allowing its application to more complex environments. We are happy to remove the list or move it to the related work if deemed preferable.
> > > >
> > > > ***Your introduction ... showing the problem(s).*** The rationale behind the structure of our introduction was precisely to systematically motivate our work by presenting the narrative of the research that has been done so far in MFGs, and arguing that most of the existing work requires assumptions and methods that limits their application to certain complex deployed settings. I.e. we illustrate that existing work does not meet our stated desiderata for applicability to our setting, in order to show the contributions of our work in contrast. Nevertheless we are happy to move the paragraphs at the bottom of page 2 and top of page 3 to the related works section if you prefer. We are also happy to add a diagram of the centralised vs networked architectures if you think it would be useful.
> > > >
> > > > ***(Contribution #2, page 4) … obvious to do.*** In our setting of learning from a continued, non-episodic run of the system, in which Yardim et al. (2023) and Yongacoglu et al. (2024) are the mostly closely related works, our experience replay buffer is indeed novel. Much of the theoretical analysis in Yardim et al. (2023) centres on methods to ensure the independence of samples that are collected along this continued system trajectory and used once before being discarded. This makes the inclusion of a buffer that is cycled through repeatedly not obvious a priori. In any case, we respectfully remind you that novelty is not an acceptance criterion for TMLR.
> > > >
> > > > ***Why is anonymity … explained in the paper.*** Anonymity facilitates the beneficial scaling properties of MFGs with respect to very large populations. It allows agents to be exchangeable and their strategies symmetrical, massively reducing complexity. This allows agents to interact via the aggregate distribution instead of via individual identities, which would hinder the simplification afforded by MFGs, and in turn their scalability. We are happy to clarify this in the manuscript if deemed helpful.
> > > >
> > > > ***When might we realistically … impractical.*** As we note in the paper, the solution of the infinite-agent game can be used as an approximate solution to a game involving a large but finite number of players, for which finding the optimal solution directly may have been infeasible. Nevertheless, there are numerous problems involving extremely large populations where the infinite population itself is a reasonable modelling choice, such as traffic flows, crowd motion, financial systems, electricity markets and epidemic modelling.
> > > >
> > > > *Continued in the next comment...*

---

> > > > > ### Author Response · Authors · 2025-10-27
> > > > > **Response part 3 of 3**
> > > > >
> > > > > *... Continued from the previous comment.*
> > > > >
> > > > > ***”Our algorithms are … complex tasks.*** Yes, the tabular nature of our algorithms potentially limits scalability; our networked algorithms compare with the previous SOTA algorithms for the centralised and independent case (Yardim et al. (2023), which were also tabular. We discuss the extension to non-tabular algorithms in the future work of Section 8 at the bottom of page 29.
> > > > >
> > > > > ***What are these constants in Assumption 3.9?*** These are Lipschitz constants that give smoothness assumptions on $P$ and $R$. These in turn ensure that the population-evolution and policy-update operators are smooth and hence contractive, guaranteeing convergence. We are happy to emphasise this in the text if deemed helpful.
> > > > >
> > > > > ***Why do we need the single-step population update operator (Definition 3.10)?*** The single-step operator tells us how the mean field evolves by one step when the whole population uses a certain policy, which allows us in turn to give the stable population operator as the fixed point of repeated updates. This is plugged into the policy-improvement operator, allowing us to obtain the fixed-point consistency and hence the stationary MFG-NE. We are happy to emphasise this in the text if deemed helpful.
> > > > >
> > > > > ***Why do we need Lemma 3.11?*** By ensuring that the population updates are smooth, we can in turn ensure that they are contractive, giving a unique and stable steady population via Assumption 2. This in turn ensures convergence. We are happy to emphasise this in the text if deemed helpful.
> > > > >
> > > > > ***I cannot follow … theoretical material.*** We are happy to add similar explanations to the subsequent definitions if the ones above are deemed helpful.
> > > > >
> > > > > Thank you very much again for your time and effort in considering our paper.

---

> > > > > > ### Author Response · Authors · 2025-11-14
> > > > > >
> > > > > > Unless we are mistaken, it is nearly time to enter the final recommendation. We hope that the clarifications above are sufficient to help you support acceptance - please do let us know if there is anything else we can clarify. Thank you very much again for your time.

---

### Review · Reviewer_PXH7 · 2025-10-07

**Summary Of Contributions:**

This paper introduces networked communication to the mean-field game framework for online learning during continuous (non-episodic) system runs. On the theoretical side, the authors prove that their proposed networked algorithm achieves sample-efficiency guarantees bounded between those of centralized and fully independent learning. However, the assumptions underlying these results make learning impractically slow in practice. Therefore, the authors develop practical variants of the three algorithms by incorporating experience-replay buffers and a performance-aware policy-exchange scheme. Empirically, the networked algorithm outperforms the independent baseline and often performs comparably to the centralized learner, while avoiding the latter’s reliance on a single coordinating node.

Strengths:

(1) The work presented in the manuscript is well-motivated in the context of existing literature and addresses key knowledge gaps. The paper offers a comprehensive introduction to mean-field games (MFGs), standard theoretical and empirical approaches to solving them, and challenges in the practical implementation of algorithms to find MFG Nash equilibria (MFG-NE).

(2) A key contribution of this work is to move beyond approaches that are primarily theoretical or empirical but still reliant on unrealistic assumptions—such as oracle access or the ability to arbitrarily reset system states. The paper demonstrates empirical, online learning of mean-field game equilibria during a single continuous system run, achieving this without the limiting assumptions of traditional methods.

(3) The incorporation of experience replay buffers and a practical scheme for policy communication are straightforward in hindsight, yet creative and effective ways to begin to bridge the gap between theoretical convergence guarantees and practical implementation in real-world multi-agent settings.

(4) The empirical results are generally thorough and convincingly demonstrate the utility of replay buffers in enabling stable learning in the single-run setting.

Weaknesses/Minor Comments:

(1)  The paper does not perform a dedicated ablation on the policy-exchange rule itself. However, the comparison among the independent, networked, and centralized variants does implicitly demonstrate the value of adding communication and local policy sharing.

(2) It would be interesting to see how the empirical results hold up when the environments explicitly admit multiple equilibria. Although the `Target Agreement' environment can theoretically exhibit multiple equilibria, the experiments likely converge consistently to a single solution. The authors do not explore whether the proposed algorithm can identify or distinguish among multiple equilibrium branches, leaving open the question of how robust the approach is in genuinely multi-equilibrium settings. I do not want to propose anything too onerous but would be curious to hear some discussion/background on how this added axis of complexity affects the discovery of MFG-NEs.

(3) The performance aware $\sigma$ update is only briefly described, leaving ambiguity about its precise functional form.

(4) Minor presentation suggestion: perhaps the authors would consider lowering the alpha on the error bands in all the figures to improve readability.

**Audience:**

Yes

**Audience Explanation:**

Yes, please see "Strengths" above.

**Broader Impact Concerns:**

None.

**Claims And Evidence:**

Yes

**Claims Explanation:**

Yes, please see "Strengths" above.

**Requested Changes:**

Please see "Weaknesses/Minor Comments" above. None of the proposed adjustments are critical to securing my recommendation. They would simply strengthen the work in my view.

---

> ### Author Response · Authors · 2025-10-13
> **Review response - part 1**
>
> Thank you so much for your time and effort in reviewing our paper, and for your supportive and helpful feedback.
>
> **Weakness 1.** As you note, the independent variant is simply a special case of the communication paradigm where there is no policy exchange, making the independent case equivalent to an ablation of the policy-exchange rule. In Sec. 7.4.4 we also give an ablation on the softmax temperature parameter in the policy-exchange rule. Is there something else you had in mind by a ‘dedicated ablation on the policy-exchange rule'?
>
> **Weakness 2.** The 'target agreement' and 'cluster' tasks both *do* explicitly admit multiple Nash equilibria. In a given trial of the 'target agreement' task, all the agents could converge to remaining stationary at any one of the four corners, and any one of these four situations would lead to the highest possible returns. It is not the case that "the experiments likely converge consistently to a single solution" - we found in our experiments that with the different random seeds for each trial, agents did end up converging to a different corner at random each time. Similarly in the 'cluster' task: for a given trial all the agents could converge to remaining stationary in any one of the grid points, and any one of these $d\times d$ situations would lead to the highest possible returns. (As it happens, empirically we found that the agents usually converged at random to one of the corners in the 'cluster' task as well, rather than to anywhere on the grid. This is because in the early stages of the trial, when agents start with random policies, they already spend more time visiting corners, because at any corner three actions will keep them in place, since they cannot move off the edge of the grid).
>
> The statements above apply as long as the entropy regularisation parameter $\lambda$ is 0, as we note in the fourth bullet point of page 23. For sufficiently high $\lambda$ the Nash equilibrium is unique, and involves all the agents constantly moving about with high entropy, at the cost of biasing the problem.
>
> The discussion so far applies to *Nash* equilibria, i.e. the situations where agents end up with the highest possible returns (equivalent to a normalised average return of 10 in the plots). Population distributions can also be at an equilibrium that is not Nash nor one that receives particularly high returns: we can broadly characterise three situations here:
>
> 1. Agents, which begin the trial with random policies, never manage to reach any critical mass that breaks the ties between the possible coordination points, so continue moving about the grid with a high degree of entropy forever, even if $\lambda$ is 0. This is most likely what is happening for the independent agents across the experiments, and is why they usually converge to low returns.
>
> 2. The population gets segregated into two or more isolated parts of the grid, each of which would otherwise give a Nash equilibrium if the whole population were present e.g. half the population learns a policy that remains in the top left corner while the other half learns to stay at the bottom right. If the policies don't retain enough exploration, the agents will never discover the other isolated groups with which they could combine for mutual benefit (whilst if there is too much exploration, we revert to one of the other suboptimal situations, depending on the value of $\lambda$).
>
> 3. The population is not segregated, but oscillates between two or more locations that would otherwise represent Nash equilibria, without ever being able to settle on stable policies that agree on one location. This is similar to case 1, but with the number of meeting points that are visited having been narrowed down.
>
> Case 1 is likely to receive the worst returns. How much worse Case 2 and 3 are than the Nash equilibria depends on the size of the segregated populations / the frequency of the visitations caused by the oscillations. The ability of learning architectures to align the behaviour of the *whole* population on a *single* choice of Nash equilibrium location determines how close to the maximum return the population will receive. As already mentioned, the independent case has no way to align policies outside of the signal from the returns themselves; if no critical mass ever forms to show differentiation in the returns, then the independent population will always remain at a low performing equilibrium. The central-agent case obviously has an automatic method for aligning the policies of the whole population, but these policies may still oscillate between locations that would otherwise be Nash equilibria, which is why central-agent populations do not always reach the maximum returns in our plots.
>
> (Continued in next comment...)

---

> ### Author Response · Authors · 2025-10-13
> **Review response - part 2 of 2**
>
> (... continued from previous comment.)
>
> Our communication algorithm provides a method for both 1) aligning agents' policies, and 2) choosing better performing policies on which to align (where both of these elements contribute to the "identification"/selection of better equilibria). This is precisely why we see our networked populations receiving higher returns than the independent ones, as our algorithm helps agents to get out of the worse performing equilibria. (In principle, under the right conditions, our communication paradigm could even outperform the central-agent case: the latter aligns the population on a policy update of arbitrary quality, generated by arbitrary agent $i=1$, rather than aligning on better performing policies.) The degree to which our communication algorithm leads to policy consensus depends upon the network connectedness and the number of communication rounds. Since in our experiments we use $C=1$, it is the network connectedness - determined by the size of the broadcast radius - that has the greatest effect (for greater numbers of communication rounds, this would matter less). This is why we see the populations with higher broadcast radii converging to higher returns than populations with lower broadcast radii, which are in turn more capable than entirely independent agents - the former are better able to align the population so as to converge to equilibria that are closer to optimal (Nash) equilibria.
>
> In summary, the fact that different populations in our experiments don't just improve their returns at different speeds, but actually appear to converge to different final returns, is reflective of them settling at different equilibria that give different returns. Our communication algorithm actively helps populations to settle at equilibria that are closer to optimal (and in so doing, to choose between multiple possible Nash equilibria). So our algorithm precisely does help solve the problem raised in your question, which is what we mean when we refer to finding "preferable" equilibria in the work. We are very happy to discuss this further / clarify this in the text / tighten wording in the text, e.g. in Sec. 7.2.2. Please let us know if you have any additional questions!
>
> **Weakness 3.** As described at the top of page 19 and highlighted in orange in Algorithm 2, the performance-aware update is achieved by each agent collecting rewards for $E$ steps, which is used to calculate a $\gamma$-discounted sum of these regularised rewards. $\sigma^{i}_{k+1}$ is set to the value of this sum and then used as before from Line 24 of Algorithm 2. Can you please let us know if something remains ambiguous so we can clarify this further?
>
> **Weakness 4.** We are very happy to lower the alpha on the error bands. Additionally, we could reduce them from 2-sigma to 1-sigma intervals if deemed desirable?
>
> Thank you very much again for your time, input and positive reception; we are extremely grateful.

---

> > ### Comment · Reviewer_PXH7 · 2025-10-26
> > **Post-rebuttal comments**
> >
> > Thank you to the authors for the detailed and thoughtful responses. The clarifications were helpful and address my earlier comments. In particular, the explanation that the independent case and the temperature ablation (Sec. 7.4.4) already serve as an implicit ablation of the policy-exchange rule is reasonable. The additional discussion of multi-equilibrium behavior in the Target Agreement and Cluster tasks is informative and, if incorporated into the paper (perhaps in Sec. 7.2.2 as suggested), would strengthen the manuscript’s clarity on how the method behaves under multiple equilibria. None of these changes are critical to acceptance, but if space allows, incorporating the clarifications above would make the paper even stronger. I continue to recommend acceptance.

---

> > > ### Author Response · Authors · 2025-10-28
> > >
> > > Thank you so much for your kind and supportive feedback. We are glad the clarifications were useful, and we are more than happy to make these changes when we update the manuscript. Due to time constraints this week, we will mostly likely do that next week. Thank you again for your time and effort in approving our paper.

---

> > > > ### Author Response · Authors · 2025-11-07
> > > >
> > > > We have posted a revised version of the manuscript with 1) the alpha lowered; 2) the fact that the independent case is an implicit ablation made explicit; 3) the discussion on multiple equilibria included in Sec. 7.4.1. Thank you very much again for your help in improving our paper.

---

### Review · Reviewer_TRCd · 2025-10-17

**Summary Of Contributions:**

The work considers the problem of  mean field games, i.e., stochastic multiplayer games, where the objective is for each player, to best respond in the mean response of the population of the other players assuming that they do the same. To this end, the authors propose a learning algorithm that is based on a communication network between the players that the authors allow to vary with time. The algorithm is based on temporal difference learning updates—a common learning rule in reinforcement learning—for each player followed by a communication round where players that are connected through the network exchange information about their policies. The authors propose a single run of their algorithm and provide one theoretical version with finite sample guarantees as well as a practically applicable version using a replay buffer.  For the theoretical algorithm, the authors show that it requires in the worst case more samples than an equivalent algorithm learning a single centralized policy, and less samples than one learning the policy of each player independently.

**Strengths**
* The paper is very well written and strongly motivated
* The proposed approach appears quite interesting supported by theoretical guarantees proving its efficiency
* The experimental evaluation is very thorough

**Weaknesses**
* The results in Figures 2-9 are a bit hard to parse due to the large variance.

**Audience:**

Yes

**Audience Explanation:**

The proposed communication network in the context of the learning algorithm appears conceptually novel and quite interesting as it sets up the foundations to develop algorithms that are able to realistically capture and optimize the behavior of teams of agents in real world applications, such as e.g. drone swarms or teams of robots working in warehouses, other than just providing a method to speed-up policy learning. In addition, the theoretical perspective of the authors presenting their approach as a generalization of the centralized and independent learning algorithms seems of great theoretical interest, as it seems to provide a way for one to control the trade-off between those, by controlling the broadcasting radius of the network. Finally, the proposed network communication framework could inspire further research for multi and many agent training (e.g. setups with partial observability fro each agent and/or asymmetrical multiplayer games).

**Broader Impact Concerns:**

The are no broader impact concerns.

**Claims And Evidence:**

Yes

**Claims Explanation:**

The paper does an exceptional work on  discussing related literature, with a very thorough and in-depth analysis of existing algorithms in the context of mean field games. It is also very well organized and properly structured.  Further, it is very nicely written in great detail, strongly motivated, while also being upfront with its limitations and clearly and adequately discusses them as well as highlighting interesting avenues for future work. The theoretical  results are clearly presented, properly introduced and appear sound. The experimental evaluation appears also quite thorough, as the authors consider different tasks with smaller and larger state spaces, several evaluation metrics, as well as an ablation study for the use of the replay buffer. In addition, the experimental setup is described meticulously.

**Requested Changes:**

The greatly overlapping confidence intervals on the experimental results (especially Figures 2-9) make the results a bit cumbersome to parse of as well as hard to draw conclusions with statistical significance. The authors may like to consider increasing the number of trials of their experiments in order to reduce the variance in their metrics. (non-critical)

---

> ### Author Response · Authors · 2025-10-20
> **Review response**
>
> Thank you so much for your time in reading and reviewing our paper, and for your very kind and supportive feedback; it is much appreciated.
>
> Regarding the variance - we currently report 2-sigma confidence intervals, which are perhaps too wide: we would be very happy to reduce them to 1-sigma intervals if you think it would be helpful for clarity.
>
> Thank you so much again for your effort and input!

---

> > ### Comment · Reviewer_TRCd · 2025-10-27
> >
> > Thank you for your reply. Reducing from 2 sigma to 1 sigma would reduce the statistical significance from around 95% to 68% so perhaps it would be better to keep the 2 sigma. Lowering the alpha following Reviewer's PXH7 suggestion would be better. Though this is a minor point and not critical for acceptance.

---

> > > ### Author Response · Authors · 2025-10-28
> > >
> > > Thank you for your suggestion - we are very happy to make this change when we update the manuscript. Due to time constraints this week, we will mostly likely do that next week. Thank you again for your time and support!

---

> > > > ### Author Response · Authors · 2025-11-07
> > > >
> > > > We have posted a revised version of the manuscript with the alpha lowered. Thank you very much again for your input.

---

### Decision · Action_Editor_R8Yc · 2025-11-29

**Recommendation:** Reject

**Additional Comments:**

Reviewer ChYz critique accurately identifies critical structural issues that affect the reader’s ability to understand and evaluate the technical contribution. His comments about the unfocused narrative, overloaded introduction, and bulk citation practices reflect real barriers to comprehension that neither of the positive reviews address. More importantly, he highlights a genuine conceptual gap: the paper builds substantial theoretical machinery for algorithms that the authors themselves acknowledge do not learn in practice without replay buffers, yet the replay-based practical algorithm diverges from the assumptions underlying the stated guarantees. This disconnect is not cosmetic—it goes to the core of how the contribution should be interpreted. The other reviewers offer praise but do not contest or resolve these structural concerns; their comments are high-level and do not engage with the coherence of the exposition or the theory–experiment alignment. Reviewer ChYz, however, provides technically grounded concerns that point directly to actionable revisions the authors can implement, including adding intuition for theoretical constructs, restructuring the introduction, and clarifying the relationship between theory and practice. These issues must be addressed for the manuscript to meet TMLR/ICLR standards of clarity and accessibility. Thus, it is not meeting the bar for acceptance at this time, and a major revision is required. The authors should revise and resubmit at a later time.

**Audience:**

Yes

**Audience Explanation:**

See response to Additional comment.

**Claims And Evidence:**

No

**Claims Explanation:**

See response to Additional comment.

**Resubmission Of Major Revision:**

The authors may consider submitting a major revision at a later time.